



# A Bayesian statistical method to estimate the climatology of extreme temperature under multiple scenarios: the ANKIALE package

Yoann Robin[1], Mathieu Vrac[1], Aurélien Ribes[2], Occitane Barbaux[1,2,3], and Philipe Naveau[1]

[1]Laboratoire des Sciences du Climat et de l'Environnement, UMR 8212 CEA-CNRS-UVSQ, Université Paris-Saclay / IPSL, Sorbonne Université, Gif-sur-Yvette, 91191, France
[2]CNRM, Université de Toulouse, Météo France, CNRS, Toulouse, France
[3]Autorité de sûreté nucléaire et de radioprotection (ASNR), PSE-ENV/SCAN/BEHRIG, F-92260, Fontenay-aux-Roses, France

**Correspondence:** Yoann Robin (yoann.robin@lsce.ipsl.fr)

**Abstract.** We describe an improved method and the associated package for estimating the statistics of temperature extremes in a Bayesian framework. Building on previous work, this method uses a range of climate model simulations to provide a prior of the real-world changes, and then considers observations to derive a posterior estimate of past and future changes. The new version described in this study makes it possible to process several scenarios simultaneously, while keeping one single

counterfactual world (i.e., the world without human influence). We offer a free licensed, easy-to-use command-line tool called ANKIALE (ANalysis of Klimate with bayesian Inference: AppLication to extreme Events), which can be used to reproduce the analyses presented here, as well as to process user-defined events. ANKIALE is based on a python code, but is designed to be used from the command line interface. ANKIALE is natively parallel, enabling it to be used on a personal computer as well as on a supercomputer. The potential of this method and tool is illustrated via an application to maximum temperature

over Europe until 2100, at a 0.25°- resolution, for a range of four emission scenarios, including a particular focus on the city of Paris (France).

## 1 introduction

Heatwaves are extreme phenomena whose frequency and intensity have increased with global warming (e.g., Seneviratne

et al., 2021; IPCC, 2022a). Over recent decades, these findings have led to the development of the so-called *attribution studies*, which consists in establishing the weight of human influence in the occurrence or intensity of an extreme event (Perkins-Kirkpatrick et al., 2024). A number of methods and protocols have been developed (e.g., Ribes et al., 2020; Philip et al., 2020; Robin and Ribes, 2020a), which have enabled the analysis and attribution of a number of extreme events. The World Weather Attribution (WWA, 2024) group has made a specialty of producing attribution studies within a short time (delay of the order

of a week) following the occurrence of an event. Notable examples include the heatwave in Siberia in 2020 (Ciavarella et al.,



2021), the heatwave in the USA and Canada in 2021 (Philip et al., 2022), and the wet heatwave in India in 2023 (Zachariah et al., 2023b). Other types of event can also be analyzed, such as extreme rainfall (Zachariah et al., 2023a, 2024; Clarke et al., 2024b), drought (Clarke et al., 2024a), or wildfire (Barnes et al., 2023), and others.

The attribution methods listed above typically infer the climatology of the extremes of interest by assuming that the maxima of a variable (such as annual temperature) follow a Generalized Extreme Value distribution (GEV, see, e.g., Coles, 2001). This distribution is characterized by three parameters, which vary with external forcings (such as global or regional mean temperature). This statistical model is inferred either independently from observations, from climate models (Philip et al., 2020), or from both.

Several recent studies have proposed to implement the later option, i.e., compbining models and observations, within a Bayesian (Auld et al., 2023; Zeder and Fischer, 2023). In this context, a synthesis of climate models is used as *a prior* of the reality, and then observations are used to derive *the posterior* distribution of past and future changes (Robin and Ribes, 2020a; Ribes et al., 2022). The Bayesian approach enables the estimation of the statistics of extremes at the end of the century according to a climate scenario and conditioned on observed data over the historical period. However, inferences made without simultaneously considering multiple scenarios may lead to two potential inconsistencies.

Here, we describe several improvements to the Robin and Ribes (noted RR20, 2020a) method. Firstly, in RR20, the statistical method has to be re-run separately for each emission scenario considered, with no guarantee for consistency across scenarios. In particular, the inferred counter-factual world (i.e., the world without human influence), is different according to the scenario, leading to communication issues for key attribution diagnoses such as probability ratio. Our improved implementation enables us to infer all scenarios simultaneously, which ensures that only one counterfactual world is calculated. Second, we revise the sampling procedure – based on a Metropolis Hasting Monte-Carlo (MCMC, Metropolis et al., 1953; Hastings, 1970) method–, to make it consistent with recent progresses in the Bayesian community. This revised implementation runs much faster than the previous one, and offers many guarantees in terms of properties and convergence of the MCMC chain.

The improved method comes with a deeply revised python package and command-line tool. The original method of RR20 used a python code (Robin and Ribes, 2020b) developed for the attribution of a univariate extreme event. This code is not parallelized and required advanced knowledge of python in order to be used. Running this package over a high-resolution grid could require as long as around 20 years of CPU time. We therefore propose a new massively parallel code, developed in Python but with a command-line interface, which can process the entire domain in less than a week (in wall time), and which is designed to be more accessible.

An an illustration of the potential of our revised method and packages, we analyse extreme temperature over Europe, extended to the Mediterranean basin, giving us a box from 22°W to 45.5°E, and from 26.5°N to 72.5°N, as shown in Fig. 1a. This domain contains 54 countries, 11 of which are only partially included. The exact ratio and list are given in Tab. S1. Our attribution study focuses on the analysis of observed events, but the statistical methods and models used can describe their future evolution. Here, we focus on estimating the climatology of the strongest temperature events already observed for each grid point in Europe (see Fig. 1b).





The paper is organized as follows. In Sec. 2, we present the data used: observations, climate models, and the variables we derive from them. In Sec. 3, the methodology is presented, using extreme temperatures at the Paris location (France) as an example. Sec. 4 analyses the improvements of the new method compared to the case where the scenarios are estimated independently of each other. Section 5 describes the new code, how it is used and what is implemented. Section 6 then looks at current and future worst possible events over Europe, on the one hand using a method derived from classical attribution, and

on the other based on a specific definition. Finally, conclusions and perspectives are provided in Sec. 7.

## 2 Data used

### 2.1 Observations

We use the "European Reanalysis of the Atmosphere, version 5" data (ERA5, Hersbach et al., 2020) to characterize the historical observation-based extremes, and in the following, ERA5 will be refer to as "observations". This atmospheric reanalysis

combines data from weather forecasting models with observations using assimilation to produce a large number of atmospheric variables. The ERA5 reanalysis provides variables by pressure level at hourly time steps, with surface values obtained by interpolation.

From this dataset, we retain temperatures over our European zone, aggregated on a daily time step, taking daily maxima between 0h and 23h (UTC), from 1940 to 2024, at the spatial resolution $0.25°$. We only retain the land grid points ($\sim 52\%$ of

$185 \times 271 = 50\,135$ grid points), see Fig. 1.

Let us now take a look at how the variable representing a heatwave is constructed for each ERA5 grid point. Starting with daily maximum temperatures (TX), to account for a heatwave extending over several days, we work with the *annual maximum of the* 3-*day moving average*, noted TX3x. To illustrate our methodology we zoom over the location of Paris (France), and we use observations from a weather station located at the Paris-Montsouris site, provided by Météo-France (2023). This station

has a much longer chronology (1873 / now), which will allow us to better verify the contribution of our approach by reducing the uncertainty in the estimation of the statistical model.

For external forcing, the annual mean temperature over Europe (regional mean) or the world (global mean) is inferred from HadCRUT5 (Morice et al., 2021; Osborn et al., 2021) and GISTEMP (Lenssen et al., 2019), respectively, available from 1850 to the present day.

### 2.2 Climate models used in this study

Global Climate Models (GCMs) from the Climate Model Intercomparison Project phase 6 (CMIP6, Eyring et al., 2016) simulate climate evolution on a global scale, with a spatial resolution of the order of 100 to 200 km. The simulations feed into numerous scientific projects to understand physical mechanisms, evaluate models, lead multidisciplinary impact studies and serve as a reference for IPCC reports (see, e.g., AR6 reports, IPCC, 2021, 2022a, b).





Several emission scenarios covering the historical period (1850 / 2014) to the end of the century (2015 / 2100) are called Shared Socio-economic Pathways (SSP, O'Neill et al., 2014; van Vuuren et al., 2014; O'Neill et al., 2016), and describe climate evolution under assumptions of socio-economic evolution of human societies. Four scenarios will be used in this study, describing four levels of warming: the SSP1-2.6 (+1.8K by the end of the century with respect to 1850/1900 period), the SSP2-4.5 (+2.8K), the SSP3-7.0 (+4.1K) and the SSP5-8.5 (+5.2K) see, e.g. Ribes et al. (2021). The current trajectory takes

us towards a warming of the order of magnitude of the SSP2-4.5 scenario, estimated at +2.8K (C.I. from +2.1K to +3.4K) by the IPCC (2023).

For each model we take the same variables as for the observations: TX3x on each European grid point, mean annual temperature over Europe, and over the world. These variables cover the period from 1850 to 2100, thus including the historical part as well as the four SSPs scenarios described above.

## 95    3    Methodology

The method described here uses the same steps as the RR20 method. We therefore only describe the differences between the two approaches, and refer to (Robin and Ribes, 2020a) for the other elements.

### 3.1    The statistical model

In order to analyze how the occurrence of extreme temperature is modified in the future, we will first model their probability

distribution. The variable studied will be noted $T_t$ (it varies along time, and will represent either the TX3x or any maximum based temperatures), and its observation will be noted $T_t^o$. As $T_t$ models maxima, the statistical model inferred will be a GEV, whose parameters $\mu_t$ and $\sigma_t$ depend on a regional covariate $X_t^{\mathrm{R}}$:

$$
\begin{cases}
T_t \sim \mathrm{GEV}(\mu_t, \sigma_t, \xi_t) \\
\mu_t := \mu_0 + \mu_1 X_t^{\mathrm{R,F}} \\
\log(\sigma_t) := \sigma_0 + \sigma_1 X_t^{\mathrm{R,F}} \\
\xi_t \equiv \xi_0 \\
X_t^{\mathrm{R,F}} := X^{\mathrm{R,0}} + X_t^{\mathrm{R,N}} + X_t^{\mathrm{R,A}}
\end{cases}
\tag{1}
$$

The covariate $X_t^{\mathrm{R,F}}$ is a proxy for the temperature response to external forcings that apply to the climate system. The index F

means the Factual world, including natural and anthropogenice forcings. The covariate $X_t^{\mathrm{R,F}}$ is splitted as the sum of a constant $X^{\mathrm{R,0}}$, a response to natural forcings $X_t^{\mathrm{R,N}}$ and a response to anthropogenic forcings $X_t^{\mathrm{R,A}}$. We also add to the model defined by Eq. 1 the global mean temperature $X_t^{\mathrm{G,F}}$, constructed in a similar way to the regional temperature. This allows the effect of the global temperature on the regional temperature and local extremes to be taken into account indirectly. The observations of $X_t^{*,\mathrm{F}}$ are denoted $X_t^{*,\mathrm{o}}$. Note that this model can be seen as a linear model, where the noise of $T_t$, instead of being Gaussian,

follows the GEV distribution.





When $X_t^{*,\mathrm{F}}$ is constructed from historical forcings or scenarios, so we are in the so-called *Factual* world. Once $X_t^{*,\mathrm{F}}$ is known, we can construct its *Counterfactual* equivalent $X_t^{*,\mathrm{C}}$ (i.e. without human influence) by setting $X_t^{*,\mathrm{A}} \equiv 0$.

An important point of Eq. (1) is that the choice of factual world, counterfactual world, or the scenario in projection, are entirely defined by the term $X_t^{*,\mathrm{A}}$. The other parameters $\mu_0, \mu_1, \sigma_0, \sigma_1, \xi$ and $X_t^{*,\mathrm{N}}$ are supposed to be the same whatever the world or scenario chosen. In Robin and Ribes (2020a, for the GEV part), and Ribes et al. (2022, for the multi-covariate part), this leads to work on the following vector of parameters:

$$\theta := (\underbrace{X^{\mathrm{R},0}, X_t^{\mathrm{R},\mathrm{N}}, X_t^{\mathrm{R},\mathrm{A}}}_{\text{Regional covariate}}, \underbrace{X^{\mathrm{G},0}, X_t^{\mathrm{G},\mathrm{N}}, X_t^{\mathrm{G},\mathrm{A}}}_{\text{Global covariate}}, \underbrace{\mu_0, \mu_1, \sigma_0, \sigma_1, \xi}_{\text{GEV}}).$$

When $\theta$ is estimated on the climate scenarios independently from each other, even with the common historical part, different values can be found. In particular, the parameters may not coincide on the historical part, and the counterfactual becomes scenario dependent. We then propose the following $\theta$ vector for the scenarios SSP1-2.6, SSP2-4.5, SSP3-7.0 and SSP5-8.5 simultaneously (a more general definition is given in App. A1), which imposes the same natural forcings for all the scenarios – thus imposing a common counterfactual – and an anthropic term specific to each scenario:

$$\begin{aligned}
\theta := & (X^{\mathrm{R},0}, X_t^{\mathrm{R},\mathrm{N}}, X_t^{\mathrm{R},\mathrm{A},\mathrm{SSP1}\text{-}2.6}, X_t^{\mathrm{R},\mathrm{A},\mathrm{SSP2}\text{-}4.5}, X_t^{\mathrm{R},\mathrm{A},\mathrm{SSP3}\text{-}7.0}, X_t^{\mathrm{R},\mathrm{A},\mathrm{SSP5}\text{-}8.5}, \\
& X^{\mathrm{G},0}, X_t^{\mathrm{G},\mathrm{N}}, X_t^{\mathrm{G},\mathrm{A},\mathrm{SSP1}\text{-}2.6}, X_t^{\mathrm{G},\mathrm{A},\mathrm{SSP2}\text{-}4.5}, X_t^{\mathrm{G},\mathrm{A},\mathrm{SSP3}\text{-}7.0}, X_t^{\mathrm{G},\mathrm{A},\mathrm{SSP5}\text{-}8.5}, \\
& \mu_0, \mu_1, \sigma_0, \sigma_1, \xi) \\
= & (\theta^{\mathrm{R}}, \theta^{\mathrm{G}}, \theta^{\mathrm{GEV}})
\end{aligned} \tag{2}$$

To estimate $\theta$, we adopt the strategy initially described by Ribes et al. (2020) in the case of the Normal distribution, and extended to laws of extremes by Robin and Ribes (2020a). This procedure can be summarized as follows:

1. Fit of an estimation $\theta^m$ of $\theta$ for each climate model $m$ (see Tab. 1) in a frequentist way, with an uncertainty covariance matrix $\Sigma_{\theta^m}$, see Sec. 3.2.

2. Construct the random variable $\kappa$ of a multi-model synthesis, incorporating model uncertainty, to be used as a *a prior* of reality, see Sec. 3.3,

3. Derive a *posterior distribution* given observations, i.e. $(\kappa \mid X_t^{*,\mathrm{o}}, T_t^o)$, using Bayesian methods, see Sec. 3.4,.

In what follows, we will review the main elements of this method, detailing only the improvements and referring to Robin and Ribes (2020a) work for the parts of the methodology that have not changed.

## 3.2 Estimation in climate models

Our aim is to estimate the parameter vector $\theta^m$ defined by Eq. (2) for each climate model $m$, as well as a covariance matrix $\Sigma_{\theta^m}$ describing the uncertainty of this estimate. Recall that for each climate model $m$, and for each scenario, we have three





time series: European annual mean temperatures $X_t^{\mathrm{R,SSP}}$, Global annual mean temperatures $X_t^{\mathrm{G,SSP}}$ and annual maxima of the 3-days moving average at the Paris-Montsouris station (France) $T_t^{\mathrm{SSP}}$; SSP can be scenarios SSP1-2.6, SSP2-4.5, SSP3-7.0 or SSP5-8.5. Let us start with the covariate $X_t^{\mathrm{R,F}}$ and $X_t^{\mathrm{G,F}}$ (defining the parameters $\theta_t^{\mathrm{R}}$ and $\theta_t^{\mathrm{G}}$), which, following Robin and Ribes (2020a) are Generalized Additive Models (GAM, see e.g., Hastie, 2017):

$$140 \quad X_t^{*,\mathrm{SSP}} = X_t^{*,\mathrm{F,SSP}} + \varepsilon^* = X^{*,0} + X_t^{*,\mathrm{N}} + X_t^{*,\mathrm{A,SSP}} + \varepsilon^* \qquad (3)$$

The model is composed of the following elements:

- $X^{*,0}$: a constant;

- $X_t^{*,\mathrm{N}}$: either the response of an Energy Balance Model (Energy Balance Model, Held et al., 2010; Geoffroy et al., 2013) for CMIP5, or radiative forcings for CMIP6 (Smith, 2020), modeling natural forcings;

- $X_t^{*,\mathrm{A,SSP}}$: a smoothing spline of residuals with 6 degrees of freedom, modelling the anthropogenic forcing of the scenario SSP;

- $\varepsilon^*$: a white noise Gaussian error term describing natural variability.

In general, additive models can be inferred using *backfitting algorithms* (see, e.g., Breiman and Friedman, 1985). These methods iteratively infer the coefficients of each component on the residuals of the previous component until convergence, (see Alg. 1). Here, we have a model where the term $X^{*,\mathrm{A,SSP}}$ is specific to each scenario, but $X^{*,\mathrm{N}}$ is common, which does not exactly fit within the scope of this algorithm. To estimate the coefficients $\theta^{\mathrm{R}}$ and $\theta^{\mathrm{R}}$, we propose a modification of the backfitting algorithm, detailed in Alg. 2. We retain the iterative approach: the anthropogenic terms $X^{*,\mathrm{A,SSP}}$ are inferred for each scenario, but the natural term $X^{*,\mathrm{N}}$ is calculated on the *average (with respect to the scenarios) of the residuals*:

$$\mathrm{residual}_t := \sum_{\mathrm{SSP}} X_t^{*,\mathrm{SSP}} - X_t^{*,\mathrm{A,SSP}}.$$

This approach enables us to find an estimate $\hat{\theta}^{\mathrm{R}}$ of $\theta^{\mathrm{R}}$, and an estimate $\hat{\theta}^{\mathrm{G}}$ of $\theta^{\mathrm{G}}$.

Figure S2 shows the result of this decomposition for the global mean temperature of the IPSL model (IPSL-CM6A-LR), for four scenarios. We can see that $X_t^{\mathrm{G,F}}$ represents the mean value of the model realizations, with a 95% confidence interval encompassing these values. The peaks are due to the volcanoes, while the small oscillations are due to the sun cycles. The inferred counterfactual forcings do have a constant signal, with the exception of volcanoes and the sun.

It remains to estimate $\hat{\theta}^{\mathrm{GEV}}$, the coefficients of the GEV, as well as the covariance matrix $\hat{\Sigma}_{\theta^m}$ describing the uncertainty. We propose the following method to draw the vector $(\hat{\theta}^{\mathrm{R}}, \hat{\theta}^{\mathrm{G}}, \hat{\theta}^{\mathrm{GEV}})$ taking into account the four scenarios:

1. The following $n$-uplet is resampled:

$$(X_t^{\mathrm{R,SSP1-2.6}}, \ldots, X_t^{\mathrm{R,SSP5-8.5}}, X_t^{\mathrm{G,SSP1-2.6}}, \ldots, X_t^{\mathrm{G,SSP5-8.5}}, T_t^{\mathrm{SSP1-2.6}}, \ldots, T_t^{\mathrm{SSP5-8.5}});$$



2. Using the method described above, the vector $(\hat{\theta}^{\mathrm{R}}, \hat{\theta}^{\mathrm{G}})$ is found;

3. The vector $\hat{\theta}^{\mathrm{R}}$ is used to generate the four smoothed covariates $X_t^{\mathrm{R,F,SSP}}$;

4. These covariates are used to calculate the GEV coefficients by maximum likelihood estimation (see Robin and Ribes, 2020a, App. A) with the $T_t^{\mathrm{SSP}}$ series, generating four estimates of the $\hat{\theta}^{\mathrm{GEV}}$ vector;

5. Four vectors $(\hat{\theta}^{\mathrm{R}}, \hat{\theta}^{\mathrm{G}}, \hat{\theta}^{\mathrm{GEV}})$ are thus constructed, differing only in the GEV part. These four vectors are draws of $\theta^m$

This method is applied 1000 times, generating $4 \times 1000 = 4000$ estimates of $\theta^m$. We then define $\hat{\theta}^m$ as the mean of these
draws, and $\hat{\Sigma}_{\theta^m}$ as the empirical covariance matrix.

Figure S3 shows the location parameter $\mu_t$ (in factual and counter-factual world), and the scale parameter is represented as $\mu_t \pm \sigma_t$. We can see that the factual and counterfactual parameters correspond until the 1970s, before diverging due to external forcing. To represent the shape parameters $\xi$ (which is negative), we have added the upper bound, given by (see, e.g. Coles, 2001):

$$B_t := \mu_t - \frac{\sigma_t}{\xi}.$$

Note in particular that certain values for the second half of the 21$^{\mathrm{st}}$ century are higher than the upper limit of the counterfactual world $B_t^{\mathrm{C}}$. These values would therefore be *impossible* without the effect of the anthropic term $X_t^{*,\mathrm{A}}$.

### 3.3   Prior from a synthesis of climate models

At this stage, for each climate model we have a pair $(\hat{\theta}^m, \hat{\Sigma}_{\theta^m})$, the solution to the statistical model defined by Eq. (1). In order
to construct *a prior* of the reality, we change our point of view and move from *frequentist* statistics – where $\theta$ is the estimate of the parameters, and $\Sigma_\theta$ is the uncertainty – to *Bayesian* statistics where $\theta$ is a random variable. We therefore assume that $\theta^m$ is a random variable with the following normal distribution:

$$\theta^m \sim \mathcal{N}(\hat{\theta}^m, \hat{\Sigma}_{\theta^m}).$$

A prior of the reality is constructed as a synthesis of different climate models using the following hypothesis: *models are*
*statistically indistinguishable from reality*, developed by Ribes et al. (2017). Denote $\kappa \sim \mathcal{N}(\nu_\kappa, \Sigma_\kappa)$ the random variable of the multi-model synthesis we are looking for – to simplify, $\kappa$ is a vector with the same components as $\theta$ –, and $\mathcal{N}(\nu_\eta, \Sigma_\eta)$ the "reality". Let's start by decomposing the $\theta^m$ of each model $m$ as follows:

$$\theta^m \sim \mathcal{N}(\nu_\eta + \nu^m, \Sigma_u + \Sigma^m)$$

$$\nu^m \sim \mathcal{N}(0, \Sigma_u).$$

Here $\nu_\eta + \nu^m = \hat{\theta}^m$ is the mean of model $m$ (and $\nu_m$ is the bias of the model with respect to the truth), $\Sigma_u$ is a common
part of the internal variability shared by all models and $\Sigma^m$ is the additional internal variability of model $m$. The multi-model synthesis is itself a normal distribution that takes into account the modelling uncertainty, and the intra- and inter-model





uncertainty. Intuitively, the uncertainty of the synthesis (defined by the covariance matrix of $\kappa$) "covers" the spread of the models. The multi-model synthesis is given by the following equations, and the calculation method is described in Robin and Ribes (2020a, App. B):

$$
\begin{cases}
\nu_\kappa = \dfrac{1}{n}\sum_m \hat{\theta}^m = \dfrac{1}{n}\sum_m (\nu_\eta + \mu^m) \\[2mm]
\Sigma_\kappa = \left(1 + \dfrac{1}{n}\right)\hat{\Sigma}_u + \dfrac{1}{n^2}\sum_m \hat{\Sigma}^m.
\end{cases}
\tag{4}
$$


Figure 2 illustrates the effect of this synthesis for the grid point containing Paris. In Fig. 2a, we have represented the factual and counter-factual world (in red and blue) of the mean of each model (historical and SSP5-8.5) of the regional covariate Europe (the lines), as well as the 95% confidence interval of the multi-model synthesis (filled zone). This interval contains all the means of each model, showing that the prior is a representative of all the possibilities of these models for the covariate.

Figures 2c-f show the level line at 95% of the covariance matrix between the parameter $\mu_0$ and the other parameters $\mu_1$, $\sigma_0$, $\sigma_1$ and $\xi_0$. The grey ellipses are each of the models, while the black ellipse is the multi-model synthesis. This synthesis again covers most climate models, with the exception of one which is very different. As the weights are uniform across the models during the synthesis, this model has been excluded from the synthesis.

### 3.4 Posterior from observations

#### 3.4.1 Bayesian method

At this stage, we have the random variable $\kappa$ (the multi-model synthesis), and our aim is to use it as the prior of the observation parameters. Recall that we have observations of the local extremes $T_t^o$, the global variable $X_t^{\mathrm{G,o}}$, and the regional $X^{\mathrm{R,o}}$. Our goal is to estimate the distribution $\mathbb{P}\left(\kappa \mid T_t^o \cap X_t^{*,o}\right)$. Following the same calculations as Robin and Ribes (2020a, ,Sec. 3.5), we have the relationship:

$$
\mathbb{P}\left[\kappa \mid \left(T_t^o \cap X_t^{*,o}\right)\right] = \frac{\mathbb{P}\left[T_t^o \mid (\kappa|X_t^{*,o})\right]\mathbb{P}(\kappa \mid X_t^{*,o})}{\mathbb{P}(T_t^o)}.
\tag{5}
$$

In other words, we start from the prior $\kappa$, from which we derive the posterior $(\kappa \mid X_t^{*,o})$ from the observations $X_t^{\mathrm{G,o}}$ and $X_t^{\mathrm{R,o}}$. For this first posterior, only the global and regional covariates are constrained. Equation (5) indicates that the posterior $(\kappa \mid X_t^{*,o})$ can also be seen as the prior allowing the parameters of the GEV distribution to be constrained by the $T_t^o$ observations. We will now look at how these two successive posteriors are constructed.

#### 3.4.2 Simultaneous constraint of external forcings

To determine the posterior of global and regional temperatures, our aim is to apply the Gaussian conditioning theorem (see, e.g., Eaton, 2007). Starting from the prior $\kappa$ and the observations $X^{*,o}$, if we find a matrix $A$ such that:

$$
X_t^{\mathrm{GR,o}} := \begin{pmatrix} X_t^{\mathrm{G,o}} \\ X_t^{\mathrm{R,o}} \end{pmatrix} = A \cdot \kappa + \varepsilon^o.
\tag{6}
$$





Then as $\kappa \sim \mathcal{N}(\nu_\kappa, \Sigma_\kappa)$, according to the Gaussian conditioning theorem, the random variable $(\kappa \mid X_t^{*,\mathrm{o}})$ is also a normal distribution with the following parameters:

$$\begin{cases} \nu_{(\kappa \mid X_t^{*,\mathrm{o}})} = \nu_\kappa + (\Sigma_\kappa A^T) \cdot (A \Sigma_\kappa A^T + \Sigma_o)^{-1} \cdot (X_t^{\mathrm{GR,o}} - A\nu_\kappa) \\ \Sigma_{(\kappa \mid X_t^{*,\mathrm{o}})} = \Sigma_\kappa - (\Sigma_\kappa A^T) \cdot (A \Sigma_\kappa A^T + \Sigma_o)^{-1} \cdot (A\Sigma_\kappa). \end{cases} \tag{7}$$

We propose to give the matrix $A$ directly, and to explain briefly why it is of this form. The details are given in the App. A2. The matrix $A$ we are looking for is of the following form:

$$A := \begin{pmatrix} R^{\mathrm{G,o}} & R^{\mathrm{R,o}} \end{pmatrix} \cdot \begin{pmatrix} A^1 & A^0 & A^{\mathrm{GEV}} \\ A^0 & A^1 & A^{\mathrm{GEV}} \end{pmatrix},$$

with the following sub-matrix:

- $A^1$ is a matrix which transforms $\kappa^{\mathrm{G}}$ or $\kappa^{\mathrm{R}}$ into the mean covariate, taken along the SSPs. In other words for $\kappa^{\mathrm{G}}$:

$$A^1 \cdot \kappa^{\mathrm{G}} = \frac{1}{N^{\mathrm{SSP}}} \sum_{\mathrm{SSP}} X_t^{\mathrm{G,F,SSP}}$$

- $A^0$ is the same dimension as $A^1$, but with null values.

- $A^{\mathrm{GEV}}$ is a null matrix such that $A^{\mathrm{GEV}} \cdot \kappa^{\mathrm{GEV}} = 0$.

- $R^{\mathrm{G,o}}$ are $R^{\mathrm{R,o}}$ are matrices which restrict the time axis to that of the observations.

We then have:

$$A \cdot \kappa + \varepsilon^o = \begin{pmatrix} R^{\mathrm{G,o}} & R^{\mathrm{R,o}} \end{pmatrix} \cdot \begin{pmatrix} A^1 & A^0 & A^{\mathrm{GEV}} \\ A^0 & A^1 & A^{\mathrm{GEV}} \end{pmatrix} \cdot \underbrace{\begin{pmatrix} \kappa^{\mathrm{G}} \\ \kappa^{\mathrm{R}} \\ \kappa^{\mathrm{GEV}} \end{pmatrix}}_{=\kappa} + \underbrace{\begin{pmatrix} \varepsilon^{\mathrm{G,o}} \\ \varepsilon^{\mathrm{R,o}} \end{pmatrix}}_{\varepsilon^o}$$

$$= \begin{pmatrix} R^{\mathrm{G,o}} & R^{\mathrm{R,o}} \end{pmatrix} \cdot \begin{pmatrix} A^1 \cdot \kappa^{\mathrm{G}} + A^0 \cdot \kappa^{\mathrm{R}} + A^{\mathrm{GEV}} \cdot \kappa^{\mathrm{GEV}} \\ A^0 \cdot \kappa^{\mathrm{G}} + A^1 \cdot \kappa^{\mathrm{R}} + A^{\mathrm{GEV}} \cdot \kappa^{\mathrm{GEV}} \end{pmatrix} + \varepsilon^o$$

$$= \begin{pmatrix} R^{\mathrm{G,o}} \cdot A^1 \cdot \kappa^{\mathrm{G}} \\ R^{\mathrm{R,o}} \cdot A^1 \cdot \kappa^{\mathrm{R}} \end{pmatrix} + \varepsilon^o$$

$$= \begin{pmatrix} R^{\mathrm{G,o}} \cdot \left( \frac{1}{N^{\mathrm{SSP}}} \sum_{\mathrm{SSP}} X_t^{\mathrm{G,F,SSP}} \right) \\ R^{\mathrm{R,o}} \cdot \left( \frac{1}{N^{\mathrm{SSP}}} \sum_{\mathrm{SSP}} X_t^{\mathrm{R,F,SSP}} \right) \end{pmatrix} + \varepsilon^o = \begin{pmatrix} X_t^{\mathrm{G,o}} \\ X_t^{\mathrm{R,o}} \end{pmatrix} = X_t^{\mathrm{GR,o}}$$

Note that other choices for $A^1$ exists. Here we have made the assumption that over the observed period, the average of the scenarios differs from the observations only by white noise. If we accept as an additional hypothesis that a particular scenario is similar to the observations (such as SSP2-4.5), then we can take for $A^1$ the matrix that restricts to this scenario, i.e.:

$$A^1 \cdot \kappa^* = X_t^{*,\mathrm{F,SSP2\text{-}4.5}}.$$





Here we prefer to remain more general and take for $A^1$ the matrix which uses the average of the scenarios.

A tricky point is the estimation of $\varepsilon^o \sim \mathcal{N}(0, \Sigma_o)$. To do this, we remove the trend of the prior from the observations, i.e. we calculate $X_t^{*,o} - A\nu_\kappa$, and $\Sigma_o$ is the variance of the residuals.

We can see the effect of this constraint in Fig. 2b: the confidence interval of the constrained variable is much tighter around the median (a reduction of 2K on each side). The covariate Europe passes through the observations (the black dots). We can also see a reduction in the confidence interval of the counterfactual covariate, which depends only on the natural terms that have also been constrained. These results are consistent with the work of Qasmi and Ribes (2022).

### 3.4.3 Metropolis-Hasting method for local extremes

With our knowledge of the distribution $(\kappa \mid X_t^{*,o})$, we now want to obtain samples of the distribution $\mathbb{P}\left[(\kappa \mid X_t^o) \mid T_t^o\right]$. The whole problem is that $T_t$ follows a GEV distribution, and there is no explicit expression for the posterior. Let us start again from the Eq. (5):

$$\mathbb{P}\left[(\kappa \mid X_t^{*,o}) \mid T_t^o\right] = \frac{\mathbb{P}\left[T_t^o \mid (\kappa \mid X_t^{*,o})\right]\mathbb{P}(\kappa \mid X_t^{*,o})}{\mathbb{P}(T_t^o)}.$$

The distribution $\mathbb{P}(\kappa \mid X_t^{*,o})$ is known, it is our prior. The term $\mathbb{P}\left[T_t^o \mid (\kappa|X_t^o)\right]$ is directly calculable: the draws of $(\kappa|X_t^o)$

generate the parameters of the GEV law, which can thus be evaluated. When the denominator is analytically intractable, numerical methods are necessary to sample from the posterior distribution.

A common approach to perform this sampling is the Metropolis-Hasting algorithm (Metropolis et al., 1953; Hastings, 1970) This is the sampling algorithm originally used by Robin and Ribes (2020a). This Markov chain Monte Carlo algorithm relies on a random walk proposal: a new proposal is created by starting from an initial value $\kappa_0$ and adding a random noise to generate

a $\kappa_1$. The new value is either accepted or rejected with a probability defined using the likelihood ratio of the proposal and the previous value. A key element of this procedure is the transition function between $\kappa_i$ and $\kappa_{i+1}$ that is used to sample successive possible values of the posterior.

In the Robin and Ribes (2020a) original implementation, the transition function was of the form $\kappa_{i+1} = \kappa_i + \varepsilon$ where $\varepsilon$ follows a normal distribution with the same scale for all parameters. This can become an issue when the scale of the target parameters is very different from one another. The transition also determines the rate of convergence and mixing, so this

implementation can be computationally sub-optimal. Various diagnostics showed the algorithm suffered from slow-mixing chains (Gelman et al., 1997), high autocorrelation (Brooks et al., 2011), and low effective sample size (Gelman et al., 2015).

To deal with these issues, we leverage the *No-U-Turn Sampler* algorithm (NUTS, Hoffman and Gelman, 2014), as implemented in STAN (Stan Development Team, 2024). This algorithm is based on the Hamiltonian Monte Carlo algorithm (Radford,

2011), a variant of the Metropolis-Hasting algorithm where the proposal is not generated using a random walk. Instead, the proposal is created through a series of gradient-informed steps (Betancourt, 2018). This allows for better parameter space exploration, especially in the multidimensional case. The NUTS variant relies on a specific criteria to select adaptively various hyper-parameters such as the steps length and stopping conditions. This adaptation makes the algorithm more robust against





correlation in the posterior. The NUTS algorithm is particularly effective when the posterior dimensions are correlated or of

different scales. It is very efficient to explore the parameter space and draw samples from the posterior.

The effect of the constraint can be seen in figures 2c/f. First of all, let's note that the posterior is itself Gaussian (in this case), so we have show green ellipses representing the 95% confidence level of the parameters $\mu_0$, $\mu_1$, $\sigma_0$, $\sigma_1$ and $\xi_0$. For comparison, we have also added the direct inference of observations in orange with a maximum likelihood estimate, the covariate being that of our posterior. The 95% confidence interval is constructed by bootstrap. We can see that the posterior is systematically within

the uncertainty of both the prior and the observations. The median is the same for all the parameters, except for $\mu_1$ which is twice as large for the observations, with a large uncertainty.

## 4 Comparison between the independent and dependent scenarios method

We propose now to compare the result of our calculation between the case where the scenarios are estimated independently and the case where the scenarios are estimated simultaneously. In other words, we carry out the estimates either for each scenario

separately as in Robin and Ribes (2020a), or as developed above. Based on the estimates of the laws, the attribution of the 2019 heatwave is performed for comparison purpose. First, let's recall that in the case of the $\mathrm{GEV}(\mu_t, \sigma_t, \xi)$ law, the survival and quantile functions are given explicitly:

$$\mathbb{P}_t(T > I) = 1 - \exp\left[-\left(1 + \xi \frac{I - \mu_t}{\sigma_t}\right)^{-1/\xi}\right],$$

$$\mathcal{Q}_t(p) = \mu_t + \frac{\sigma_t}{\xi}\left((-\log p)^{-\xi} - 1\right).$$

In an attribution context, an event occurs in year $t = \tau (= 2019)$ with an intensity $I_\tau^{\mathrm{F}}$, and we look for the probabilities that $T_t$

exceeds this intensity in the factual (noted $p_t^{\mathrm{F}}$) and counterfactual (noted $p_t^{\mathrm{C}}$) world for all years $t$ (including $t = \tau$). This is written:

$$\begin{cases} p_t^{\mathrm{F}} = \mathbb{P}_t^{\mathrm{F}}(T_t \geq I_\tau^{\mathrm{F}}), \\ p_t^{\mathrm{C}} = \mathbb{P}_t^{\mathrm{C}}(T_t \geq I_\tau^{\mathrm{F}}). \end{cases}$$

Each year $t$, we can also define the intensity of an event that has the same probability as that of exceeding the intensity $I_\tau^{\mathrm{F}}$ that was observed in year $\tau$:

$$\begin{cases} I_t^{\mathrm{F}} = \mathcal{Q}_t^{\mathrm{F}}(1 - p_\tau^{\mathrm{F}}), \\ I_t^{\mathrm{C}} = \mathcal{Q}_t^{\mathrm{C}}(1 - p_\tau^{\mathrm{F}}). \end{cases}$$

Note that this definition ensures that $I_{t=\tau}^{\mathrm{F}} = I_\tau^{\mathrm{F}}$. We can thus deduce the change in probability and intensity due to human influence $\mathrm{PR}_t$ and $\Delta I_t$, as (see, e.g. Hannart et al., 2016):

$$\begin{cases} \mathrm{PR}_t = \dfrac{p_t^{\mathrm{F}}}{p_t^{\mathrm{C}}}, \\ \Delta I_t = I_t^{\mathrm{F}} - I_t^{\mathrm{C}}. \end{cases}$$





The probabilities in the factual and counterfactual world $p_t^{\mathrm{F}}$ and $p_t^{\mathrm{C}}$, and the intensities $I_t^{\mathrm{F}}$ and $I_t^{\mathrm{C}}$ for the two cases (dependent

and independent) and four scenarios have been plotted with their 95% intervals in Fig. S4. For each of these indicators, we
generated 5000 trajectories from the posterior, i.e. 5000 realizations per year, from which we estimated the distributions. To
verify the contribution of our methodology, we calculate for the independent case and the simultaneous case, for each pair of
scenarios, and each year, the difference between the obtained distributions in the following way:

   – For the factual world, these differences are calculated over the historical period (1850/2014);

– For the counter-factual world, the entire series is used;

   – The metric for calculating the difference between distributions is the Wasserstein distance, taken from optimal transport (see,
     e.g., Santambrogio, 2015; Robin et al., 2019). This distance represents the average energy required to transform one
     distribution into another.

The calculated distances are shown in Fig. 3. The first column corresponds to the independent case, the second to the simultaneous

or dependent case, the last being the relative difference representing the percentage reduction in inter-distances: the smaller
the inter-distances, the more similar the different counter-factual and historical scenarios. In all cases, the distances decrease
by 80% or even 100%, showing a significant improvement in consistency between the historical scenarios and between the
counterfactuals. We note that pairs with a slight improvement or even deterioration already have low inter-distances. We can
also see that in the independent case, inter-distances are altered by natural forcings (peaks), an alteration that has disappeared

with our new methodology.

Note also that we have compared the GEV coefficients on Fig. S5. We can see that the SSP3-7.0, if estimated independently,
is slightly ahead of the other scenarios, especially for the $\sigma_1$ parameter. Our approach therefore ensures consistency between
the scenarios We also see that the posterior retains a Gaussian structure, allowing us to approximate it by a normal distribution.

To summarize, our new approach does a good job at ensuring historical and counterfactual consistency between the different

climate scenarios.

## 5   ANKIALE: ANalysis of Klimate with bayesian Inference: AppLication to extreme Events

The original method, proposed by Robin and Ribes (2020a), was accompanied by a package written in python (Van Rossum
and Drake, 2009) or R (R Core Team, 2024): *Non-Stationary Statistics for Extreme Attribution* (NSSEA, Robin and Ribes,
2020b) to reproduce their results. Although this package can be used for attribution studies, the construction of its non-parallel

code is not suitable for the simultaneous analysis of several thousand grid points, as is the case for a domain the size of Europe.
Furthermore, its use requires in-depth knowledge of either the Python language or the R language.

We are therefore proposing a new package, which although written in Python, is presented as a command line tool that can
be called in a bash script with the command `ank`. The architecture of the package is described in Sec.5.1. The various steps
in Sec. 3 are broken down into sub-commands allowing them to be estimated, and are described in Sec. 5.2. Examples are

provided within the package, allowing to reproduce the results presenting in this paper.





## 5.1 Architecture

The ANKIALE package contains two main classes: `ANKParams` which contains the computer parameters (temporary directories, number of CPUs, amount of memory, etc) and `Climatology` which describes the $\theta$ law. These two classes are instantiated when ANKIALE is launched. The first by the parameters of the user and the configuration of the system, the second either by a file passed by the user, or it is waiting to be built. The sub-module `ANKIALE.stats` then contains the classes and functions necessary for the estimations of $\theta$:

- Class `ANKIALE.stats.MultiGAM`: inference of the covariates,

- Function `ANKIALE.stats.nslaw_fit`: maximum likelihood estimation, this function is generic and accepts the different laws grouped in the sub-module `ANKIALE.stats.models`. Note that minimisation calls the external package SDFC (Statistical Distribution Fit with Covariates Robin, 2020).

- Function `ANKIALE.stats.synthesis`: to build the multi-model synthesis.

- Function `ANKIALE.stats.gaussian_conditionning`: application of the Gaussian conditioning theorem.

- As explained in Sec 3.4, the bayesian constraint uses the STAN (Stan Development Team, 2024) tool, which is used by default. It is possible to revert to the original algorithm with the `--no-STAN` option.

Furthermor, the display functions are grouped in the sub-module `ANKIALE.plot`, the commands in the sub-module `ANKIALE.cmd` and the data in the sub-module `ANKIALE.data`.

## 5.2 Package commands

**ank --help** Displays the documentation.

**ank fit** Starts the estimation of $\theta$ in the models (see Sec. 3.2). The models should be netcdf files of dimension (`time`, `period`, `run`), where `time` is the time axis, `period` the scenarios (historical and SSPs) and `run` the different members available. Additional dimensions can be added, representing spatial coordinates (e.g. latitude and longitude). The $\theta$ parameters are saved as a netcdf file containing the mean and covariance matrix for each estimated spatial dimensions.

**ank synthesize** Performs the multi-model synthesis calculation described in Sec. 3.3. All the netcdf files produced by the previous command must be supplied.

**ank constrain** Starts the observation-based constraint estimation described in Sec. 3.4, from the output file of the previous command.

**ank attribute** Starts an attribution by imposing either an event or a return time.

**ank draw** Draws $\theta$ parameters, and constructs the parameters of the statistical model given by Eq. (1).

**ank show** Construct figures to analyze the different stages of the method.

**ank example** Places in a directory ready-to-use examples including data and scripts. Currently the following examples are supported:



- GSMT, global warming estimation, allowing to reproduce the Fig. S1.

- Paris-Montsouris, estimation and attribution of TX3x at Paris-Montsouris French station, allowing to reproduce the example used in the Sec. 3,

- Ile-de-France, estimation and attribution of TX3x over Ile-de-France, France, with ERA5, allowing to reproduce a sub-part of the results presented in the Sec. 6.

**Optional arguments** The optional arguments `--n-workers` and `--total-memory` allow to user to control the number of CPUs to be used, as well as the memory available. The parallelization and memory management tools are based on the package dask (automatic parallelization, Dask Development Team, 2016) as well as zarr (temporary files on disk to minimise

memory usage, Miles et al., 2024).

## 6    Highest temperatures in Europe

In order to study how the observed maxima behave (see Fig. 1b) and could behave in the future, we propose to carry out their attribution. Classically, attributions, such as those carried out by the WWA (see, e.g., Ciavarella et al., 2021; Philip et al., 2022; Zachariah et al., 2023b), consider as a statistical variable the average of a climate variable (temperature, heat index,

precipitation, etc.) over a domain (geographical area, country), and study an observed event and its impacts. For us, on the one hand, each ERA5 grid point in our domain will be a variable to be analyzed, and, on the other hand, we are not analyzing a specific event. No form of spatial dependency is taken into account, so the existence of an event at one place does not imply anything at another.

     We start by looking at the current state of return durations and intensity change, see Sec. 6.1. We then continue with the near

future in 2040, see Sec. 6.2. We finish with the end of the 21st century, see Sec. 6.3.

### 6.1    Current situation: 2024

Figure 4 shows the return times of the maximum observed in 2024 since 1940 in the counterfactual and factual world (Figs 4a,b), as well as the change in intensity (Fig. 4c). The 95% confidence intervals are given in figures S6 and S7.

     We can see that the counterfactual world shows return times (Fig 4a) greater than 1000 years over almost the whole of

Europe, showing that the maxima currently recorded are almost impossible without anthropogenic climate change. The 95% confidence interval shows values down to 30 years over North Africa, Central Europe and Northern Europe, but almost the entire domain shows return periods of the order of at least 500 years.

     In the factual world, North Africa shows return periods of 2 to 5 years (Fig 4b), whereas in the counter-factual world they were in excess of 1000 years, showing that near-impossible events are currently becoming the new standard in this part of

Europe. The same phenomenon can be seen over Western and Southern Asia, with equivalent values. The 95% confidence intervals show the same phenomenon. Generally speaking, with the exception of part of England, Belgium and Russia, the





entire domain shows return periods for maximums of up to 100 years, and down to 10 years, which shows that maximums, which are supposed to be records (and therefore rare), are becoming the norm.

The temperature increase from the conter-factual to the factual world (Fig 4c) is fairly uniform across the domain, with values around +2K. The change is nevertheless marked in Northern Europe, with values of around +1.5K. The signal remains clearly positive, with the low value of the confidence interval around +1.5K, and falling to +0.6K in Northern Europe. The high end of the confidence interval is closer to +3K, with peaks at +3.7K.

In line with all the studies on the attribution of extreme temperatures, it is clear that anthropogenic climate change implies a sharp increase in extreme temperatures. The sign of this change is unambiguous, as the low value of the confidence interval does not include a zero or negative change.

## 6.2 Mid-term: 2040

First and third rows of the Fig. 5 show the return times in 2040 in the counterfactual and factual world (Fig. 5a/e), as well as the change in intensity (Fig. 5k-n). The 95% confidence intervals are given in figures S8a/e,k/n and S9a/e,k/n. Table 2 gives a summary of the statistics by scenario and country. The 95% confidence interval is given in tables S2 and S3.

For return times, the counterfactual world (Fig. 5a) is the same as that in 2024 (Fig. 4a), and shows return times of over 1000 years. The SSP2-4.5, SSP3-7.0 and SSP5-8.5 scenarios (Fig. 5c/e) are extremely close to each other, with values between 10 and 30 years over most of Europe, falling to 1 to 2 years over North Africa. Overall, the events are more likely than at present, reflecting the rise in temperatures over 16 years. However, these 3 scenarios have not yet been differentiated, unlike SSP1-2.6, which shows slightly longer return periods. Northern France, Belgium, Great Britain and Russia also show slightly longer return periods, between 50 and 500 years. The 95% confidence interval (figures S8a/e and S8a/e) shows a similar message: the three scenarios SSP2-4.5, SSP3-7.0 and SSP5-8.5 are extremely close, and SSP1-2.6 has slightly longer return times.

The change in intensity in 2040, visible in Fig. 5k-n, shows, similarly to the return times, very close values – around +3K to +3K – for the three scenarios SSP2-4.5, SSP3-7.0 and SSP5-8.5, and a scenario SSP1-2.6 with lower intensity changes of around 0.5K. Northern Europe shows lower values, below +2K, while Eastern and Southern Europe reach almost +4K. The 95% confidence intervals (figures S8k-n and S9k-n) show a similar spatial dispersion of values, with 1K lower values for the lower bound, and 1K higher values for the upper bound. Some countries even show changes of more than +5K.

## 6.3 Long-term: 2100

Fig. 5f shows the return times in 2100 of the maximums observed in the counterfactual world. Almost the entire domain shows return periods of over 1000 years, with the exception of central Eastern Europe, western North Africa, the far north of Northern Europe and Ireland, where return periods of up to 100 years are rare.

Let us continue with the scenarios, represented on the Fig. 5g/j. Return times decrease with simulated climate change intensity. For SSP1-2.6, only 12 countries (on 55) show return times beyond 50 years, with 28 countries already having a return time of 10 years or less. From SSP2-4.5 onwards, the current maximums are almost commonplace, with only one country showing a return period in excess of 50 years. From SSP3-7.0 onwards, current maximums are the "normal" situation,



with return times between 1 to 10 years and 1 to 2 years. The 95% confidence interval is shown in figures S8g/d and S9g/d, and shows that return periods can fall below 10 years over the whole of Europe, making annual events supposed to be centennial.

    The scenarios for intensity change are represented on the Fig. 5o/r. For scenario SSP1-2.6, the change ranges from +1.7K for Northern Europe to +4.1K for Southern Europe, with the change relative to 2024 being almost the same everywhere, around +1K. For the SSP2-4.5 scenario, the change ranges from +2.7K for Northern Europe to +6.5K for Southern Europe.

The different regions of Europe show different changes compared to today, ranging from +1.5K to +3.8K. The SSP3-7.0 and SSP5-8.5 scenarios show increasing intensity increases, from +5K to +9K. The 95% confidence interval (figures S8o/r and S9o/r) even shows changes in intensity of up to +18K.

## 7   Conclusions and perspectives

### 7.1   Conclusions

In this paper, we have presented an extension of the Robin and Ribes (2020a) method for estimating probabilities of extremes following a GEV law. Our new method allows us, on the one hand, to treat several scenarios simultaneously, and on the other, to force a counter-factual world that is common to all scenarios. We first applied this method to temperature extremes over Paris (France), and demonstrated not only its validity, but also that it drastically reduces differences in the historical part, reinforcing the inter-scenario consistency of our estimates. We have also verified that our estimates of current and future climate change

are consistent with current literature.

    We also offer an open-source software that can be easily used to reproduce our results and easily applied to other fields. This tool is natively parallelized, with particular attention paid to the memory used. It can be deployed just as easily on a personal computer, a computing cluster or a supercomputer. This software is also extensible, and other probability distributions – such as the Normal or Generalized Pareto Distribution – may be integrated in the future.

We have applied this new approach and tool to the attribution of observed maxima over Europe, enabling us to analyze these statistics up to the end of the 21st century for four climate scenarios. In the future, the observed maxima will become the new norm for scenarios greater than the SSP2-4.5 and SSP3-7.0, and will be 2K to 3K warmer even for a low-emission scenario like the SSP1-2.6. An increase in extreme temperatures of more than 10K is conceivable within the 95% confidence interval.

    We have focused on Europe here, but an extension to the rest of the world and to temperature-like variables such as the

heat-index would enable a global map of future heat risks to be drawn up.

### 7.2   Perspectives

Even improvements to the GEV model are possible. For example, the GEV model tends to overestimate return times (see, e.g., Diffenbaugh, 2020; Zeder et al., 2023), and recent work by Noyelle et al. (2025) proposed a new GEV model where the upper bound on temperatures is imposed by physics (Zhang and Boos, 2023; Noyelle et al., 2023). This approach would fit in

naturally with the tools developed here.



Further work is also needed to extend our model to other variables such as wind and precipitation.

It should also be noted that, on the one hand, we have remained in a univariate context, while the estimation of concurrent events increasing impacts appears increasingly necessary; and on the other hand, spatial structures are ignored.

Finally, the analyses produced here provide local information on the worst possible future events, and show the need for
rapid adaptation to extremes warming faster than global warming.

. GISTEMP data are available at data.giss.nasa.gov/gistemp (Lenssen et al., 2019). HadCRUT5 data were obtained from www.metoffice. gov.uk/hadobs/hadcrut5 (Morice et al., 2021; Osborn et al., 2021) on 2025 and are © British Crown Copyright, Met Office 2020, provided under an Open Government License, www.nationalarchives.gov.uk/doc/open-government-licence/version/3/ ERA5 data are available in the Climate Data Store at doi.org/10.24381/cds.adbb2d47 (Hersbach et al., 2020). The CMIP6 model simulations can be downloaded
through the Earth System Grid Federation portals. Instructions to access the data are available here: https://pcmdi.llnl.gov/mips/cmip6/ data-access-getting-started.html.

The current version of ANKIALE is available from the project website: github.com/yrobink/ANKIALE under the GNU-GPL3 licence. The exact version of the model used to produce the results used in this paper is archived on Zenodo under DOI:10.5281/zenodo.15038388 (Robin, 2025), as are input data and scripts to run the model and produce the plots for all the simulations presented in this paper.

. YR had the initial idea of the study, which has been completed and enriched by all co-authors. YR developed the multi-scenarios methods, and OB developed the MCMC, both helped by MV, AR, and PN for the statistical modelling and inferential schemes. YR developed the ANKIALE package and applied it to Europe for the different experiments and wrote the codes for the analyses and to plot the figures. All authors contributed to the methodology and the analyses. YR wrote the first draft of the article with inputs from all the co-authors.

. The authors declare no competing interests.

. This work has benefited from state aid managed by the National Research Agency under France 2030 bearing the references ANR-22-EXTR-0005 (TRACCS-PC4-EXTENDING project), and has been supported by the European Union's Horizon 2020 research and innovation programme under grant agreement No. 101003469 ("XAIDA").

MV has also been supported by the "COMBINE" project funded by the Swiss National Science Foundation (grant n° 200021_200337 /
475 1).





Part of PN's research work was supported by the French national programs: 80 PRIME CNRS-INSU, Agence Nationale de la Recherche (ANR) under reference ANR EXSTA, the PEPR TRACCS programme under grant number ANR-22-EXTR-0005, and the Mines Paris / INRAE chair Geolearning.

We acknowledge the World Climate Research Programme, which, through its Working Group on Coupled Modelling, coordinated and promoted CMIP6. We thank the climate modeling groups for producing and making available their model output, the Earth System Grid Federation (ESGF) for archiving the data and providing access, and the multiple funding agencies who support CMIP6 and ESGF.

Hersbach et al. (ERA5, 2020) was downloaded from the Copernicus Climate Change Service (2025). The results contain modified Copernicus Climate Change Service information 2020. Neither the European Commission nor ECMWF is responsible for any use that may be made of the Copernicus information or data it contains.




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



**Figures**

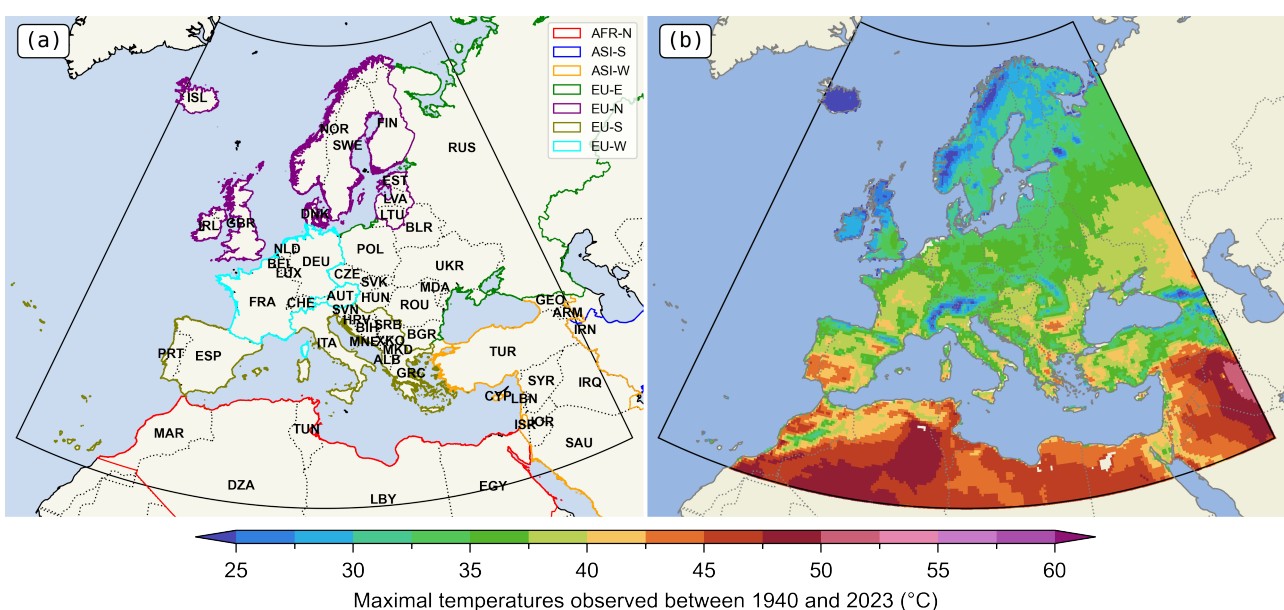

**Figure 1.** The European domain of this study, stretching from 22°W to 45.5°E, and from 26.5°N to 72.5°N, here delimited by the black box. **(a)** The ISO-3166-1 codes of the countries in the domain have been added (see Tab. S1). The colored areas follow the UNSD (2020) M49 norm. **(b)** Maximum temperature observed (ERA5) between 1940 and 2024.





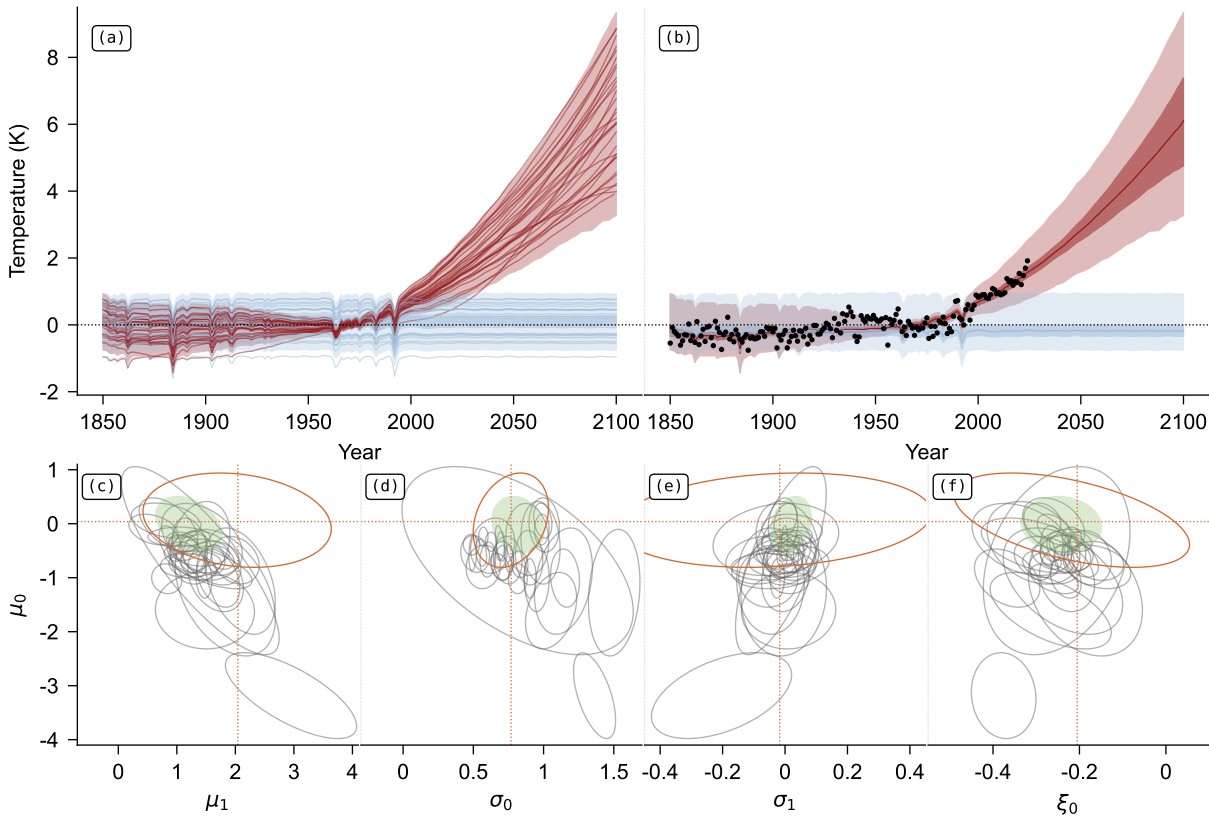

**Figure 2. a.** European covariate $X_t^{\mathrm{R},\dagger}$ in anomaly with respect to 1961/1990 period, in factual (red) and counter-factual (blue) world for the historical and SSP5-8.5 scenarios. The lines (red or blue) are the mean values of each climate model. The filled area (red or blue) is the 95% confidence interval of the multi-model synthesis (see Eq. (4)). **b.** Same as a. without the mean of the climate models, and with the 95% confidence interval of the constraint by observations (the red or blue dark filled area). Black points are the observations. **c.** GEV parameter $\mu_0$ in function of $\mu_1$. Ellipses are the 95% confidence interval defined by the covariance matrix of the parameters. The grey ellipses are the climate models, the black ellipse is the multi-model synthesis, the green filled ellipse is the Bayesian constraint by observations. The orange ellipse comes also from the observations, but by direct estimation with maximum likelihood estimation (the covariate is that given by observational constraint). **d.** Same as c., but for $\mu_0$ in function of $\sigma_0$. **e.** Same as c., but for $\mu_0$ in function of $\sigma_1$. **f.** Same as c., but for $\mu_0$ in function of $\xi_0$.



**Figure 3. a.** Pairwise wasserstein distances ($\mathcal{W}$) between scenarios for probability in factual world $p_t^{\mathrm{F}}$, each year of the historical period (1850/2014) when the scenarios are infered independently. **b.** Pairwise wasserstein distances ($\mathcal{W}$) between scenarios for $p_t^{\mathrm{F}}$, each year of the historical period (1850/2014) when the scenarios are infered simultaneously. **c.** Percent of distance decrease between b. and a. **d./f.** Same as a/c, but for the intensity in the factual world $I_t^{\mathrm{F}}$. **g.** Pairwise wasserstein distances ($\mathcal{W}$) between scenarios for $p_t^{\mathrm{C}}$, each year of all the period (1850/2100) when the scenarios are infered independently. **h.** Pairwise wasserstein distances ($\mathcal{W}$) between scenarios for $p_t^{\mathrm{C}}$, each year of all the period (1850/2100) when the scenarios are infered simultaneously. **i.** Percent of distance decrease between h. and g. **j./l.** Same as g/i, but for the intensity $I_t^{\mathrm{C}}$.





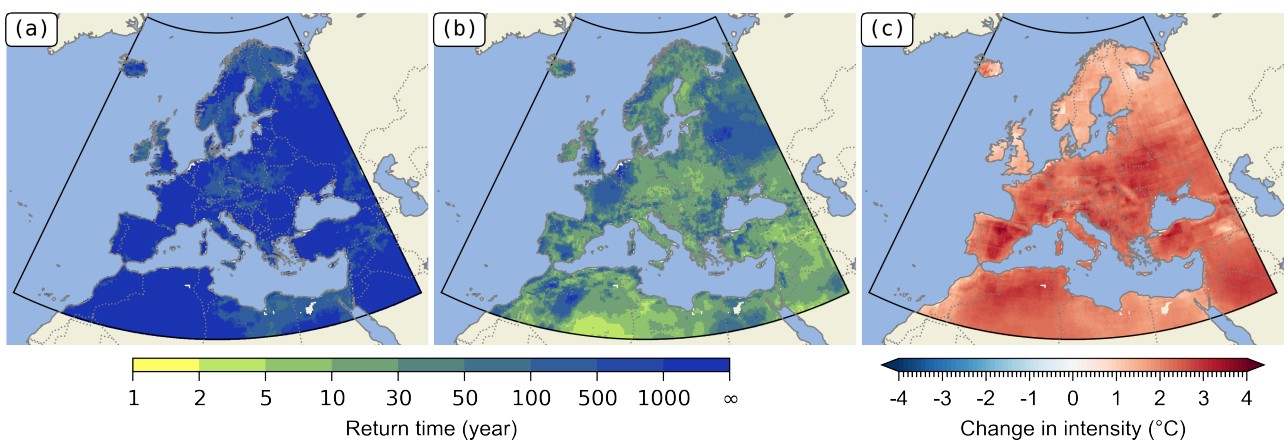

**Figure 4. a.** Return time of the maximum observed between 1940 and 2024 in `TX3x` over Europe, in 2024, without human influence. **b.** Same as a., but for the mean of all scenarios. **c.** Change in intensity in 2024. Lower and upper confidence intervals (95%) are given in figures S6 and S7.





**Figure 5.** Projection of return time (1st and 2nd row) and change in intensity (3rd and 4th row) in 2040 (1st and 3rd row) and 2100 (2nd and 4th row) of the attribution of the maximum event observed in `TX3x` between 1940 and 2023. In columns: in the counter-factual world and for the four scenarios SSP1-2.6, SSP2-4.5, SSP3-7.0 and SSP5-8.5. Lower and upper confidence intervals (95%) are given in figures S8 and S9.



## Tables

| GID | GCM | Hist. | SSP 1-2.6 | 2-4.5 | 3-7.0 | 5-8.5 | References |
|-----|-----|-------|-----------|-------|-------|-------|------------|
| AS-RCEC | TaiESM1 | 1 | 1 | 1 | 1 | 1 | Lee et al. (2020) |
| AWI | AWI-CM-1-1-MR | 5 | 1 | 1 | 5 | 1 | Semmler et al. (2018) |
| BCC | BCC-CSM2-MR | 3 | 1 | 1 | 1 | 1 | Wu et al. (2019) |
| CAMS | CAMS-CSM1-0 | 1 | 1 | 1 | 1 | 1 | Rong et al. (2018) |
| CAS | FGOALS-g3 | 5 | 4 | 4 | 5 | 4 | Pu et al. (2020) |
| CCCma | CanESM5 | 50 | 50 | 50 | 50 | 50 | Swart et al. (2019); Virgin et al. (2021) |
| CMCC | CMCC-ESM2 | 1 | 1 | 1 | 1 | 1 | Lovato et al. (2021) |
| CNRM-CERFACS | CNRM-CM6-1 | 30 | 1 | 1 | 3 | 1 | Voldoire et al. (2019) |
| CNRM-CERFACS | CNRM-CM6-1-HR | 1 | 1 | 0 | 0 | 1 | Voldoire (2019) |
| CNRM-CERFACS | CNRM-ESM2-1 | 10 | 1 | 1 | 3 | 1 | Séférian et al. (2019) |
| CSIRO | ACCESS-ESM1-5 | 40 | 10 | 18 | 10 | 40 | Ziehn et al. (2020) |
| CSIRO-ARCCSS | ACCESS-CM2 | 10 | 3 | 3 | 3 | 5 | Dix et al. (2019) |
| EC-Earth-Consortium | EC-Earth3 | 71 | 2 | 29 | 52 | 58 | (EC-Earth) (2019a) |
| EC-Earth-Consortium | EC-Earth3-CC | 10 | 0 | 1 | 0 | 1 | (EC-Earth) (2020a) |
| EC-Earth-Consortium | EC-Earth3-Veg | 9 | 5 | 6 | 4 | 8 | (EC-Earth) (2019b) |
| EC-Earth-Consortium | EC-Earth3-Veg-LR | 3 | 3 | 3 | 3 | 3 | (EC-Earth) (2020b) |
| INM | INM-CM4-8 | 1 | 1 | 1 | 1 | 1 | Volodin et al. (2018) |
| INM | INM-CM5-0 | 10 | 1 | 1 | 5 | 1 | Volodin et al. (2019) |
| IPSL | IPSL-CM6A-LR | 33 | 6 | 11 | 11 | 7 | Boucher et al. (2020) |
| KIOST | KIOST-ESM | 1 | 1 | 1 | 0 | 1 | Kim et al. (2019) |
| MIROC | MIROC-ES2L | 31 | 3 | 30 | 1 | 1 | Hajima et al. (2019) |
| MIROC | MIROC6 | 50 | 50 | 50 | 3 | 50 | Tatebe and Watanabe (2018) |
| MOHC | HadGEM3-GC31-LL | 5 | 1 | 4 | 0 | 4 | Ridley et al. (2018) |
| MOHC | HadGEM3-GC31-MM | 4 | 1 | 0 | 0 | 4 | Ridley et al. (2019) |
| MOHC | UKESM1-0-LL | 16 | 16 | 16 | 16 | 5 | Good et al. (2019) |
| MPI-M | MPI-ESM1-2-LR | 31 | 10 | 10 | 10 | 30 | Mauritsen et al. (2019); Maher et al. (2019); Mauritsen and Roeckner (2020) |
| MRI | MRI-ESM2-0 | 12 | 1 | 9 | 5 | 5 | Yukimoto et al. (2019) |
| NCC | NorESM2-LM | 3 | 1 | 3 | 3 | 1 | Seland et al. (2020) |
| NCC | NorESM2-MM | 3 | 1 | 2 | 1 | 1 | Seland et al. (2020); Bentsen et al. (2019) |
| NOAA-GFDL | GFDL-ESM4 | 3 | 1 | 1 | 1 | 1 | Krasting et al. (2018) |
| NUIST | NESM3 | 5 | 2 | 2 | 0 | 2 | Cao et al. (2018) |

**Table 1.** List of CMIP6 models used. The value is the number of members for each scenarios.



| Area | Country | Code | Observed | | Natural Only | | SSP1-2.6 | | SSP2-4.5 | | SSP3-7.0 | | SSP5-8.5 | |
|------|---------|------|------|------|------|------|------|------|------|------|------|------|------|------|
| AFR-N | Algeria | DZA | 50 | +2.5 | >1000 | 0 | 10 | +2.9 | 5 | +3.3 | 5 | +3.3 | 2 | +3.7 |
| | Egypt | EGY | 30 | +1.7 | >1000 | 0 | 30 | +2.1 | 10 | +2.4 | 10 | +2.4 | 10 | +2.6 |
| | Libya | LBY | 10 | +1.9 | >1000 | 0 | 10 | +2.3 | 10 | +2.6 | 10 | +2.6 | 5 | +2.9 |
| | Morocco | MAR | 100 | +2 | >1000 | 0 | 30 | +2.4 | 10 | +2.7 | 10 | +2.7 | 10 | +3 |
| | Tunisia | TUN | 50 | +2.4 | >1000 | 0 | 30 | +2.8 | 10 | +3.2 | 10 | +3.2 | 10 | +3.5 |
| ASI-S | Iran | IRN | 10 | +2 | >1000 | 0 | 5 | +2.4 | 5 | +2.8 | 5 | +2.8 | 2 | +3.2 |
| ASI-W | Armenia | ARM | 10 | +2.3 | >1000 | 0 | 5 | +2.7 | 5 | +3.1 | 5 | +3.1 | 2 | +3.4 |
| | Cyprus | CYP | 30 | +1.7 | >1000 | 0 | 10 | +2.1 | 10 | +2.3 | 10 | +2.3 | 10 | +2.6 |
| | Georgia | GEO | 30 | +2.4 | >1000 | 0 | 10 | +2.8 | 10 | +3.2 | 10 | +3.2 | 5 | +3.6 |
| | Iraq | IRQ | 10 | +2.4 | >1000 | 0 | 5 | +2.9 | 2 | +3.4 | 2 | +3.3 | 2 | +3.7 |
| | Israel | ISR | 30 | +2 | >1000 | 0 | 10 | +2.4 | 10 | +2.7 | 10 | +2.7 | 10 | +2.9 |
| | Jordan | JOR | 30 | +2.4 | >1000 | 0 | 10 | +2.9 | 10 | +3.3 | 10 | +3.3 | 5 | +3.6 |
| | Lebanon | LBN | 30 | +2 | 1000 | 0 | 10 | +2.4 | 10 | +2.7 | 10 | +2.7 | 10 | +2.9 |
| | Saudi Arabia | SAU | 10 | +2.6 | >1000 | 0 | 5 | +3.2 | 2 | +3.6 | 2 | +3.6 | 2 | +4 |
| | Syria | SYR | 10 | +2.3 | >1000 | 0 | 10 | +2.8 | 5 | +3.1 | 5 | +3.1 | 5 | +3.4 |
| | Türkiye | TUR | 30 | +2.3 | >1000 | 0 | 10 | +2.8 | 10 | +3.1 | 10 | +3.1 | 5 | +3.5 |
| EU-E | Belarus | BLR | 100 | +2.6 | >1000 | 0 | 50 | +3.2 | 30 | +3.5 | 30 | +3.6 | 30 | +3.9 |
| | Bulgaria | BGR | 50 | +2.5 | >1000 | 0 | 30 | +2.9 | 10 | +3.3 | 10 | +3.3 | 10 | +3.6 |
| | Czechia | CZE | 10 | +2.4 | 1000 | 0 | 10 | +2.9 | 5 | +3.3 | 5 | +3.3 | 5 | +3.6 |
| | Hungary | HUN | 30 | +2.4 | >1000 | 0 | 10 | +2.9 | 10 | +3.2 | 10 | +3.2 | 5 | +3.6 |
| | Moldova | MDA | 50 | +2.7 | >1000 | 0 | 30 | +3.3 | 10 | +3.7 | 10 | +3.7 | 10 | +4 |
| | Poland | POL | 30 | +2.4 | >1000 | 0 | 10 | +2.9 | 10 | +3.3 | 10 | +3.3 | 10 | +3.6 |
| | Romania | ROU | 50 | +2.5 | >1000 | 0 | 30 | +3 | 10 | +3.4 | 10 | +3.4 | 10 | +3.7 |
| | Russian Federation | RUS | 100 | +1.9 | >1000 | 0 | 50 | +2.3 | 50 | +2.6 | 50 | +2.6 | 30 | +2.9 |
| | Slovakia | SVK | 10 | +2.6 | >1000 | 0 | 10 | +3.2 | 5 | +3.6 | 5 | +3.6 | 5 | +3.9 |
| | Ukraine | UKR | 30 | +2.6 | >1000 | 0 | 10 | +3.1 | 10 | +3.5 | 10 | +3.5 | 10 | +3.9 |
| EU-N | Denmark | DNK | 50 | +1.5 | >1000 | 0 | 30 | +1.8 | 30 | +2 | 30 | +2.1 | 10 | +2.3 |
| | Estonia | EST | 30 | +2 | >1000 | 0 | 10 | +2.4 | 10 | +2.7 | 10 | +2.7 | 10 | +3 |
| | Finland | FIN | 30 | +1.5 | >1000 | 0 | 30 | +1.8 | 10 | +2.1 | 10 | +2.1 | 10 | +2.3 |
| | Iceland | ISL | 100 | +1.6 | >1000 | 0 | 50 | +1.9 | 50 | +2.1 | 50 | +2.2 | 30 | +2.4 |
| | Ireland | IRL | 30 | +1.2 | 500 | 0 | 30 | +1.5 | 30 | +1.6 | 30 | +1.7 | 30 | +1.8 |
| | Latvia | LVA | 30 | +2 | >1000 | 0 | 10 | +2.5 | 10 | +2.8 | 10 | +2.8 | 10 | +3.1 |
| | Lithuania | LTU | 30 | +2.1 | >1000 | 0 | 10 | +2.6 | 10 | +2.9 | 10 | +2.9 | 10 | +3.2 |
| | Norway | NOR | 50 | +1.4 | >1000 | 0 | 50 | +1.7 | 30 | +2 | 30 | +2 | 30 | +2.2 |
| | Sweden | SWE | 50 | +1.5 | >1000 | 0 | 30 | +1.8 | 30 | +2 | 30 | +2 | 10 | +2.3 |
| | United Kingdom | GBR | 500 | +1.3 | >1000 | 0 | 100 | +1.5 | 100 | +1.7 | 100 | +1.8 | 100 | +1.9 |
| EU-S | Albania | ALB | 10 | +2.3 | >1000 | 0 | 10 | +2.8 | 5 | +3.1 | 5 | +3.1 | 5 | +3.5 |
| | Bosnia and Herzegovina | BIH | 10 | +2.7 | >1000 | 0 | 10 | +3.3 | 5 | +3.7 | 5 | +3.8 | 5 | +4.1 |
| | Croatia | HRV | 30 | +2.6 | >1000 | 0 | 10 | +3.1 | 5 | +3.5 | 5 | +3.5 | 5 | +3.8 |





| | Country | Code | | | | | | | | | | | | |
|---|---|---|---|---|---|---|---|---|---|---|---|---|---|---|
| | Greece | GRC | 30 | +2.2 | > 1000 | 0 | 30 | +2.7 | 10 | +3 | 10 | +3 | 10 | +3.3 |
| | Italy | ITA | 30 | +2.3 | > 1000 | 0 | 30 | +2.7 | 10 | +3.1 | 10 | +3.1 | 10 | +3.4 |
| | Kosovo | XKO | 50 | +2.7 | > 1000 | 0 | 30 | +3.3 | 10 | +3.7 | 10 | +3.8 | 10 | +4.1 |
| | Macedonia | MKD | 50 | +2.6 | > 1000 | 0 | 30 | +3.1 | 10 | +3.5 | 10 | +3.5 | 10 | +3.9 |
| | Montenegro | MNE | 30 | +2.8 | > 1000 | 0 | 10 | +3.4 | 10 | +3.9 | 10 | +3.9 | 5 | +4.3 |
| | Portugal | PRT | 100 | +1.9 | > 1000 | 0 | 50 | +2.3 | 50 | +2.6 | 50 | +2.6 | 30 | +2.8 |
| | Serbia | SRB | 30 | +2.5 | > 1000 | 0 | 10 | +3 | 10 | +3.4 | 10 | +3.5 | 10 | +3.8 |
| | Slovenia | SVN | 30 | +2.7 | > 1000 | 0 | 10 | +3.3 | 10 | +3.7 | 10 | +3.7 | 10 | +4 |
| | Spain | ESP | 50 | +2.5 | > 1000 | 0 | 30 | +3 | 10 | +3.4 | 10 | +3.4 | 10 | +3.7 |
| EU-W | Austria | AUT | 30 | +2.4 | > 1000 | 0 | 10 | +2.8 | 10 | +3.2 | 10 | +3.2 | 5 | +3.5 |
| | Belgium | BEL | 100 | +2.3 | > 1000 | 0 | 100 | +2.7 | 100 | +3.1 | 100 | +3.1 | 50 | +3.4 |
| | France | FRA | 100 | +2.4 | > 1000 | 0 | 50 | +2.8 | 30 | +3.2 | 30 | +3.2 | 30 | +3.5 |
| | Germany | DEU | 50 | +2.2 | > 1000 | 0 | 30 | +2.6 | 30 | +2.9 | 30 | +3 | 10 | +3.2 |
| | Luxembourg | LUX | 100 | +2.4 | > 1000 | 0 | 50 | +2.9 | 30 | +3.2 | 30 | +3.2 | 30 | +3.5 |
| | Netherlands | NLD | 500 | +2.3 | > 1000 | 0 | 100 | +2.7 | 100 | +3.1 | 100 | +3.1 | 100 | +3.4 |
| | Switzerland | CHE | 50 | +2.4 | > 1000 | 0 | 30 | +2.9 | 10 | +3.2 | 10 | +3.2 | 10 | +3.5 |

**Table 2.** Average values for return periods and change in intensity in 2040 in `TX3x` over the Europe (See Fig. 5). In columns: values are given for a world without human influence (Natural Only) and 4 CMIP6 scenarios. Observed values correspond to the year 2024. For each column, the first value is the return period, rounded to the nearest 1, 2 , 5 , 10, 30, 50, 100, 500 and 1000 years. The second is the difference with the world without human influence (i.e. this is an estimator of local climate change extremes at the end of the century). The 95% confidence interval is given in tables S2 and S3.




| Area | Country | Code | Observed | | Natural Only | | SSP1-2.6 | | SSP2-4.5 | | SSP3-7.0 | | SSP5-8.5 | |
|------|---------|------|------|------|------|------|------|------|------|------|------|------|------|------|
| AFR-N | Algeria | DZA | 50 | +2.5 | > 1000 | 0 | 2 | +3.4 | 1 | +5.4 | 1 | +7.3 | 1 | +8.9 |
| | Egypt | EGY | 30 | +1.7 | > 1000 | 0 | 10 | +2.4 | 5 | +3.9 | 1 | +5.3 | 1 | +6.8 |
| | Libya | LBY | 10 | +1.9 | > 1000 | 0 | 5 | +2.7 | 2 | +4.3 | 1 | +5.9 | 1 | +7.5 |
| | Morocco | MAR | 100 | +2 | > 1000 | 0 | 10 | +2.8 | 5 | +4.4 | 2 | +5.9 | 2 | +7.1 |
| | Tunisia | TUN | 50 | +2.4 | > 1000 | 0 | 10 | +3.3 | 2 | +5.1 | 1 | +6.6 | 1 | +8.2 |
| ASI-S | Iran | IRN | 10 | +2 | > 1000 | 0 | 2 | +2.9 | 1 | +5 | 1 | +7.2 | 1 | +9.8 |
| ASI-W | Armenia | ARM | 10 | +2.3 | > 1000 | 0 | 5 | +3.2 | 1 | +5.2 | 1 | +7.2 | 1 | +9.3 |
| | Cyprus | CYP | 30 | +1.7 | > 1000 | 0 | 10 | +2.4 | 2 | +3.8 | 1 | +5.2 | 1 | +6.5 |
| | Georgia | GEO | 30 | +2.4 | > 1000 | 0 | 10 | +3.3 | 2 | +5.4 | 1 | +7.5 | 1 | +9.7 |
| | Iraq | IRQ | 10 | +2.4 | > 1000 | 0 | 2 | +3.4 | 1 | +5.6 | 1 | +7.8 | 1 | +10.1 |
| | Israel | ISR | 30 | +2 | > 1000 | 0 | 10 | +2.7 | 2 | +4.2 | 1 | +5.5 | 1 | +6.6 |
| | Jordan | JOR | 30 | +2.4 | > 1000 | 0 | 10 | +3.4 | 1 | +5.3 | 1 | +7.1 | 1 | +8.9 |
| | Lebanon | LBN | 30 | +2 | 500 | 0 | 10 | +2.8 | 2 | +4.3 | 1 | +5.6 | 1 | +6.9 |
| | Saudi Arabia | SAU | 10 | +2.6 | > 1000 | 0 | 2 | +3.7 | 1 | +6.2 | 1 | +8.9 | 1 | +11 |
| | Syria | SYR | 10 | +2.3 | > 1000 | 0 | 5 | +3.2 | 1 | +5 | 1 | +6.7 | 1 | +8.3 |
| | Türkiye | TUR | 30 | +2.3 | > 1000 | 0 | 10 | +3.3 | 2 | +5.2 | 1 | +7.2 | 1 | +9.2 |
| EU-E | Belarus | BLR | 100 | +2.6 | > 1000 | 0 | 30 | +3.7 | 5 | +5.6 | 2 | +7.5 | 1 | +9.2 |
| | Bulgaria | BGR | 50 | +2.5 | > 1000 | 0 | 10 | +3.4 | 2 | +5.3 | 1 | +7.1 | 1 | +8.7 |
| | Czechia | CZE | 10 | +2.4 | 1000 | 0 | 5 | +3.4 | 2 | +5.3 | 1 | +7.2 | 1 | +9 |
| | Hungary | HUN | 30 | +2.4 | > 1000 | 0 | 10 | +3.4 | 2 | +5.3 | 1 | +7.4 | 1 | +9.4 |
| | Moldova | MDA | 50 | +2.7 | > 1000 | 0 | 10 | +3.8 | 2 | +5.8 | 1 | +7.6 | 1 | +9.3 |
| | Poland | POL | 30 | +2.4 | > 1000 | 0 | 10 | +3.4 | 2 | +5.1 | 1 | +6.7 | 1 | +8.1 |
| | Romania | ROU | 50 | +2.5 | > 1000 | 0 | 10 | +3.6 | 2 | +5.5 | 1 | +7.4 | 1 | +9.2 |
| | Russian Federation | RUS | 100 | +1.9 | > 1000 | 0 | 50 | +2.7 | 10 | +4.3 | 2 | +5.9 | 1 | +7.6 |
| | Slovakia | SVK | 10 | +2.6 | > 1000 | 0 | 5 | +3.7 | 1 | +5.8 | 1 | +8 | 1 | +10.1 |
| | Ukraine | UKR | 30 | +2.6 | > 1000 | 0 | 10 | +3.7 | 2 | +5.6 | 1 | +7.6 | 1 | +9.5 |
| EU-N | Denmark | DNK | 50 | +1.5 | > 1000 | 0 | 30 | +2.1 | 5 | +3.4 | 2 | +4.7 | 1 | +6 |
| | Estonia | EST | 30 | +2 | > 1000 | 0 | 10 | +2.8 | 2 | +4.3 | 1 | +5.9 | 1 | +7.4 |
| | Finland | FIN | 30 | +1.5 | > 1000 | 0 | 10 | +2.1 | 5 | +3.4 | 2 | +4.8 | 1 | +6.1 |
| | Iceland | ISL | 100 | +1.6 | > 1000 | 0 | 30 | +2.2 | 10 | +3.6 | 5 | +5.1 | 1 | +6.6 |
| | Ireland | IRL | 30 | +1.2 | 500 | 0 | 30 | +1.7 | 10 | +2.7 | 5 | +3.7 | 2 | +4.7 |
| | Latvia | LVA | 30 | +2 | > 1000 | 0 | 10 | +2.9 | 2 | +4.5 | 1 | +6.3 | 1 | +7.9 |
| | Lithuania | LTU | 30 | +2.1 | > 1000 | 0 | 10 | +3 | 2 | +4.6 | 1 | +6.2 | 1 | +7.7 |
| | Norway | NOR | 50 | +1.4 | > 1000 | 0 | 30 | +2 | 10 | +3.3 | 5 | +4.7 | 2 | +6.1 |
| | Sweden | SWE | 50 | +1.5 | > 1000 | 0 | 30 | +2.1 | 5 | +3.4 | 2 | +4.8 | 1 | +6.2 |
| | United Kingdom | GBR | 500 | +1.3 | > 1000 | 0 | 100 | +1.8 | 30 | +2.9 | 10 | +4.1 | 5 | +5.3 |
| EU-S | Albania | ALB | 10 | +2.3 | > 1000 | 0 | 5 | +3.3 | 1 | +5.2 | 1 | +7.3 | 1 | +9.3 |
| | Bosnia and Herzegovina | BIH | 10 | +2.7 | > 1000 | 0 | 5 | +3.9 | 1 | +6.2 | 1 | +8.5 | 1 | +10.7 |
| | Croatia | HRV | 30 | +2.6 | > 1000 | 0 | 5 | +3.7 | 1 | +5.7 | 1 | +7.8 | 1 | +9.7 |





| | Country | | | | | | | | | | | | | | | | |
|---|---|---|---|---|---|---|---|---|---|---|---|---|---|---|---|---|
| | Greece | GRC | 30 | +2.2 | > 1000 | 0 | 10 | +3.1 | 2 | +4.9 | 1 | +6.6 | 1 | +8.3 |
| | Italy | ITA | 30 | +2.3 | > 1000 | 0 | 10 | +3.2 | 2 | +5 | 1 | +6.8 | 1 | +8.5 |
| | Kosovo | XKO | 50 | +2.7 | > 1000 | 0 | 10 | +3.9 | 2 | +6.4 | 1 | +9.1 | 1 | +11.9 |
| | Macedonia | MKD | 50 | +2.6 | > 1000 | 0 | 10 | +3.7 | 2 | +5.9 | 1 | +8.3 | 1 | +10.7 |
| | Montenegro | MNE | 30 | +2.8 | > 1000 | 0 | 5 | +4.1 | 1 | +6.5 | 1 | +9.3 | 1 | +12 |
| | Portugal | PRT | 100 | +1.9 | > 1000 | 0 | 30 | +2.7 | 10 | +4.2 | 2 | +5.7 | 2 | +7.2 |
| | Serbia | SRB | 30 | +2.5 | > 1000 | 0 | 10 | +3.6 | 2 | +5.7 | 1 | +7.8 | 1 | +9.9 |
| | Slovenia | SVN | 30 | +2.7 | > 1000 | 0 | 10 | +3.8 | 2 | +5.7 | 1 | +7.5 | 1 | +9.1 |
| | Spain | ESP | 50 | +2.5 | > 1000 | 0 | 10 | +3.5 | 2 | +5.5 | 1 | +7.3 | 1 | +9.1 |
| EU-W | Austria | AUT | 30 | +2.4 | > 1000 | 0 | 10 | +3.3 | 2 | +5.2 | 1 | +7 | 1 | +8.8 |
| | Belgium | BEL | 100 | +2.3 | > 1000 | 0 | 50 | +3.2 | 10 | +4.9 | 2 | +6.5 | 1 | +8.1 |
| | France | FRA | 100 | +2.4 | > 1000 | 0 | 30 | +3.3 | 5 | +5.2 | 2 | +7 | 1 | +8.8 |
| | Germany | DEU | 50 | +2.2 | > 1000 | 0 | 10 | +3.1 | 5 | +4.8 | 1 | +6.5 | 1 | +8.2 |
| | Luxembourg | LUX | 100 | +2.4 | > 1000 | 0 | 30 | +3.3 | 5 | +5 | 2 | +6.5 | 1 | +7.9 |
| | Netherlands | NLD | 500 | +2.3 | > 1000 | 0 | 100 | +3.2 | 10 | +4.9 | 5 | +6.6 | 2 | +8.2 |
| | Switzerland | CHE | 50 | +2.4 | > 1000 | 0 | 10 | +3.4 | 2 | +5.1 | 1 | +6.9 | 1 | +8.5 |

**Table 3.** Average values for return periods and change in intensity in 2100 in `TX3x` over the Europe (See Fig. 5). In columns: values are given for a world without human influence (Natural Only) and 4 CMIP6 scenarios. Observed values correspond to the year 2024. For each column, the first value is the return period, rounded to the nearest 1, 2 , 5 , 10, 30, 50, 100, 500 and 1000 years. The second is the difference with the world without human influence (i.e. this is an estimator of local climate change extremes at the end of the century). The 95% confidence interval is given in tables S4 and S5.



## Algorithms

---

**Algorithm 1** Backfitting algorithm for estimating the parameters of an additive model (GAM).

---

**Require:** The $X_t^*$ series to be decomposed,

**Require:** The $X_t^{*,N}$ series of responses to natural forcings,

**Require:** The number of degree of freedom $d = 1 + 1 + dof$ (constant, natural, splines).

**Require:** Two constants $\Delta \gg \delta$ (typically $10^9$ and $0.001$).

  Set $\theta_0 = (\Delta, \Delta, \Delta, \ldots, \Delta)$

  Set $\theta_1 = (0, 0, 0, \ldots, 0)$

  Set $\mathcal{R}_t^* = X_t^*$

  **while** $\|\theta_1 - \theta_0\| > \delta$ **do**

    Calculate $\mathcal{B}_t^*$, regression with $B$-splines at $dof$ degrees of freedom (without constant) of $\mathcal{R}_t^*$.

    This gives coefficients $(s_0^{*,A}, \ldots, s_{dof}^{*,A})$.

    Calculate $\mathcal{R}_t^* = X_t^* - \mathcal{B}_t^*$.

    Calculate $\mathcal{L}_t^*$, linear regression of $\mathcal{R}_t^*$ against $X_t^{*,N}$. This gives the coefficients $(X^{*,0}, \alpha^{*,N})$.

    Calculate $\mathcal{R}_t^* = X_t^* - \mathcal{L}_t^*$.

    Set $\theta_0 = \theta_1$.

    Update $\theta_1 = (X^{*,0}, \alpha^{*,N}, s_0^{*,A}, \ldots, s_{dof}^{*,A})$.

  **end while**

  Calculate $H$ the projection operator,

  Calculate $\sigma$, the standard deviation of the residuals,

  Set $\mu_\theta = \theta_1$,

  Set $\Sigma_\theta = (H^T \cdot H)^{-1} \sigma^2 / d$.

  **return** The parameters $\mu_\theta$ and $\Sigma_\theta$ given by the equations (**??**) and (**??**).

---





---

**Algorithm 2** Backfitting algorithm for estimating the parameters of several additive models simultaneously (GAM).

---

**Require:** The $X_t^*$ series to be decomposed, for each scenario,

**Require:** The $X_t^{*,N}$ series of responses to natural forcings,

**Require:** The number of degree of freedom $d = 1 + 1 + dof$ (constant, natural, splines).

**Require:** Two constants $\Delta \gg \delta$ (typically $10^9$ and 0.001).

   Set $\theta_0 = (\Delta, \Delta, \Delta, \ldots, \Delta)$

   Set $\theta_1 = (0, 0, 0, \ldots, 0)$

   Set $\mathcal{R}_t^* = X_t^*$ for each scenarios

   **while** $\|\theta_1 - \theta_0\| > \delta$ **do**

      **for** Each scenario SSP **do**

         Calculate $\mathcal{B}_t^*$, regression with $B$-splines at $dof$ degrees of freedom (without constant) of $\mathcal{R}_t^*$.

         This gives coefficients $(s_0^{*,A,\text{SSP}}, \ldots, s_{dof}^{*,A,\text{SSP}})$ (one per scenario).

         Calculate $\mathcal{R}_t^* = X_t^* - \mathcal{B}_t^*$.

      **end for**

      Calculate $\mathcal{L}_t^*$, linear regression of the mean (of the scenarios) of $\mathcal{R}_t^*$ against $X_t^{*,N}$.

      This gives the coefficients $\left(X^{*,0}, \alpha^{*,N}\right)$.

      Calculate $\mathcal{R}_t^* = X_t^* - \mathcal{L}_t^*$, for each scenario.

      Set $\theta_0 = \theta_1$.

      Update $\theta_1 = (X^{*,0}, \alpha^{*,N}) \oplus \bigoplus_{\text{SSP}} (s_0^{*,A,\text{SSP}}, \ldots, s_{dof}^{*,A,\text{SSP}})$.

   **end while**

   Calculate $H$ the projection operator,

   Calculate $\sigma$, the standard deviation of the residuals,

   Set $\mu_\theta = \theta_1$,

   Set $\Sigma_\theta = (H^T \cdot H)^{-1} \sigma^2 / d$.

   **return** The parameters $\mu_\theta$ and $\Sigma_\theta$ given by the equations (**??**) and (**??**).

---





**Appendix A: Detailed mathematics of the methodology**

**A1 Decomposition of the covariables**

We will then decompose the temperatures (global or regional) so as to isolate forcings of natural origin on the one hand, and anthropogenic forcings on the other hand. Recall that we have the following equation, already given in Eq. 1, with the difference that the linear coefficient of $X^{*,\mathrm{N}}$ is explained:

$$X_t^* = X^{*,0} + \alpha^{*,\mathrm{N}} X_t^{*,\mathrm{N}} + X_t^{*,\mathrm{A}} + \varepsilon.$$

Where:

- $X^{*,0}$: a constant,

- $X_t^{*,\mathrm{N}}$: either the response of an Energy Balance Model (Energy Balance Model, Held et al., 2010; Geoffroy et al., 2013) for CMIP5, or radiative forcings for CMIP6 (Smith, 2020); modeling natural forcings,

- $X_t^{*,\mathrm{A}}$: a smoothing spline, modelling the anthropogenic forcing (see below),

- $\varepsilon$: a white noise Gaussian error term describing natural variability.

This equation is a Generalized Additive Model (GAM, see e.g., Hastie, 2017), with a parameter to be estimated for the constant (the constant itself in fact), a parameter $\alpha^{*,\mathrm{N}}$ for the natural term (the linear regression term). The term $X_t^{*,\mathrm{A}}$ is obtained from a smoothing spline.

Let $dof$ denote the number of degrees of freedom of the (unbiased) spline basis, and $N_T$ the number of time steps (here 780 $N_T = 251$ years). The term $X^{*,\mathrm{A}}$ can then be reformulated as a matrix product between a matrix $B^S$ ($B$ for *Basis*) of size $N_T \times dof$, and a vector of coefficients of size $dof$:

$$X_t^{*,\mathrm{A}} = B^S \cdot (s_0, \ldots, s_{dof-1})^T = \begin{pmatrix} B_0^{S,0} & \cdots & B_0^{S,dof-1} \\ B_1^{S,0} & \cdots & B_1^{S,dof-1} \\ \vdots & \ddots & \vdots \\ B_{N_T-1}^{S,0} & \cdots & B_{N_T-1}^{S,dof-1} \end{pmatrix} \cdot (s_0, \ldots, s_{dof-1})^T.$$

Following Robin and Ribes (2020a), we assume $dof = 6$ for the smoothing spline of the anthropogenic part. Temperatures (global or regional) can therefore be described by an 8-parameter $\theta$ vector of the following form (the $\oplus$ symbol designates the direct sum of vectors or matrices, i.e. concatenation along dimensions):

$$\theta := \left( X^{*,0}, \alpha^{*,\mathrm{N}} \right) \oplus \left( s_0^{*,A}, \ldots, s_5^{*,A} \right) = \left( X^{*,0}, \alpha^{*,\mathrm{N}}, s_0^{*,A}, \ldots, s_5^{*,A} \right).$$





In a similar way to $B^S$, we will note $B^N$ the design matrix of the natural part, which allows us to write:

$$
X_t^* = \left[\underbrace{\begin{pmatrix} 1 & X_0^{*,\mathrm{N}} \\ 1 & X_1^{*,\mathrm{N}} \\ \vdots & \vdots \\ 1 & X_{N_T-1}^{*,\mathrm{N}} \end{pmatrix}}_{B^N} \oplus \underbrace{\begin{pmatrix} B_0^{S,0} & \cdots & B_0^{S,5} \\ B_1^{S,0} & \cdots & B_1^{S,5} \\ \vdots & \ddots & \vdots \\ B_{N_T-1}^{S,0} & \cdots & B_{N_T-1}^{S,5} \end{pmatrix}}_{B^S}\right] \cdot \theta^T + \varepsilon = \begin{pmatrix} 1 & X_0^{*,\mathrm{N}} & B_0^{S,0} & \cdots & B_0^{S,5} \\ 1 & X_1^{*,\mathrm{N}} & B_1^{S,0} & \cdots & B_1^{S,5} \\ \vdots & \vdots & \vdots & \ddots & \vdots \\ 1 & X_{N_T-1}^{*,\mathrm{N}} & B_{N_T-1}^{S,0} & \cdots & B_{N_T-1}^{S,5} \end{pmatrix} \cdot \theta^T + \varepsilon. \quad \text{(A1)}
$$

From this equation we can then construct the detailed version of Eq. (2):

$$
\begin{aligned}
\theta &= \quad \theta^{\mathrm{G}} \\
&\oplus \quad \theta^{\mathrm{R}}. \\
&= \left(X^{\mathrm{G},0}, \alpha^{\mathrm{G,N}}\right) \quad \oplus \quad \left(s_0^{\mathrm{G,A,SSP1\text{-}2.6}}, \ldots, s_5^{\mathrm{G,A,SSP1\text{-}2.6}}\right) \\
&\qquad\qquad\qquad\quad \oplus \quad \left(s_0^{\mathrm{G,A,SSP2\text{-}4.5}}, \ldots, s_5^{\mathrm{G,A,SSP2\text{-}4.5}}\right) \\
&\qquad\qquad\qquad\quad \oplus \quad \left(s_0^{\mathrm{G,A,SSP3\text{-}7.0}}, \ldots, s_5^{\mathrm{G,A,SSP3\text{-}7.0}}\right) \\
&\qquad\qquad\qquad\quad \oplus \quad \left(s_0^{\mathrm{G,A,SSP5\text{-}8.5}}, \ldots, s_5^{\mathrm{G,A,SSP5\text{-}8.5}}\right) \\
&\oplus \left(X^{\mathrm{R},0}, \alpha^{\mathrm{R,N}}\right) \quad \oplus \quad \left(s_0^{\mathrm{R,A,SSP1\text{-}2.6}}, \ldots, s_5^{\mathrm{R,A,SSP1\text{-}2.6}}\right) \\
&\qquad\qquad\qquad\quad \oplus \quad \left(s_0^{\mathrm{R,A,SSP2\text{-}4.5}}, \ldots, s_5^{\mathrm{R,A,SSP2\text{-}4.5}}\right) \\
&\qquad\qquad\qquad\quad \oplus \quad \left(s_0^{\mathrm{R,A,SSP3\text{-}7.0}}, \ldots, s_5^{\mathrm{R,A,SSP3\text{-}7.0}}\right) \\
&\qquad\qquad\qquad\quad \oplus \quad \left(s_0^{\mathrm{R,A,SSP5\text{-}8.5}}, \ldots, s_5^{\mathrm{R,A,SSP5\text{-}8.5}}\right) \\
&= \left(X^{\mathrm{G},0}, \alpha^{\mathrm{G,N}}\right) \quad \bigoplus_{\mathrm{SSP}} \quad \left(s_0^{\mathrm{G,A,SSP}}, \ldots, s_5^{\mathrm{G,A,SSP}}\right) \\
&\oplus \left(X^{\mathrm{R},0}, \alpha^{\mathrm{R,N}}\right) \quad \bigoplus_{\mathrm{SSP}} \quad \left(s_0^{\mathrm{R,A,SSP}}, \ldots, s_5^{\mathrm{R,A,SSP}}\right)
\end{aligned} \quad \text{(A2)}
$$

We therefore add the GEV parameters to the statistical model in Eq. (A2):

$$
\begin{aligned}
\theta &= \theta^{\mathrm{G}} \quad \oplus \quad \theta^{\mathrm{R}} \quad \oplus \quad (\mu_0, \mu_1, \sigma_0, \sigma_1, \xi_0) \\
&= \theta^{\mathrm{G}} \quad \oplus \quad \theta^{\mathrm{R}} \quad \oplus \quad \theta^{\mathrm{GEV}}
\end{aligned}
$$

## A2  Simultaneous constraint of external forcings: construction of the $A^1$ matrix

Recall the Eq. (6):

$$
X_t^{\mathrm{GR,o}} := \begin{pmatrix} X_t^{\mathrm{G,o}} \\ X_t^{\mathrm{R,o}} \end{pmatrix} = A \cdot \kappa + \varepsilon^o \quad \text{(A3)}
$$





The $A$ matrix is the following form:

$$A := \begin{pmatrix} R^{\mathrm{G,o}} & R^{\mathrm{R,o}} \end{pmatrix} \cdot \begin{pmatrix} A^1 & A^0 & A^{\mathrm{GEV}} \\ A^0 & A^1 & A^{\mathrm{GEV}} \end{pmatrix}$$

where:

– $A^1$ is a matrix which transforms $\kappa^{\mathrm{G}}$ or $\kappa^{\mathrm{R}}$ into the mean covariate, taken along the SSPs. In other words for $\kappa^{\mathrm{G}}$:

$$A^1 \cdot \kappa^{\mathrm{G}} = \frac{1}{N^{\mathrm{SSP}}} \sum_{\mathrm{SSP}} X_t^{\mathrm{G,F,SSP}}$$

– $A^0$ is the same dimension as $A^1$, but with null values.

– $A^{\mathrm{GEV}}$ is a null matrix such that $A^{\mathrm{GEV}} \cdot \kappa^{\mathrm{GEV}} = 0$.

– $R^{\mathrm{G,o}}$ are $R^{\mathrm{R,o}}$ are matrices which restrict the time axis to that of the observations.

We propose here to detail the construction of $A^1$. Let's start with Eq. (A1) and (A2), we can write the following relation for $\kappa^{\mathrm{G}}$:

$$\left[ B^N \oplus \bigoplus_{\mathrm{SSP}} B^S \right] \cdot \kappa^{\mathrm{G}} = X^{\mathrm{G,0}} + X_t^{\mathrm{G,N}} + \sum_{\mathrm{SSP}} X_t^{\mathrm{G,A,SSP}}$$

By multiplying the matrix $B^N$ by the number of SSP scenarios $N^{\mathrm{SSP}}$, we have:

$$\frac{1}{N^{\mathrm{SSP}}} \left[ (N^{\mathrm{SSP}} B^N) \oplus \bigoplus_{\mathrm{SSP}} B^S \right] \cdot \kappa^{\mathrm{G}} = \frac{1}{N^{\mathrm{SSP}}} \sum_{\mathrm{SSP}} \left[ X^{\mathrm{G,0}} + X_t^{\mathrm{G,N}} + X_t^{\mathrm{G,A,SSP}} \right] = \frac{1}{N^{\mathrm{SSP}}} \sum_{\mathrm{SSP}} X_t^{\mathrm{G,F,SSP}}$$

The matrix $A^1$ is then given by:

$$A^1 := \frac{1}{N^{\mathrm{SSP}}} \left[ (N^{\mathrm{SSP}} B^N) \oplus \bigoplus_{\mathrm{SSP}} B^S \right] \tag{A4}$$