# Peer review of "A Bayesian statistical method to estimate the climatology of extreme temperature under multiple scenarios: the ANKIALE package"

_EGUsphere, 2025_

## Referee Comment (RC1)

Review, Robin et al., GMD

I think this is an important piece of work – it extends the use of observational constraints to the estimation of the characteristics of temperature extremes for unobserved past and future periods under both "factual" and "counterfactual" conditions, using a rigorous Bayesian framework in which extreme temperature distribution is described with a suitable extreme value distribution. The ANKIALE package that implements the method should help to make this sophisticated methodology relatively accessible to a broad range of users.

Unfortunately, however, the paper would be VERY challenging for the target audience to understand. For this work to be impactful, I think it will be necessary for the authors to think much more carefully about presentation issues, providing more complete and more accessible explanations for the choices that they have made in implementing the package. I think they also need to provide substantially more insight into choices users will have to make when applying the package, with an emphasis on physical considerations as well as statistical and pragmatic considerations. Also, to make the paper accessible to users will require a careful redesign of the notation that is used in the paper, which is hopelessly complex.

Some specific comments:

37:     It is unclear why using different scenarios would result in different estimates of the counter-factual world.

It would help if a bit more were said here. Reading ahead, it turns out that this is discussed more beginning at line 113 and I can see from that discussion how this could arise. It is not made clear, however, whether the selection of a particular scenario would strongly affect inferences about observed events based on the posterior that results from using that scenario versus inferences that would be made with another scenario. Sensitivity to scenario choice, particularly if large (and especially under counterfactual conditions) would be of concern, but making that go away artificially by using information from all available scenarios doesn't really solve the problem in a satisfying way. It would remain a concern that information from models about the future can somehow affect our understanding of the past unless there is a convincing physical argument about why that makes sense. On the other hand, if the sensitivity is small then there wouldn't be a very compelling reason to bother with the added complexity of the prior and its dependence on the particular experimental design that was adopted in CMIP6. In summary, I think this is crucial point that needs clarification (and in each application, physical justification).

70:  What is the point of the right-hand panel in Figure 1? It adds a bit of confusion by hinting that you will make inferences about the record event during 1940 to 2024 at every land point in the domain, irrespective of when that event happened or the spatial extent of the event that produced the record.

75-75: Comparison with a single long record that is almost surely inhomogeneous (e.g., due to instrument changes, observing procedure changes, development of the urban environment around the station, etc) is not going to do a great deal to increase confidence.

77:  Usually, "external" forcing would mean external to the climate system (solar, volcanic, ghg's, aerosols ...) rather than external for France ....

77-79: If read literally, the sentence could be interpreted as saying that you extract European mean temperatures from HadCRUT5 and global mean temperatures (more correctly, temperature anomalies) from GISTEMP...

Why do you use these two datasets rather than just using one?

In addition, Fig. S1 notes the use of the BEST dataset (Berkeley Earth) – why use yet another global dataset when consistent use of one, well regarded dataset, would probably suffice?

85-86: My understanding is that the historical forcing prescription used in CMIP6 is NOT part of the SSPs, which only cover the period from 2015 onwards.

90-91: I'm not aware that the IPCC assessed, in its synthesis report, that the current emissions trajectory is leading us towards SSP2-4.5. This is discussed by others, however, so I think you should provide a more suitable reference or delete this statement. Indeed, if you have high confidence in this statement, then it would seem that there would be no need to use the other scenarios.

101:  The certainty expressed here that the variable of interest will be GEV distributed seems a bit of an overstatement. The GEV distribution is a limiting distribution for block maxima that is (sometimes) achieved as the block length grows without bound. Convergence to the limiting distribution (if it happens at all) can only be demonstrated theoretically under very idealized conditions. Nature, and climate models, do not comply with those conditions (we have awkward things like an annual cycle and the presence of multiple extremes processes that complicate life considerably, with the result that the upper tail does not always behave like that expected under idealized mathematical conditions. While we can't really look into the deep upper tail with observations, and can do so, albeit with some difficulty,

with climate models (e.g., see Ben Alaya et al, 2020, DOI: 10.1175/JCLI-D-19-0011.1). Experience shows that the GEV is nevertheless often useful for approximating the distribution of block maxima for blocks of even modest size (e.g., a year, which effectively only samples part of the year due to the annual cycle). The authors know all of this, and it would be good if some of this could be reflected in the paper, particularly as it is intended to introduce the methods and the ANKIALE package to a wide audience who are not as knowledgeable about the application of the GEV distribution and its limitations.

103:   What is the time range considered? Also, I find the notation here somewhat confusing. Readers in a hurry will confound the index F (for factual) with "future", and might confound the index "0" (zero) with "O" (for "observed"). A further question is whether readers should think of the three components of X as being random or fixed.

109:   The use of the * to indicate the reader should make a substitution for R or G is awkward and mostly just makes comprehension a bit more difficult for the reader.

114-115: Replace "supposed" with "assumed". Also, this assumption merits some discussion.

119-120: See the comment concerning line 37 above. This needs discussion – particularly why including different futures would affect our understanding of the past.

126-127: It might just be a French/English problem, but what this first step in the procedure entails could be better explained. This would include saying what the assumptions are that are implicit in calculating the uncertainty covariance matrix. It is not clear from the notation if there is one such covariance matrix that describes the spread amongst the different $\theta_m$'s (which I am guessing is the case) or whether each $\theta_m$ has its own uncertainty matrix.

131-132: It would be much better if this paper could be self-contained rather than sending readers off to another reference for the parts of the methodology that have not changed.

137, 139: While the paper is generally readable, there are many minor grammatical errors. Two examples are mentioned here. This is less excusable these days given the wide availability of tools for polishing text (assuming that GMD authors are permitted to use them).

At line 137, replace "3-days moving average" with "3-day moving averages".

At line 139, where the sentence seems unclear. In that sentence, rather than "are", do you mean "are estimated with"?

147: The white noise assumption needs some justification. This might be roughly suitable for European regional mean annual surface air temperature anomalies, but the while noise assumption seems a bit less obvious for annual global mean surface temperature anomalies.

148-154: I find myself struggling to understand what is really done here, both because of the notation, which is increasingly complex, and because it is not obvious what model output is being used. If a model has 50 ensemble members, do you use all 50? And if so, do you treat that model differently from a model with only 1 ensemble member? What period is considered, how was the choice to use a smoothing spline with only 6 degrees of freedom made, how are the knots placed, do you worry about the fact that the knot placement is arbitrary and that this imposes wave-like fluctuations that are probably not part of the forcing response?

156: Figure S2 is referenced well before Fig S1 …

177: This statement is made with a lot of certainty and conviction, but whether an event would be judged to be impossible, even under anthropogenic forcing, is highly uncertain. It seems clear from Fig. S3 that the value of the shape parameter is driven by the extreme temperature that is farthest from the location parameter and hence must be very uncertain. This relation between the shape parameter and the most extreme temperature presumably occurs because the parameter estimation process enforces the feasibility of the fitted GEV distribution to a variable, temperature, that tends to have light-tailed extreme value distributions.

184-185: Why is this assumption needed to construct the prior? It's a prior distribution (i.e., a proposal) that will be updated using the observations when the posterior is derived. It seems to me that this assumption is not needed to construct the prior. Given the way the prior is constructed, it would certainly be helpful if we can regard the models as being indistinguishable from each other (i.e., something hopefully like a simple random sample from model space), but even without that, couldn't we construct a prior from the models, understanding that it may not do a good job of representing model uncertainty? The updating does require us to have a joint distribution for model simulated and observed quantities, and developing that joint model may require some additional assumptions – perhaps that's where the "indistinguishable from the truth" assumption comes into play?

189-190: I think this needs discussion – in particular, why internal variability plays a role at all (haven't you filtered it out with the splines?) and what is being partitioned into two components. What is being referred to when you talk about the "common part" of internal variability and each model's additional internal variability??

200: I have no idea what is being referred to here (95% of the covariance matrix)...

202: Which model is excluded? Note that the UK models maybe be problematic due to a known problem in the coupling between the land surface and atmosphere that leads to extreme high localized daily maximum temperatures. The problem is documented at https://errata.ipsl.fr/static/view.html?uid=76b3f818-d65f-c76b-bfd8-cae5bc27825c. Note that the Australian ACCESS models, which also use a version of the UK MetOffice atmospheric model, are not affected but the Korean KACE model, which uses both the atmospheric model and the Jules land surface model, is affected (an erratum has not been published for the KACE model).

238-239: Why not use internal variability estimated from climate models rather than relying on the one, very limited realization we have been able to observe?

---

## Referee Comment (RC3)

**Manuscript EGUsphere-2025-1121 'A Bayesian statistical method to estimate the climatology of extreme temperature under multiple scenarios: the ANKIALE package' by Robin et al. Referee's report**

The main purpose of this paper seems to be to improve on a methodology for extreme event attribution that was proposed by two of the authors. There are two main improvements: one is the ability to consider multiple emissions scenarios simultaneously, while the other relates to the adoption of faster algorithms for the analysis. In scientific terms, the first of these is much more interesting. Regarding the algorithms: it's useful to know that the software is faster and I'm aware that the implementation can be a lot of work, but I wouldn't regard it as a headline message for the paper, especially it seems to come just from adopting a more modern (but now fairly standard) MCMC method.

I'm not aware of other work that attempts to incorporate information from multiple emissions scenarios simultaneously within the framework of a single statistical representation, as is done here: most analyses treat individual scenarios separately which, as the authors correctly point out, leads to potential inconsistencies when focusing on quantities that are not scenario-dependent. As a first attempt to highlight this issue therefore — albeit in an application where the role of the scenarios needs to be explained more clearly (see general comments below) — this paper is welcome.

Unfortunately however, I think it needs a very major rewrite before it can be considered publishable. As far as I can tell, the methodology is *probably* consistent with many approaches to attribution — although I think it may be unnecessarily complicated and the approach to fitting the GAMs is possibly not optimal (see below). However, the manuscript is poorly written and, as a result, I find it hard to understand exactly what is being done and why. The scientific content is therefore hard to evaluate.

The main problem is that the paper doesn't contain all of the information needed to make sense of it. There are also problems with the organisation of the material; with terminology that is not properly defined or is used incorrectly; and with mathematical notation that is not always defined or explained. Conversely, the paper makes heavy weather of some things that are actually rather straightforward (for example, the estimation of GAMs with a common component across scenarios — see below).

Comments and questions follow. Some of them are probably due to my lack of understanding, for which I apologise. I have done my best, within the 12 hours or so that I'm willing to spend on a review.

**General comments**

1. The paper is not self-contained. The authors state explicitly that they will only describe the changes relative to their earlier paper (denoted as RR20); but a reader should not have to look at another paper in order to understand this one. A brief but complete summary of the relevant points from RR20 is needed.

2. There is no clear statement of what the analysis is for. The abstract (line 1) says that it's about estimating the statistics of temperature extremes; but the paper itself is framed around the methodology of attribution studies. Reading through the paper for the first time, I didn't know where it was going or why. To give just one example: at some point the aim seems to be the estimation of a quantity called $\kappa$ which is described as "the random variable of a multi-model synthesis" (line 128): I still don't know what this represents, and from here on I started to get completely lost. It would

be helpful to include, early on, a clear statement of what you're trying to estimate and to explain, in broad terms, the information requirements (I don't mean the precise datasets used: rather, the need for real-world time series of TX3x and whatever else you need for the real world; and GCM simulations of [etc. etc.]). It might also be helpful to include a schematic diagram — again, early in the paper — showing how all of these quantities fit together in the steps of the analysis.

3. I don't understand the role of the scenarios in the context of an attribution study. Scenarios relate to *future* climate (albeit starting in 2015 for the CMIP6 runs), whereas attribution studies typically work with information on the past and present. It's possible that the scenarios allow more precise estimation of anthropogenic effects using the GCM outputs, but this isn't explained anywhere. It's also possible that I have missed the point — but in that case it hasn't been explained clearly enough.

4. In some way related to the previous two points: throughout, the paper attempts to describe *what* was done, but there needs to be more justification for *why* it's being done.

5. The approach uses the GCM outputs to place a prior distribution on real-world quantities, based on an assumption that the GCMs are centred on reality (lines 184–185). This is known to be unrealistic: climate models are collectively biased relative to the real-world, and the biases are not independent. I am aware that the authors' assumption is often made in the literature. However, the paper should at least provide an honest acknowledgement that it is known to be (very) wrong — this is obvious to anyone who understands CMIP, and Reto Knutti and co-authors have demonstrated it empirically beyond doubt. There is also literature that aims to address the problem by relaxing the assumption e.g. to a notion of co-exchangeability (defined by Rougier et al. 2013, doi 10.1080/01621459.2013.802963).

   I can imagine an argument that the precise assumptions about the GCMs don't matter too much because they are only being used to set a prior. My response to that would be: if the prior doesn't influence the results then you don't need it because the real-world data are already sufficiently informative; but if the prior *does* influence the results then it needs to be defensible. The work of Knutti et al. shows that this prior is not even *close* to being defensible. At the very least, there should be some investigation of sensitivity to plausible alternative prior choices.

   A minor point, related to this isssue, is that the formulation doesn't date back to Ribes et al. (2017) as claimed on line 185 — it goes back much further.

6. The proposed model seems needlessly complicated. I'm not 100% sure of this, because the presentation around equation (1) is unclear (e.g. in lines 106–107 we learn that this equation wasn't actually used). I am also aware that the proposed methodology arises from a line of reasoning that is accepted in the attribution literature. Nonetheless:

   - As presented, the model seems overparameterised. For example, there are constant terms in the model for $X_t^{R,F}$ in equation (1), but also in the models for $\mu_t$ and $\log \sigma_t$. These constant terms can presumably be merged: the only reason for distinguishing between them is the workflow which splits the analysis into a first stage analysing the global and regional temperatures, and a second stage in which the results are fed into the GEV estimation. It's not clear to me why the analysis should be split in this way: surely the Bayesian computational machinery allows you to do everything in one step? A clear justification for the two-step approach is needed.

   - More fundamentally perhaps: I am aware of the fashion for regressing on global and regional mean temperatures in the attribution literature. I am also aware that this literature tends

to focus on partitioning variation into natural and anthropogenically-forced components. If I understand correctly however, the partitioning here is derived (lines 140–147) from a regression on either the outputs of an energy balance model (EBM), or the net radiative forcings. It's not clear which option has been used to obtain the presented results (or, if it was an EBM, which one was used). However, the results in (for example) Figures S1 and S2 show a strong resemblance between the "naturally-attributed" component of global mean surface temperature and the natural component of the forcings (not shown in the paper, but I'm familiar with the forcing time series). I guess that the correlation between this component and the input forcings is at least 0.95, and possibly greater than 0.99 — indeed, it will be 1 if the forcings themselves have been used in the estimation procedure. Moreover, the estimated anthropogenic components of the global and regional temperatures will be smooth curves, just like the anthropogenic component of the forcings (or of the EBM outputs). I will therefore bet a moderate sum of money that one can obtain almost identical results to those reported in the paper by ignoring the global and regional mean temperatures altogether, and instead using the natural and anthropogenic components of the net radiative forcing in the models for the GEV parameters. This would simplify the analysis both conceptually and computationally (e.g. it would eliminate the need for GAMs).

The attribution community may consider this suggestion to be very oversimplistic. Nonetheless, I would be interested to see how it performs. With the ANKIALE software as described, this should not be hard to implement: just replace the estimates of $X_t^{*,N}$ and $X_t^{*,A}$ with the corresponding components of the net radiative forcing, and see what happens.

7. The split of figures and tables between the main paper and supplement seems strange. For example, Figures S1–S3 are helpful in understanding the process and the results, but they have been relegated to the supplement. On the other hand, Tables 2 and 3 in the main paper take up a lot of space and will be of interest to relatively few potential readers: these could reasonably be moved to the supplement.

**Specific / detailed comments and queries**

1. Line 1 "estimating the statistics of temperature extremes": as noted above, it isn't clear from this what you're *actually* doing.

2. Line 29: typo "compbining".

3. Line 30: a Bayesian *framework*?

4. Lines 33–34: I would remove the sentence "However … inconsistencies". It is redundant given the content of the subsequent paragraph, and only one of the issues is actually an inconsistency.

5. Line 47: the time taken to "process the entire domain" presumably depends on the size of the domain! To put this in context, it would be helpful to indicate the approximate number of grid cells being referred to here.

More generally: if there are any real applications in which it's necessary to carry out an analysis over a domain of this size, then a time scale of up to a week is perhaps still not fast enough. It is certainly too slow for the kinds of real-time attribution studies carried out by World Weather Attribution as

described in lines 18–20, given that the time available for data analysis in such studies is only a fraction of the total time available. For those kinds of applications though, the focus is typically on specific events in much smaller regions. It's certainly helpful to give an indication of computational cost; but it would be more relevant to indicate the cost for realistic applications of the methodology. Alternatively, give (e.g.) an indicative cost per thousand grid cells, so that readers can figure out for themselves the likely cost for their own applications.

6. Line 51: what's the relevance of the number of countries and their partial inclusion? What does "exact ratio" refer to?

7. Line 64: "refer to" $\Rightarrow$ "referred to".

8. Lines 73–74 "we use observations from a weather station ...: But you just said you were using ERA5. Perhaps the station data are used to illustrate the methodology and then, as a separate study, the ERA5 data are used to examine the entire European area. If this is the case, probably it would be helpful for the reader if the paper is reorganised clearly along these lines.

9. Lines 75–76: how does the use of a longer dataset allow you to "better verify the contribution"? Given more data, *any* reasonable method is going to do better. And, if you need an unusually long dataset to identify the benefits of the new approach, then the implication is presumably that the benefits aren't obvious for datasets of more realistic size! (I'm playing devil's advocate, but you need to think more carefully about what you're trying to say).

10. Lines 77–78: as noted above, I believe it is fairly common to use global or regional mean temperatures in attribution studies. Nonetheless, it would be helpful to clarify this — perhaps as part of an introductory summary of the basic steps / logic of the approach (see general point 2 above).

11. Line 85: do the emissions scenarios really cover the historical period? I think the intended meaning is probably that historical estimates of emissions were used for the period 1850–2014, while scenarios based on alternative SSPs were used for the period 2015–2100.

12. Lines 89–90: is there a reference to support the claim that the current warming trajectory is close to the SSP2-4.5 scenario? Or is IPCC (2023) intended to be this reference? As written, it seems to be a reference just for the 2.8K estimate.

13. Line 92 (also line 137): the GCM simulations of TX3x are broadly comparable with the ERA5 estimates because the latter are gridded. However, they are not comparable with the Paris station-based observations. How is this handled?

    While on the subject of gridded data: each GCM has its own grid, and many of them are different from the grid used in the ERA5 dataset. How was this dealt with?

14. Lines 86–97: see general comment 1 above.

15. Line 104: this is where the paper starts to become hard to follow. There is no statement of what the covariate $X_t^{R,F}$ actually *is*: all we're told is that it's a proxy for something else. And how do the quantities in the final line of equation (1) relate to the regional covariate $X_t^R$ mentioned immediately beforehand? And why are these quantities needed, given that the GEV parameters are modelled using the original quantity $X_t^R$?

The answers to some of these questions emerge implicitly later in the paper. If I understand correctly, equation (1) and the subsequent paragraph do not correctly describe what was done. My best guess is that $\mu$ is represented $\mu_0 + \sum_{j \in \{G,R\}} \left( \mu_{N,j} x^{N,j} + \mu_{A,j} x^{A,j} \right)$ with a corresponding expression for $\log \sigma$, where $x^{N,j}$ and $x^{A,j}$ represent respectively the 'natural' and 'anthropogenic' components of variation in the global or regional temperature series. Setting the $\{\mu_{A,j}\}$ and $\{\sigma_{A,j}\}$ to zero then allows an estimate of the GEV parameters in the absence of anthropogenic influence. If this *is* correct, then the presentation in the manuscript seems very over-complicated.

16. Line 105: typo "anthropogenice".

17. Lines 106–107 "We also add . . .": this confirms that equation (1) does not describe what was done. See above.

18. Lines 109–10: it is not correct that the model can be seen as a linear model. It's not linear, and linear models have additive noise terms. Suggest deleting this sentence.

19. Line 112: "construct" $\Rightarrow$ "estimate"? Also, the 'factual / counterfactual' vocabulary seems unnecessary and confusing. I know that it comes from the literature on causal inference, but the present paper doesn't really deal with causality. If my summary in point 15 above is correct, then none of it is really needed — and other explanations can be simplified as well.

20. Line 116: it's not clear to me that all elements of $\boldsymbol{\theta}$ should be described as 'parameters'. The $X$s would be more precisely described as latent or hidden variables. Arguably this is a minor point, but it does affect the readability of the paper.

21. Line 128: what does $\kappa$ represent? See general point 2 above.

22. Lines 131–132: this repeats lines 96–97 (but see general comment 1).

23. Line 143: 'Energy Balance Model' appears twice. And which EBM are you using?

24. Line 145: residuals from what? (actually I don't think they *are* residuals).

25. Line 147: how is the variance of $\varepsilon^*$ represented? Is it the same for all SSPs?

26. Lines 148–151: This presentation is needlessly complicated. If GAMs are needed at all (see general point 6), the model could be fitted directly using the `gam()` function in the `mgcv` package in R, e.g. using the `by()` argument to fit different smooths for the SSPs while retaining the common natural forcing term. The required setup is actually quite straightforward: it may even be available somewhere in python, although I appreciate that there's nothing in python that gets close to the capabilities of `mgcv`. We probably don't need to know the details of backfitting either: this is a completely standard approach to fitting GAMs.

27. Line 153: if it *is* necessary to retain the details on how the GAMs are fitted, then the expression in line 154 is a sum rather than an average as described: either the expression or the description is wrong. And in line 154, I assume both terms are summed: they should be enclosed in parentheses to make this clear, therefore.

28. Line 156: as noted above, it would be helpful to move Figure S2 into the main paper. Also however, there is no indication as to whether this was obtained using radiative forcings or EBM outputs.

29. Line 157 "We can see that ...": how can we see that? What are we supposed to be looking at? How many model realisations are there? If more than one, this hasn't been mentioned in the text (or how you have dealt with it) — it is noted in the caption to Table 1, but if affects the interpretation of results then it needs to be discussed in the text as well.

30. Line 158 "peaks are due to the volcanoes": they are troughs I think, not peaks.

31. Line 159: when referring to the "constant" signal, do you mean constant over time or constant across SSPs? If over time: why is this relevant? If over SSPs: why do the volcanic and solar effects differ between them as suggested at the end of the line?

32. Line 161 "we propose the following method ...": I have no idea what is being done here, or why.

33. Line 162 "resampled": what does this mean? Resampled from what?

34. Line 164: what "method described above" is being referred to here?

35. Line 165 "the four smoothed covariates": what are these? I thought the previous step (described in lines 138–155) was designed to estimate these covariates.

36. Lines 184–188: it is not correct to describe the formulation here in terms of models that are "statistically indistinguishable from reality", which implies that reality and the models are exchangeable. The representation set out here is sometimes called the 'truth plus error' representation, I think.

37. Lines 202–203: I don't know what 'weights' are referred to here, but this sentence suggests that outlying GCMs are excluded from the analysis. No clear justification is given for doing this but, if what's written is correct, it is very worrying because it suggests that the analysis uses only those models that agree with each other. This is very poor scientific practice, and will also lead to underestimation of uncertainty in the final results.

38. Line 207 'Recall that we have ...': when I read the paper, I didn't recognise that this information had been provided previously. All of the relevant information needs to be provided, clearly and unambiguously, at the outset.

39. Line 208 "Our goal is to estimate the distribution ...": really? I was unaware of this on reading the paper. And there has still been no clear statement of what $\kappa$ actually is, or why we might want to learn about it.

40. Line 210: the 'double conditioning' notation in this equation does not exist in conventional practice, as far as I'm aware. I think the denominator is incorrect, as well: what's being done here is Bayes' theorem conditional on $X_t^{*,o}$, which should lead to

$$\mathbb{P}\left(\kappa | T_t^o, X_t^{*,o}\right) = \frac{\mathbb{P}\left(T_t^o | \kappa, X_t^{*,o}\right) \mathbb{P}\left(\kappa | X_t^{*,o}\right)}{\mathbb{P}\left(T_t^o | X_t^{*,o}\right)} \ .$$

I suspect there is also an unstated assumption that $X_t^{*,o}$ is irrelevant for $T_t^o$ once $\kappa$ is known, so that $\mathbb{P}\left(T_t^o | \kappa, X_t^{*,o}\right) = \mathbb{P}\left(T_t^o | \kappa\right)$. Apart from the denominator, this then gives equation (5) in the paper.

41. Lines 211–212 "In other words ...": this makes very heavy weather of quite basic material, while at the same time failing to provide enough information as to what's being done and why.

42. Line 212: "are constrained" by what? Why?

43. Line 216: why do you want "the posterior of global and regional temperatures"?

44. Line 218: in this equation, the left-hand side depends on $t$ but the right-hand side doesn't. I have no idea what is intended. What are the dimensions of the subsequent matrices? What are you doing?

45. Line 224: again, what are the dimensions of these quantities?

46. Line 227: $N^{SSP}$ isn't defined.

47. Line 228: what do you mean by 'null values'? If you mean zeroes, say so.

48. Lines 234–235: what do you mean by "scenarios" and "observations" here? Are the "scenario" data from the climate models? If so, what's the basis for this assumption? After all, the available scenarios (assuming you mean the SSPs that are included in the analysis) are arbitrary: the observations can't be the average over *all* collections of scenarios.

    If the scenario data here are not from the climate models, where *are* they from?

49. Line 237: how is it 'more general' to take the average of the scenarios? All you're doing is to making one choice instead of another.

50. Line 240: what does 'this constraint' refer to?

51. Line 241: 'the covariate Europe'?! At this point, I'm starting to wonder whether all of the authors checked the manuscript before it was submitted.

52. Line 263: use of the NUTS is sensible, but there's no need to provide details of, e.g. the generation of proposal distributions or anything else in the paragraph. The explanation provided isn't enough for readers who are not unfamiliar with the algorithm, and they don't need to understand how NUTS works in any case. All a reader needs to know is that this is a a more modern MCMC method that can often overcome the difficulties described in the previous paragraph.

53. Line 280: what are "the laws"?

54. Line 281: again, 'recall' implies that the reader has been told already.

55. Line 284: what's an 'intensity'? Don't you just have a value of $T$ here?

56. Line 291: I don't think 'deduce' is the right word here. Rather, you can construct measures of the change.

57. Line 295: Figure S4 is another that belongs in the main paper.

58. Lines 297 'we calculate ...': why? **NB** the answer to this may be obvious to someone who understands what the authors are trying to do, but I lost track of that long ago.

59. Lines 301–303: why are you calculating the Wasserstein distance? Few readers will have a good intuition for what a given Wasserstein distance looks like, in terms of the distributions being compared. And most will be interested primarily in whether the distributions differ to an extent that would change any substantive conclusions of interest.

60. Line 312: 'slightly ahead' in what sense?

61. Line 320: is the 'simultaneous analysis of several thousand grid points' often required? Given that you're just doing one grid point at a time (I think), attribution studies would usually focus on specific events that affect a subset of the grid points. More importantly though: in these kinds of applications, the focus is usually on weather that is simultaneously extreme over a large area. As far as I can see, the present paper doesn't address that problem: some discussion of this is needed, as part of the context for the work.

62. Line 326: I wonder whether this kind of "package manual" material is appropriate for inclusion in a journal article? This query is perhaps for the journal editor.

63. Line 377: what does 'since 1940' mean in 'the maximum observed in 2024 since 1940'?

64. Line 389: typo 'conter-factual'.

65. Line 451: what would be the challenges in extending the methodology to other variables such as wind and precipitation? It seems to me that it would be basically the same: it's all just based on the GEV distribution for annual maxima.

66. Caption to Table 2: what are these values averaged over? Grid cells, I assume — it would be helpful to clarify this. See also the earlier suggestion for moving this table (and Table 3) to the supplement.

67. Algorithm 1: this is a completely standard algorithm, and there is no need to include it (but if you do include it, the cross-references at the end are broken — similarly for Algorithm 2 on the next page).

68. I don't find Appendix A very helpful. The material on GAMs in Appendix A1 is mostly fairly standard, but seems more complicated than necessary; while I don't understand Appendix A2 because I still don't know what the purpose is.

Finally: out of interest, before submitting my report I looked at the comments from the other reviewers. There is quite a lot of overlap with my comments above, particularly from Anonymous Reviewer 1 who also found the paper hard to follow. To be completely clear therefore: in this report, everything except this paragraph was written independently and without looking at the other review reports.

Richard Chandler
University College London

---

## Referee Comment (RC4)

**Review - Yoann Robin 2025**

24 November 2020      12:08

This paper extends previously published methods to estimate the changing likelihood from the past to the future of a specified extremes metric due to historical forcings and future emission scenarios. In addition the likelihoods from a counterfactual world, one without anthropogenic emissions is estimated. The new extension to the methodology is enabling a single counterfactual world to be estimated from multiple different scenario runs. The authors also revise the Bayesian approach to reflect advances in this field. For the analysis the authors have developed a software package that is publicly available to facilitate other researchers wishing to perform similar tasks.

My request for major revision is not due to significant concerns of the analysis they have performed but because of the lack of placing their work in the context of the field more generally and the belief that in its present form the text is only accessible to a very small audience. I found what the authors have exactly done and why quite difficult to ascertain and feel that I have only got the general gist rather than a good understanding.

**1) Context of other work**

There is now quite a body of work addressing the production of posteriors of climate variables from the combination of observations with climate models more generally and extreme specific. It would be helpful if they placed their work in the context of these other pieces of work.

**General Bayesian climate projection**

Making climate projections conditional on historical observations.
Ribes, A., Qasmi, S., & Gillett, N.P. (2021). Science Advances, 7(4), DOI: 10.1126/sciadv.abc067

Energy Budget Constraints on the Time History of Aerosol Forcing and Climate Sensitivity
C.J. Smith, G.R. Harris, M.D. Palmer, N. Bellouin, W. Collins, G. Myhre, M. Schulz, J.-C. Golaz,
M. Ringer, T. Storelvmo and P.M. Forster (2021) J. Geophys. Res. Atmos. doi: 10.1029/2020JD033622

Towards consistent observational constraints in climate predictions and projections
Gabriele C Hegerl, Andrew P Ballinger, Ben Booth, Leonard F Borchert, Lukas Brunner, Markus Donat, Francisco Doblas-Reyes, Glen Harris, Jason Lowe, Rashed Mahmood, Juliette Mignot, James Murphy, Didier Swingedouw, Antje Weisheimer (2021) Frontiers in Climate. doi: 10.3389/fclim.2021.678109

Comparing Methods to Constrain Future European Climate Projections Using a Consistent Framework.
Lukas Brunner, Carol McSweeney, Andrew P. Ballinger, Daniel J. Befort, Marianna Benassi, Ben Booth, Erika Coppola, Hylke de Vries, Glen Harris, Gabriele C. Hegerl, Reto Knutti, Geert Lenderink, Jason Lowe, Rita Nogherotto, Chris O'Reilly, Said Qasmi, Aurelien Ribes, Paolo Stocchi, Sabine Undorf
J. Climate (2020) 33 (20): 8671-8692. https://doi.org/10.1175/JCLI-D-19-0953

Quantifying uncertainty in European climate projections using combined performance-independence weighting
Lukas Brunner, Ruth Lorenz, Marius Zumwald and Reto Knutti
Environmental Research Letters, Volume 14, Number 12
Citation Lukas Brunner et al 2019 Environ. Res. Lett. 14 124010
DOI 10.1088/1748-9326/ab492f

UKCP18 Land Projections: Science report
Murphy JM, Harris GR, Sexton DMH, Kendon EJ, Bett PE, Clark RT, Eagle KE, Fosser G, Fung F, Lowe JA, McDonald RE, McInnes RN, McSweeney CF, Mitchell JFB, Rostron JW, Thornton HE, Tucker S, Yamazaki K (2018). https://www.metoffice.gov.uk/pub/data/weather/uk/ukcp18/science-reports/UKCP18-Landreport.pdf

A climate model projection weighting scheme accounting for performance and interdependence
Knutti R, Sedláček J, Sanderson B M, Lorenz R, Fischer E M and Eyring V 2017
Geophys.Res. Lett. 44 1909–18 https://doi.org/10.1002/2016GL072012

A representative democracy to reduce interdependency in a multimodel ensemble
Sanderson B M, Knutti R and Caldwell P (2015), J. Clim. 28 5171–94

Probabilistic Projections of Transient Climate Change.
Glen R. Harris, David M. H. Sexton, Ben B. B. Booth, Mat Collins, James M. Murphy,
Climate Dynamics (2013) doi:10.1007/s00382-012-1647-y. Supplementary Material.

**Extreme specific probabilsitic projection**

UKCP Additional Land Products: Probabilistic Projections of Climate Extremes
J.M. Murphy, S.J. Brown and G.R. Harris (2020)
https://www.metoffice.gov.uk/binaries/content/assets/metofficegovuk/pdf/research/ukcp/ukcp-probabilistic-extremes-report-september-2020.pdf

Climate projections of future extreme events accounting for modelling uncertainties and historical simulation biases
Simon J. Brown, James M. Murphy, David M. H. Sexton and Glen R. Harris
2014 Climate Dynamics https://link.springer.com/article/10.1007/s00382-014-2080-1

**2) Accessibility**

I do not underestimate how difficult this challenge is and it is difficult to know how to advise on this.  Apart from the hurdle of the Bayesian terminology  I think the reader will struggle to put all the pieces together. This could be alleviated with some text at the beginning of the Methods section giving an outline of the whole procedure, how the different parts fit together, the assumptions of the approach,  how climate model deficiencies are accounted for and how biases with observations are dealt with.

For those readers who will probably never be that comfortable with the more statistical aspects of the paper I think they will be greatly helped if there was more emphasis and care in the physical interpretation of the method and discussion of the example analysis. Also, a comparison with other posterior work who's emphasis is more towards the physical modelling uncertainty of future climate
For example such a discussion would include the impact of this being an ensemble of opportunity, how carbon cycle uncertainty and aerosol modelling uncertainty is samples and their consequences for the results presented.  Also a discussion on the relative importance of $X_R$ & $X_G$, what different do they bring to the analysis.

Other general comments
1) I know section 5 is a really important part of the paper but it really does interrupt the flow.  Might the authors consider moving it to an appendix?
2) Are return periods and their changes particularly helpful due to their extremely nonlinear behaviour? The issue with looking at return periods in the present day is that the return period will be dominated by uncertainty in the shape. The actual increase in extreme temperatures, however, will be dominated by the change in location.  Present day factual and counterfactual comparisons if looking at return periods will be dominated by shape uncertainty whereas present to future comparisons of the 100y return level will be dominated by location change and its uncertainty.
3) The adaptation sphere is very focussed on high resolution modelling of the future climate (perhaps too much).  As this paper is also concerned with determining the likelihoods of future extremes it would be useful if the authors could comment on how their method might accommodate regional climate modelling.

4) X_R and X_G - these will be highly correlated. Can you demonstrate that both are required? If you only had one will not the parameters not just get adjusted to compensate? Or put it another way, does the small bit of extra info when using both lead to a significantly better outcome?

Minor comments

**Abstract and Intro**

1. Abstract general: - could be improved to better describe the solution it is providing in more general terms (currently only 81 words)
2. Abstract general:  does not mention the improved Bayesian sampling mentioned in the introduction. This seems a key point of the paper.
3. 3: I think the abstract needs to mention that your "observations" are ERA5
4. 5: "Counterfactual world" - this is jumping in very deep very quickly into DA jargon
5. 29 combbining
6. 24-28 I would be helpful to have some non DA focussed literature on extremes in climate
7. 29-34 very limited review of other literature attempting Bayesian approaches to present/future climate, see above
8. 34-35:  Flow of text is a bit confusing.  It is not clear the next paragraph is addressing these two issues because of the way it starts.  The "Here," suggest the paragraph is setting off on a new topic.
9. 46-47: Could you please compare like with like. Currently it is CPU time vs wall time.  Surely wall time is primarily dependent on how much compute resource you have at your disposal.
10. 53: "observed"  - not true observations should be something like  "as represented in ERA5"
11. 59,60: "classical attribution…specific definition"  Not sure this will make sense to the reader

**2 Data**

1. 64: refer to refers
2. 69:  being pedantic 0h to 23h misses out one hour but I know what you mean, perhaps 0:00 to 23:59?
3. 74: Presumably three is some urban warming in the Paris observations.  Please could you comment on how this affects the results somewhere, perhaps in section 6?

**3 Method**

1. 106: I am confused by the phrase "We add to the model".  Are you suggesting $(X_R + X_G)$?  If so, put it in the definition of eq 1.  But if not perhaps "In addition, we can replace $X_R$ with $X_G$.." or something similar would be clearer.
2. 111: English has gone a bit wrong here
3. 123: The notation for the scenarios is grim.  Could we not have $X_{R,A,S_i}$ and define $S_i$ elsewhere?
4. 123: It would be good to acknowledge the unavoidable assumption that the climate system responds linearly to forcing.  e.g $Mu_t$ and $sigma_t$ are constant wrt different forcing scenarios.
5. 124-130:  This assumes all GCMs are equal.  It would be good to acknowledge this and that other approaches have seen it necessary not to make this assumption (references below)
6. 131-132: this has been said earlier.
7. 143,4: it would be good to say that the energy balance model is forced with natural forcings only and the radiative forcings are natural only.
8. 151 theta_R and theta_R - second should be theta_G?
9. 171: Fig S3 seems to indicate sigma_t is very very small.  Worth commenting on the physical significance I think.
10. 171,200: Fig S3 & Fig 2c-f and their discussion.  I think it would help the reader if you reminded them that these plots are site specific using your Paris observations
11. 177: I think you can only say *impossible* if the probability of the shape being >=0 is zero.  I don't think you have yet shown this.
12. 189: m is used as a superscript and a subscript in this line.  Is this what you mean?
13. 189: This is the first mention of bias between the climate models and truth which is very late in the paper. Too late.  There should at least be a pointer earlier in the text that climate model bias is

addressed later in the method description.  Also I don't think you can deal with the issue of climate model bias in just ~12 words!  What sources of bias is it accounting for?  In The GEV parameters?  In the covariates?  A description of how and how well is surely needed.

14. 196: "grid point containing" earlier you were fitting to station obs for the plots (l137) - has it changed?
15. 196 Fig2b: 1940-1960 All GCMs are cool wrt Obs.  Please comment.
16. 201: black ellipses missing
17. 241: covariate FOR Europe

**4 Comparison**

1. Section 4 I'd suggest renaming to "Comparison with the independent scenarios method".
2. 275: What are the consequences of mu_1 being so different for obs vs gcm? Some would argue that if the climate model is so biased can we trust its physical representation of real world extremes.
3. 280: "Based on the estimates of the laws" - not sure this will mean very much to most people.
4. 302: "average energy" - I wonder if this is the right term in a geophysical journal?  Joules?
5. 304:  I do wonder how much the average reader will get out of Fig 3 and I note that there is not that much discussion in the text for it.  Perhaps just have the last column in a 2x2 format?
6. 314: "does a good job"  - perhaps a bit too colloquial?
7. Is there a low bias in ERA5 for some regions? eg UK had 40C in 2022 although this was only a single day.  Kay et al. (2025) has much lower return periods for single day events and one would have thought that to first order changes in return period will be somewhat similar for different metrics of extreme hot temperatures
8. I think maps of GEV parameters would be very interesting to most readers, say for 2000, 2024, 2080? In the supplementary info if needs be.

**6 Example**

1. Fig 4 is an odd beast.  It seems like it is implicitly assuming that the climate has been stationary between 1940 and 2024.  For example two points near-ish to each other might have seen the same max temperature at very different times, say 1940 and 2024 for arguments sake.  The probability of those two events are very different as the 2024 climate is much hotter.  Yet the calculation of 4c assumes they have the same probability of occurring.  At least could we have a complimentary plot of the 100 year return level (or whatever) with and without human influence and the difference please? Also for the caption I found "mean of all scenarios" a bit confusing as during the observed period the forcings are the same?  I think it would be clearer to say with and without human influence.
2. 373:  given the lack of spatial dependence perhaps a warning that these numbers cannot be used to calculate the likelihood of a hot event occurring in a given region or country without a correction to account for extremal dependence.
3. 383-387: I find this spatial variation in return periods across quite small distances alarming.  Some of this will be due to the issue in my point above but it does not seem physical.  You could check how the observed exceedance rates compare between different locations. e.g.
   a. Those areas where 4b shows very high return periods.  What are the empirically observed exceedance rates?  Are hot events occurring more frequently than predicted here?
   b. Also for the regions in 4b with very frequent return rates, N Africa, E Turky are we seeing their frequent occurrence in the observations?
   c. And eastern France and western Germany (approx. Nancy & Stuttgart) to see if the different return expectations from the plots in these two places are supported by the data.
   d. Kay et al 2025 found the 2022 UK record event of exceeding 40C (admittedly a single day maximum, but one would expect different averaging periods to be somewhat in step) to be 1 in 24 years which seems rather at odds with your plot (4b) if >500y for the region where these temperatures occurred.  Comment?
4. 385  I stumbled a bit over western and southern Asia.  Perhaps "southern Caucasus" ?
5. 413: Don't think you need to repeat the results of the counterfactual world here as there is no reason for them to be different to 2040?
6. 404:  these holes really bother me and I think they are an artifact of the methodology.  Please can you diagnose their cause.  Are they coming through the GEV terms or the X* terms?  Maps of all of these

parameters at 2040 and 2100 (Mu, sigma, psi, and all the Xs) would be helpful in this regard.

**7 Conclusions**

1. Section 7 seems to be an afterthought or the authors have run out of steam (not surprising). It would be good to have:
   a. A candid discussion of the limitations of the approach
   b. A more thorough comparison of results with other studies
   c. One of the difficulties this paper will have will be convincing people that the pain of understanding it and using the code is worth the effort. At present the case for is not very clear, at least to me. What might help is a companion plot of Fig S3 For Paris for a given return period say 1 in 100 years with uncertainty (ideally including a profile likelihood approach, see Coles 2001 Fig 2.3) , for the method presented here vs a non stationary univariate GEV maximum likelihood approach with Xt's calculated directly from the GCMs used.

2. 434: From my perspective very little verification has been undertaken with respect to current literature. Please could you explain why you think this statement is justified?

3. 456: section 7.2 is rather niche and that there are far larger issues that would benefit having a discussion on. Such as how this work compares with other work that attempt to produce future posterior distributions of climate variables. I include references to such studies above.

---

## Author Comment (AC2)

**Response to Reviewer 2 comments about the article "A Bayesian Statistical Method to Estimate the Climatology of Extreme Temperature under Multiple Scenarios: the ANKIALE Package"**

ROBIN, Y., VRAC, M., RIBES, A., BARBAUX, O. and NAVEAU, P.

October 23, 2025

**Note** In this document, the text in regular format corresponds to the reviewers questions. The answers from authors are given in the grey blocks.

**1 Reviewer 2 (Anonymous)**

**1.1 General comments**

This manuscript by Robin et al. (2025) presents a significant methodological advance for the attribution of extreme temperature events, introducing a Bayesian framework and an open-source tool that enables the simultaneous treatment of multiple climate scenarios. The authors' approach addresses a key gap in the literature by ensuring consistency across scenarios and by rigorously propagating uncertainties from both models and observations. The manuscript is technically ambitious and, in my view, represents an important contribution toward making attribution studies more stringent and transparent. In particular, the explicit handling of the full range of uncertainties, rather than relying on single-scenario or point estimates, sets a new standard for rigor in this field. I commend the authors for this achievement, and I believe their work will be highly valuable for the climate science and risk assessment communities.

We thank the reviewer for her/his summary and for the appreciation of our work. We hope that the answers to the various questions below will be satisfactory.

**1.2 Specific comments**

**1.2.1 Code testing**

Since GMD encourages reproducibility, I attempted to install and test the method myself. I appreciate the effort the authors have made to explain function calls and to provide well-structured code repositories with installation instructions. However, I was unable to install the prerequisite package SDFC, and did not pursue troubleshooting further. I recommend that the authors test the installation process in a clean environment, without assuming prior package installations (such as a default conda setup) or advanced Python knowledge on the part of users, to ensure accessibility for a broader audience.

We are sorry that you were unable to test our tools, and that you were particularly blocked by the installation of SDFC. This is indeed quite tricky because this package is partially written in C++ and needs to be compiled. The SDFC documentation specifies how the installation should be done, even if it is not necessarily obvious.

Hence, we now do hope that the installation will be easier and smoother.

**1.2.2 Statistical nomenclature**

The manuscript introduces X as covariates from L104 onward, which is standard in statistical modelling. However, in line 118, the definition of the parameter vector  $\theta$  includes these covariates alongside the scalar GEV parameters ( $\mu_0, \mu_1, \dots, \xi$ ). This may cause confusion for readers familiar with statistical notation, as covariates are typically considered as observed or input variables, while parameters are the quantities to be estimated (often scalars).

I understand that in your Bayesian framework, the covariates themselves are uncertain and inferred from the data (due to model differences) and thus are treated as random variables. However, it would be helpful to add a clarifying sentence or two to explicitly distinguish between:

**Parameters** (e.g., the GEV parameters  $\mu_0$ ,  $\mu_1$ ,  $\sigma_0$ ,  $\sigma_1$ ,  $\xi_0$ , which are scalars to be estimated, and **Uncertain covariates** (e.g.,  $X^{R,0}$ ,  $X^{R,N}$ ,  $X^{R,A}$ ), which, although treated as part of the parameter vector in the Bayesian synthesis, conceptually represent covariate trajectories or functions rather than fixed parameters. (Uncertain covariates may not be the best description, maybe you find a better one).

A brief clarification in the text would help readers understand why covariates appear in the parameter vector and how their uncertainty is handled in your framework.

There are numerous parameters, covariates, and coefficients defined throughout the manuscript, often annotated with various sub- and superscripts. If possible, I recommend retaining only those notations that are essential for understanding the material, and ensuring that their use is consistent throughout the text. For example, in L189, the m in  $v_m$  appears as a subscript, but it likely should be a superscript. Inconsistent or unclear notation makes it difficult for the reader to follow the argument, as it is not always apparent whether a symbol refers to a new concept or simply a different aspect of an existing one. Careful attention to notation and a streamlined set of symbols would greatly improve the manuscript's readability.

Indeed, a delicate point that is unusual in this type of article is that the covariate itself is random, and therefore part of the vector  $\theta$ . Section 3 has been completely reworked to simplify and clarify the presentation of the method.

**1.2.3 Grammar and spelling mistakes**

While reviewing the manuscript, I noticed a considerable number of grammar and spelling mistakes throughout the text. Many of these issues could likely have been avoided with more thorough proof-reading or by using automated spelling and grammar checking tools. This has also led to a certain fatigue during the review of this manuscript, so please be prepared for additional comments in a second review round. I understand from my own experience that such errors can easily slip through, my own manuscripts have certainly not been immune to this. Nevertheless, I would encourage the authors to carefully revise the manuscript for language quality, as this will significantly improve readability and the overall impression of the work.

We apologise for this inconvenience and will be much more careful with the next iteration of the manuscript.

**1.3 Technical corrections**

Line 16 Please correct "which consists in establishing" -> "which consist in establishing".

Done.

**Line 19** The phrase "has made a specialty of producing attribution studies within a short time (delay of the order of a week) following the occurrence of an event" is awkward. Consider revising to: "has specialized in producing attribution studies within a short time (typically within a week) following the occurrence of an event."

Done.

Line 24 Do you mean "the statistical distribution"?

Obviously, we have reformulated.

Line 29 "compbining"

Corrected.

**Line 30** "within a Bayesian framwork"?

Done.

**Lines 34-37** Which are the two inconsistencies? Could you please expand on this part a bit more. Assuming that everyone has the RR20 paper in mind is probably a bit of a stretch. In which sense can inconsistencies arise across scenarios? May it be that parameters that should be estimated the same way independently of the scenario (shape parameter in the counterfactual for example) are not the same?

In the revised article, we now discuss these inconsistencies further.

Line 38 THE probability ratio.

Done.

Line 49 An an...

Done.

Fig1 referring to the UNSD (2020) M49 norm -> Not quite sure why this is important here.

Indeed, it is of little importance from a climate perspective. The idea here is simply to point out that the division of the domain is not arbitrary, but follows a standard (potentially recognised by all) that does not depend on us.

**Line 57** I believe "Section 4" should not be abbreviated in the beginning of a sentence. It is true. Corrected. Line 57 Is it not rather "where estimates are derived independently for different scenarios"? (can 'scenarios' be estimated?) Corrected. **Line 60** Which 'specific definition'? We have reformulated. Line 67 refer -> referred Done. **Line 85** 1850 – 2014 Done. **Line 94** "as well as the future projections of the four SSP scenarios described above". Done. **Line 101** As  $T_t$  is a set of maxima? This is the random variable derived from annual maxima, we have reformulated. **Line 105** "The covariate  $X_t^{R,F}$  is the sum of.." Done. **Line 107** Please add  $X_t^{G,F}$  also to Eq. 1, as it's not a different model (right?), otherwise it's not quite clear what you mean. Added in the new methodology section. **Line 108** Indirectly or independently? Indirectly. The dependence between the global and regional covariates is taken into account. **Line 111** "so we are.." -> this sounds too colloquially. We have reformulated. **Line 113** the choice [...], is entirely Done.

**Lines 115-116** In Robin.. => Please rephrase this sentence.

We have reformulated.

**Line 124** "in the case" -> "for the case"?

Done.

**Line 126** I believe does not fit an estimation, rather use infer?

Done.

**Line 128** Please (already here!) provide some more context on what you mean with a "multimodel synthesis. Is it a random vector estimated across various models?

Yes, it is. We have reformulated.

**Line 134** There is no  $\theta^m$  in Eq. (2)

We have reformulated.

Line 138 "Let us start" -> too colloquial, covariateS

Done.

**Line 139** ".. are derived using GAMs" (as  $X_t^{R,F}$  and  $X_t^{G,F}$  are fits of a GAM model to data, I don't think one should refer to the fits as models).

The entire Section 3 has been rewritten.

**Line 143** twice Energy Balance Model. Can you please provide more context what this refers to, as all GCMs are in some sense energy balance models.

The entire Section 3 has been rewritten.

**Line 151** twice  $\theta^R$

Done.

**Line 154** I.e. per time step, is it the average over four values?

Yes. However, we changed our approach between the two iterations. So, this has been modified.

**Line 156** I would strongly suggest to keep Fig S2 in the main text.

A new figure synthesising Figures 2, S2 and S3 has been created and is now included in the main text.

**Line 161** Draw the vectors -> Not rather 'estimate'?

Corrected.

Line 166 derive the GEV parameters

Done.

**Line 167** with the  $T_t^{SSP}$  series as target variable?

Yes, we have reformulated.

**Line 169** Why do we get different results? Different starting values for the MLE optimisation?

No, the optimization for each scenario gives a different estimation.

Line 171 no "is"

Done.

**Line 184** Can you please outline briefly what this hypothesis means? Does it refer to the assumption that the various climate simulations could all be potential realisations of actual climate, aside from a bias term  $v^m$ ?

The entire Section 3 has been rewritten.

Line 203 Which model has been excluded?

This is the Norwegian Earth System Model NCC / NorESM2-LM (Seland et al., 2020).

**Line 238-239** Why not detrend the observations with a GAM before calculating the variance?

The whole problem lies in estimating this trend in the observations, especially if we want to take uncertainty into account.

**Line 254-270** This is a quite general description of Bayesian sampling techniques. I would suggest to drastically shorten it and put the extended version into the supplementary material.

The entire Section 3 has been rewritten.

**Line 311** Where do we see the thing about SSP370 in Fig S5?

As the differences were not clear, we completely revised our analysis in order to better highlight the potential contributions (See the new Sect. 3.4).

Line 313 Missing full stop.

Done.

**Line 312** "does a good job" -> too colloquial

We have reformulated.

**Line 325** "allowing to reproduce" -> "allowing reproduction of the results presented in this paper."

Done.

Line 344 model data

Done.

**Line 344** This is quite a specific format for a NetCDF. Do I understand correctly that most NetCDFs will require reformatting before they could be used here?

Indeed, the input data uses a specific format for NetCDF, and the data must be converted to the correct format.

**Line 357** Figures S1 is only referenced now.

This has been corrected.

**Line 361** Make it clear that this is an example application.

Done.

**Line 372** The sentence "No form of spatial dependency is taken into account, so the existence of an event at one place does not imply anything at another." could be clearer as: "No spatial dependency is considered, so the occurrence of an event at one location does not imply anything about another location."

This is corrected.

**Line 375** Maybe add maps of parameter estimates, too see how much they different from grid point to grid point.

A figure containing the maps of  $\mu_0$ ,  $\mu_1$ ,  $\sigma_0$ ,  $\sigma_1$ ,  $\xi_0$  and the 1961/1990 anomaly of TX3x has been added.

**Line 378** Change in intensity -> Does that refer to a change in intensity if the return period of the event is assumed the same under both factual and counterfactual conditions?

Yes, we have reformulated.

**Lines 391/391** I would replace the lower value by lower bound.

This could lead to confusion with the GEV distribution.

Line 421 What is the end of the sentence supposed to mean?

We have reformulated.

**Line 450** Maybe not directly applicable in this scenario, but I believe the paper by Jewson et al. (2025) is very relevant for this community.

Thanks, the citation has been incorporated into the rewritten conclusion.

**Fig S1** What are BEST observations?**

When we began this work, we used BEST to estimate the GMST. As BEST is known to have a warm bias, we then switched to GISTEMP. So it is an error that has crept into the text. This is now corrected.

**Bibliography**

Jewson, S., T. Sweeting, and L. Jewson (Feb. 2025). "Reducing Reliability Bias in Assessments of Extreme Weather Risk Using Calibrating Priors". In: *Adv. Stat. Clim. Meteorol. Oceanogr.* 11.1, pp. 1–22. ISSN: 2364-3579. DOI: 10.5194/ascmo-11-1-2025.

Robin, Y., M. Vrac, A. Ribes, O. Barbaux, and P. Naveau (May 2025). "A Bayesian Statistical Method to Estimate the Climatology of Extreme Temperature under Multiple Scenarios: The ANKIALE Package". In: *EGUsphere*, pp. 1–41. DOI: 10.5194/egusphere-2025-1121.

Seland, Ø., M. Bentsen, D. Olivié, T. Toniazzo, A. Gjermundsen, L. S. Graff, J. B. Debernard, A. K.

Gupta, Y.-C. He, A. Kirkevåg, J. Schwinger, J. Tjiputra, K. S. Aas, I. Bethke, Y. Fan, J. Griesfeller, A. Grini, C. Guo, M. Ilicak, I. H. H. Karset, O. Landgren, J. Liakka, K. O. Moseid, A. Nummelin, C. Spensberger, H. Tang, Z. Zhang, C. Heinze, T. Iversen, and M. Schulz (Dec. 2020). "Overview of the Norwegian Earth System Model (NorESM2) and Key Climate Response of CMIP6 DECK, Historical, and Scenario Simulations". In: *Geosci. Model Dev.* 13.12, pp. 6165–6200. ISSN: 1991-959X. DOI: 10.5194/gmd-13-6165-2020.

---

## Author Comment (AC3)

**Response to Reviewer 3 comments about the article "A Bayesian Statistical Method to Estimate the Climatology of Extreme Temperature under Multiple Scenarios: the ANKIALE Package"**

ROBIN, Y., VRAC, M., RIBES, A., BARBAUX, O. and NAVEAU, P.

October 23, 2025

**Note** In this document, the text in regular format corresponds to the reviewers questions. The answers from authors are given in the grey blocks.

**1 Reviewer 3 (Richard Chandler)**

**1.1 Global comment**

The main purpose of this paper seems to be to improve on a methodology for extreme event attribution that was proposed by two of the authors. There are two main improvements: one is the ability to consider multiple emissions scenarios simultaneously, while the other relates to the adoption of faster algorithms for the analysis. In scientific terms, the first of these is much more interesting. Regarding the algorithms: it's useful to know that the software is faster and I'm aware that the implementation can be a lot of work, but I wouldn't regard it as a headline message for the paper, especially it seems to come just from adopting a more modern (but now fairly standard) MCMC method.

Thank you for this summary and for highlighting these two important points. We would like to point out that the targeted journal (GMD) deals with theoretical developments as much as technical developments. We therefore believe that both messages are important.

I'm not aware of other work that attempts to incorporate information from multiple emissions scenarios simultaneously within the framework of a single statistical representation, as is done here: most analyses treat individual scenarios separately which, as the authors correctly point out, leads to potential inconsistencies when focusing on quantities that are not scenario-dependent. As a first attempt to highlight this issue therefore — albeit in an application where the role of the scenarios needs to be explained more clearly (see general comments below) — this paper is welcome.

Unfortunately however, I think it needs a very major rewrite before it can be considered publishable. As far as I can tell, the methodology is probably consistent with many approaches to attribution — although I think it may be unnecessarily complicated and the approach to fitting the GAMs is possibly not optimal (see below). However, the manuscript is poorly written and, as a result, I find it hard to understand exactly what is being done and why. The scientific content is therefore hard to evaluate.

We apologise that our initial submission were difficult to evaluate. Your comments were therefore very welcome and have greatly helped us to improve our article.

The main problem is that the paper doesn't contain all of the information needed to make sense of it. There are also problems with the organisation of the material; with terminology that is not properly defined or is used incorrectly; and with mathematical notation that is not always defined or explained. Conversely, the paper makes heavy weather of some things that are actually rather straightforward (for example, the estimation of GAMs with a common component across scenarios — see below).

Thank you for your comments. We have completely rewritten the methodology section, trying to simplify the notation as much as possible without losing the meaning. It now contains a simple general idea of the method, with the example of Paris to help build intuition. The mathematical details have been moved to the appendix and supplementary material. We hope that this new organization will make it easier to read and understand.

Comments and questions follow. Some of them are probably due to my lack of understanding, for which I apologise. I have done my best, within the 12 hours or so that I'm willing to spend on a review.

**1.2 General comments**

1. The paper is not self-contained. The authors state explicitly that they will only describe the changes relative to their earlier paper (denoted as RR20); but a reader should not have to look at another paper in order to understand this one. A brief but complete summary of the relevant points from RR20 is needed.

The methodology section has been completely rewritten, and the appendices and supplement have been completed. The paper now contains all relevant details of the methodology.

2. There is no clear statement of what the analysis is for. The abstract (line 1) says that it's about estimating the statistics of temperature extremes; but the paper itself is framed around the methodology of attribution studies. Reading through the paper for the first time, I didn't know where it was going or why. To give just one example: at some point the aim seems to be the estimation of a quantity called  $\kappa$  which is described as "the random variable of a multi-model synthesis" (line 128): I still don't know what this represents, and from here on I started to get completely lost. It would be helpful to include, early on, a clear statement of what you're trying to estimate and to explain, in broad terms, the information requirements (I don't mean the precise datasets used: rather, the need for real-world time series of TX3x and whatever else you need for the real world; and GCM simulations of [etc. etc.]). It might also be helpful to include a schematic diagram — again, early in the paper — showing how all of these quantities fit together in the steps of the analysis.

The abstract and introduction have been rewritten for clarity, as well as the methodology section. A figure illustrating the whole procedure has also been added.

3. I don't understand the role of the scenarios in the context of an attribution study. Scenarios relate to future climate (albeit starting in 2015 for the CMIP6 runs), whereas attribution studies typically work with information on the past and present. It's possible that the scenarios allow more precise estimation of anthropogenic effects using the GCM outputs, but this isn't explained anywhere. It's also possible that I have missed the point — but in that case it hasn't been explained clearly enough.

In an attribution context, scenarios allow us to project a current (real) event or class of events into the future to see how it will behave. We then speak of "prospective attribution". Using climate models also allows us to strengthen the estimation of the parameters that drive non-stationarity (in particular  $\mu_1$  and  $\sigma_1$ ). We have rephrased this to better highlight and clarify these different points.

4. In some way related to the previous two points: throughout, the paper attempts to describe what was done, but there needs to be more justification for why it's being done.

We have rephrased that.

5. The approach uses the GCM outputs to place a prior distribution on real-world quantities, based on an assumption that the GCMs are centred on reality (lines 184–185). This is known to be unrealistic: climate models are collectively biased relative to the real-world, and the biases are not independent. I am aware that the authors' assumption is often made in the literature. However, the paper should at least provide an honest acknowledgement that it is known to be (very) wrong — this is obvious to anyone who understands CMIP, and Reto Knutti and co-authors have demon-strated it empirically beyond doubt. There is also literature that aims to address the problem by relaxing the assumption e.g. to a notion of co-exchangeability defined by Rougier et al. (2013).

I can imagine an argument that the precise assumptions about the GCMs don't matter too much because they are only being used to set a prior. My response to that would be: if the prior doesn't influence the results then you don't need it because the real-world data are already sufficiently informative; but if the prior does influence the results then it needs to be defensible. The work of Knutti et al. shows that this prior is not even close to being defensible. At the very least, there should be some investigation of sensitivity to plausible alternative prior choices.

A minor point, related to this isssue, is that the formulation doesn't date back to Ribes et al. (2017) as claimed on line 185 - it goes back much further.

We never assume that GCMs are centred on reality, an assumption that is obviously false, as you point out. Indeed, there is a problem with our notations, and probably an error in what we wrote in the initial submission. Here is the approach we take, which is the one used by ANKIALE. The prior is constructed as a synthesis of different climate models, using the following hypothesis: "the models are statistically indistinguishable from truth", developed by Ribes et al. (2017). Let:

- $\theta_* \sim \mathcal{N}(\hat{\theta}_*, \Sigma_{\hat{\theta}_*})$  be the desired multi-model synthesis
- $\theta_{\mathcal{M}}$  the mean response of an infinite set of models, and  $\Sigma_{\mathcal{M}}$  the climate modelling uncertainty (assumed to be equal for each model);
- $\check{\theta}$  the "truth" (not assumed to be equal to the multi-model mean).

The indistinguishability hypothesis is equivalent to saying that  $\theta_m$ ,  $\theta_*$  and  $\check{\theta}$  all come from the same distribution, i.e.  $\theta_m$ ,  $\theta_*$ ,  $\check{\theta} \sim \mathcal{N}(\theta_{\mathcal{M}}, \Sigma_{\mathcal{M}})$ . Note the difference with the "truth plus error" hypothesis (not used here), where the  $\theta_m$  are centred on reality, i.e.  $\theta_m \sim \mathcal{N}(\check{\theta}, \Sigma_{\mathcal{M}})$ .

In the new version of our document, we have clarified the notation, and all the mathematical details are given in the supplementary material, Sect. S.1.2.

- 6. The proposed model seems needlessly complicated. I'm not 100% sure of this, because the presentation around Eq. 1 is unclear (e.g. in lines 106–107 we learn that this equation wasn't actually used). I am also aware that the proposed methodology arises from a line of reasoning that is accepted in the attribution literature. Nonetheless:
  - As presented, the model seems overparameterised. For example, there are constant terms in the model for  $X_t^{R,F}$  in equation (1), but also in the models for  $\mu_t$  and  $\log \sigma_t$ . These constant terms can presumably be merged: the only reason for distinguishing between them is the workflow which splits the analysis into a first stage analysing the global and regional temperatures, and a second stage in which the results are fed into the GEV estimation. It's not clear to me why the analysis should be split in this way: surely the Bayesian computational machinery allows you to do everything in one step? A clear justification for the two-step approach is needed.

- First, the two steps are not treated in the same way at all. Global forcings can be constrained analytically (52 parameters, using the Gaussian conditioning theorem), whereas the five parameters of the GEV law require MCMC. Mixing the constants would require using only an MCMC approach to constrain all of the parameters, i.e. in 57 dimensions, which is much more complex to implement.
- Next, ANKIALE idea, which was not sufficiently emphasised, is to offer a set of extensible tools. In this context, certain statistical models do not have a constant that can be unified. For example, the following model is used for precipitation  $P_t$  in attribution studies (see the work of, e.g., van der Wiel et al., 2017; van Oldenborgh et al., 2017; Uhe et al., 2018; Tradowsky et al., 2023):

$$\begin{cases} P_t \sim \text{GEV}(\mu_t, \sigma_t, \xi_t), \\ \mu_t = \mu_0 \exp(\alpha/\mu_0 X_t), \\ \sigma_t = \sigma_0 \exp(\alpha/\mu_0 X_t), \\ \xi_t \equiv \xi_0. \end{cases}$$
 (1)

This model is not present in ANKIALE, but it is offered in SDFC (on which ANKIALE relies for inference) and can easily be added.

• More fundamentally perhaps: I am aware of the fashion for regressing on global and regional mean temperatures in the attribution literature. I am also aware that this literature tends to focus on partitioning variation into natural and anthropogenicallyforced components. If I understand correctly however, the partitioning here is derived (lines 140-147) from a regression on either the outputs of an energy balance model (EBM), or the net radiative forcings. It's not clear which option has been used to obtain the presented results (or, if it was an EBM, which one was used). However, the results in (for example) Figures S1 and S2 show a strong resemblance between the "naturally-attributed" component of global mean surface temperature and the natural component of the forcings (not shown in the paper, but I'm familiar with the forcing time series). I guess that the correlation between this component and the input forcings is at least 0.95, and possibly greater than 0.99 — indeed, it will be 1 if the forcings themselves have been used in the estimation procedure. Moreover, the estimated anthropogenic components of the global and regional temperatures will be smooth curves, just like the anthropogenic component of the forcings (or of the EBM outputs). I will therefore bet a moderate sum of money that one can obtain almost identical results to those reported in the paper by ignoring the global and regional mean temperatures altogether, and instead using the natural and anthropogenic components of the net radiative forcing in the models for the GEV parameters. This would simplify the analysis both conceptually and computationally (e.g. it would eliminate the need for GAMs).

It is perfectly true that the correlation between the natural part of the forcings and the counterfactual signal is close to 1. The aim here is to determine the response to the forcings, constrained by observations. One of the advantages is that it allows us to attribute the corresponding regional warming, as well as the warming of extremes, to a level of global warming, elements that are absent when other forcings, such as the net radiative forcing, are used directly.

We nevertheless attempted to replace the GAM decomposition with EBMs (Leach et al., 2021; Smith et al., 2024). In the test performed, we have tested with EBMs instead of forcings because there is no reason why global temperature should directly follow forcings, whether natural or anthropogenic. The results were unsatisfactory: for reasons that remain unknown, EBMs are unable to reproduce the climate change seen in climate models.

The attribution community may consider this suggestion to be very oversimplistic. Nonetheless, I would be interested to see how it performs. With the ANKIALE software as described, this should not be hard to implement: just replace the estimates of  $X_t^{*,N}$  and  $X_t^{*,A}$  with the corresponding components of the net radiative forcing, and see what happens.

This approach would take us away from the subject of the article: to present improvements to an existing method, a tool allowing its use with an example application. We have instead proposed it as a possibility in the updated perspectives section.

7. The split of figures and tables between the main paper and supplement seems strange. For example, Figures S1–S3 are helpful in understanding the process and the results, but they have been relegated to the supplement. On the other hand, Tables 2 and 3 in the main paper take up a lot of space and will be of interest to relatively few potential readers: these could reasonably be moved to the supplement.

These figures were revised along with the methodology section and moved back into the main text. The tables were moved to the supplement.

**1.3 Specific / detailed comments and queries**

**Line 1** "estimating the statistics of temperature extremes": as noted above, it isn't clear from this what you're actually doing.

We have reworded.

Line 29 typo "compbining".

Thanks, corrected.

Line 30 a Bayesian framework?

Modified.

**Lines 33–34** I would remove the sentence "However...inconsistencies". It is redundant given the content of the subsequent paragraph, and only one of the issues is actually an inconsistency.

We have reworded.

**Line 47** the time taken to "process the entire domain" presumably depends on the size of the domain! To put this in context, it would be helpful to indicate the approximate number of grid cells being referred to here.

More generally: if there are any real applications in which it's necessary to carry out an analysis over a domain of this size, then a time scale of up to a week is perhaps still not fast enough. It is certainly too slow for the kinds of real-time attribution studies carried out by World Weather Attribution as described in lines 18–20, given that the time available for data analysis in such studies is only a fraction of the total time available. For those kinds of applications though, the focus is typically on specific events in much smaller regions. It's certainly helpful to give an indication of computational cost; but it would be more relevant to indicate the cost for realistic applications of the methodology. Alternatively, give (e.g.) an indicative cost per thousand grid cells, so that readers can figure out for themselves the likely cost for their own applications.

We have rephrased this to specify that the entire procedure takes approximately 2.5 hours per grid point.

**Line 51** what's the relevance of the number of countries and their partial inclusion? What does "exact ratio" refer to?

This is simply information about the domain, and knowing what percentage of a country is included in it allows us to know to what extent the value presented here represents the country or not.

**Line 64** "refer to" => "referred to".

Thanks, corrected.

**Lines 73–74** "we use observations from a weather station...: But you just said you were using ERA5. Perhaps the station data are used to illustrate the methodology and then, as a separate study, the ERA5 data are used to examine the entire European area. If this is the case, probably it would be helpful for the reader if the paper is reorganised clearly along these lines.

We now only use ERA5 throughout in order to simplify the presentation.

**Lines 75–76** how does the use of a longer dataset allow you to "better verify the contribution"? Given more data, any reasonable method is going to do better. And, if you need an unusually long dataset to identify the benefits of the new approach, then the implication is presumably that the benefits aren't obvious for datasets of more realistic size! (I'm playing devil's advocate, but you need to think more carefully about what you're trying to say).

You are right, and since there is a risk of consistency issues in the station series, we prefer to use only ERA5 in the revised version.

**Lines** 77–78 as noted above, I believe it is fairly common to use global or regional mean temperatures in attribution studies. Nonetheless, it would be helpful to clarify this — perhaps

as part of an introductory summary of the basic steps / logic of the approach (see general point 2 above).

Done.

**Line 85** do the emissions scenarios really cover the historical period? I think the intended meaning is probably that historical estimates of emissions were used for the period 1850–2014, while scenarios based on alternative SSPs were used for the period 2015–2100.

Of course, we have reworded it.

**Lines 89–90** is there a reference to support the claim that the current warming trajectory is close to the SSP2-4.5 scenario? Or is IPCC (2023) intended to be this reference? As written, it seems to be a reference just for the 2.8K estimate.

Sentence deleted.

**Line 92 (also line 137)** the GCM simulations of TX3x are broadly comparable with the ERA5 estimates because the latter are gridded. However, they are not comparable with the Paris station- based observations. How is this handled?

In itself, this is not a problem because the models are only used as priors. The question no longer arises since the station has been removed.

While on the subject of gridded data: each GCM has its own grid, and many of them are different from the grid used in the ERA5 dataset. How was this dealt with?

The parameters for each GCM are inferred on their own grids. It is during multi-model synthesis that the GCMs are regridded to the ERA5 grid using nearest neighbour interpolation. A fairly similar alternative approach was tested: nearest neighbour interpolation of the covariance matrices, and bilinear interpolation for the mean to smooth the effect of the GCMs' grid cells. The contribution was not very interesting. We have now added this information to the revised text.

**Lines 86–97** see general comment 1 above.

See the response to the comment 1.

**Line 104** this is where the paper starts to become hard to follow. There is no statement of what the covariate  $X_t^{R,F}$  actually is: all we're told is that it's a proxy for something else. And how do the quantities in the final line of equation (1) relate to the regional covariate  $X_t^R$  mentioned immediately beforehand? And why are these quantities needed, given that the GEV parameters are modelled using the original quantity  $X_t^R$ ?

The answers to some of these questions emerge implicitly later in the paper. If I understand correctly, equation (1) and the subsequent paragraph do not correctly describe what was done. My best guess is that  $\mu$  is represented

$$\mu_0 + \sum_{j \in \{G,R\}} \left( \mu_{N,j} x^{N,j} + \mu_{A,j}^{A,j} \right)$$

with a corresponding expression for  $\log \sigma$ , where  $x^{N,j}$  and  $x^{A,j}$  represent respectively the 'natural' and 'anthropogenic' components of variation in the global or regional temperature series. Setting the  $\{\mu_{A,j}\}$  and  $\{\sigma_{A,j}\}$  to zero then allows an estimate of the GEV parameters in the absence of anthropogenic influence. If this *is* correct, then the presentation in the manuscript seems very over-complicated.

There is a misunderstanding of the term "add", which did not imply an addition between covariates, but rather an addition of the covariate to the statistical model. This entire section has been rewritten and we do hope that the new version is now much clearer.

Line 105 typo "anthropogenice".

Thanks, corrected.

**Lines 106–107** "We also add..." this confirms that equation (1) does not describe what was done. See above.

By "add", we meant this equation:

$$\begin{cases} T_t \sim \text{GEV}(\mu_t, \sigma_t, \xi_t) \\ \mu_t := \mu_0 + \mu_1 X_t^{\text{R,F}} \\ \log(\sigma_t) := \sigma_0 + \sigma_1 X_t^{\text{R,F}} \\ \xi_t \equiv \xi_0 \\ X_t^{\text{R,F}} := X^{\text{R,0}} + X_t^{\text{R,N}} + X_t^{\text{R,A}} \\ X_t^{\text{G,F}} := X^{\text{G,0}} + X_t^{\text{G,N}} + X_t^{\text{G,A}} \end{cases}$$

It is now directly integrated into the text.

**Lines 109–10** it is not correct that the model can be seen as a linear model. It's not linear, and linear models have additive noise terms. Suggest deleting this sentence.

Sorry, it is a \*generalized\* linear model, from the exponential family. Corrected.

**Line 112** "construct" => "estimate"? Also, the 'factual / counterfactual' vocabulary seems unnec- essary and confusing. I know that it comes from the literature on causal inference, but the present paper doesn't really deal with causality. If my summary in point 15 above is correct, then none of it is really needed — and other explanations can be simplified as well.

The "counterfactual" vocabulary is widely used in application examples (e.g. in intensity changes). The section has been rewritten, but we retain this formulation.

**Line 116** it's not clear to me that all elements of  $\theta$  should be described as 'parameters'. The Xs would be more precisely described as latent or hidden variables. Arguably this is a minor point, but it does affect the readability of the paper.

The section has been entirely rewritten for clarity.

**Line 128** what does  $\kappa$  represent? See general point 2 above.

This is the multi-model synthesis. The section has been rewritten.

**Lines 131–132** this repeats lines 96–97 (but see general comment 1).

Thanks, modified.

**Line 143** 'Energy Balance Model' appears twice. And which EBM are you using?

For CMIP5 model, we use the EBM model defined by Held et al. (2010) and studied in CMIP5 by Geoffroy et al. (2013). In the new version, this is specified in the mathematical details of the supplementary material, Sect. S.1.1.

Line 145 residuals from what? (actually I don't think they are residuals).

That is indeed poorly worded. We wanted to express that it was the spline smoothing of the regional temperature without natural forcings. The section has been rewritten.

**Line 147** how is the variance of  $\varepsilon^*$  represented? Is it the same for all SSPs?

Here,  $\varepsilon^*$  appears as a multivariate normal distribution among all the smoothing hyperparameters. Since the variance may differ from one hyperparameter to another on the spline basis, it is therefore not necessarily the same for each scenario. All of this is explained in more details in Section S.1.1 of the supplementary material..

Lines 148–151 This presentation is needlessly complicated. If GAMs are needed at all (see general point 6), the model could be fitted directly using the gam() function in the mgcv package in R, e.g. using the by() argument to fit different smooths for the SSPs while retaining the common natural forcing term. The required setup is actually quite straightforward: it may even be available somewhere in python, although I appreciate that there's nothing in python that gets close to the capabilities of mgcv. We probably don't need to know the details of backfitting either: this is a completely standard approach to fitting GAMs.

The problem with this model using GAMs is that the natural term is shared, and there is no package that can take this feature into account when writing the model (or we have not found a way to do so). We have therefore explained how inference is performed.

**Line 153** if it is necessary to retain the details on how the GAMs are fitted, then the expression in line 154 is a sum rather than an average as described: either the expression or the description is wrong. And in line 154, I assume both terms are summed: they should be enclosed in parentheses to make this clear, therefore.

Indeed, we forgot to normalise by the number of scenarios. This has been corrected.

**Line 156** as noted above, it would be helpful to move Figure S2 into the main paper. Also however, there is no indication as to whether this was obtained using radiative forcings or EBM outputs.

This is a CMIP6 model, so we have used radiative forcings defined by Smith (2020). In the new version, this is specified in the mathematical details of the supplementary material, Sect. S.1.1.

**Line 157** "We can see that..." how can we see that? What are we supposed to be looking at? How many model realisations are there? If more than one, this hasn't been mentioned in the text (or how you have dealt with it) — it is noted in the caption to Table 1, but if affects the interpretation of results then it needs to be discussed in the text as well.

The section has been rewritten.

Line 158 "peaks are due to the volcanoes" they are troughs I think, not peaks.

Thanks, corrected.

**Line 159** when referring to the "constant" signal, do you mean constant over time or constant across SSPs? If over time: why is this relevant? If over SSPs: why do the volcanic and solar effects differ between them as suggested at the end of the line?

It was constant over time; we simply noted that we had the expected result: a constant counterfactual world, apart from volcanoes and solar cycles.

**Line 161** "we propose the following method..." I have no idea what is being done here, or why.

The section has been rewritten. This specific part is detailed in App. B1.

Line 162 "resampled" what does this mean? Resampled from what?

The section has been rewritten. This specific part is detailed in App. B1.

**Line 164** what "method described above" is being referred to here?

The section has been rewritten. This specific part is detailed in App. B1 and Sect. S.1.1.

**Line 165** "the four smoothed covariates" what are these? I thought the previous step (described in lines 138–155) was designed to estimate these covariates.

The section has been rewritten. This specific part is detailed in App. B1.

**Lines 184–188** it is not correct to describe the formulation here in terms of models that are "statistically indistinguishable from reality", which implies that reality and the models are exchangeable. The representation set out here is sometimes called the 'truth plus error' representation, I think.

Indeed, there is a problem with our notations, and probably an error in what we wrote in the initial submission. Here is the approach we take, which is the one used by ANKIALE. The prior is constructed as a synthesis of different climate models, using the following hypothesis: "the models are statistically indistinguishable from truth", developed by Ribes et al. (2017). Let:

- $\theta_* \sim \mathcal{N}(\hat{\theta}_*, \Sigma_{\hat{\theta}_*})$  be the desired multi-model synthesis
- $\theta_{\mathcal{M}}$  the mean response of an infinite set of models, and  $\Sigma_{\mathcal{M}}$  the climate modelling uncertainty (assumed to be equal for each model);
- $\check{\theta}$  the "truth" (not assumed to be equal to the multi-model mean).

The indistinguishability hypothesis is equivalent to saying that  $\theta_m$ ,  $\theta_*$  and  $\check{\theta}$  all come from the same distribution, i.e.  $\theta_m$ ,  $\theta_*$ ,  $\check{\theta} \sim \mathcal{N}(\theta_{\mathcal{M}}, \Sigma_{\mathcal{M}})$ . Note the difference with the "truth plus error" hypothesis, where the  $\theta_m$  are centred on reality, i.e.  $\theta_m \sim \mathcal{N}(\check{\theta}, \Sigma_{\mathcal{M}})$ . In the new version of our document, we have clarified the notation, and all the mathematical details are given in the supplementary material, Sect. S.1.2.

**Lines 202–203** I don't know what 'weights' are referred to here, but this sentence suggests that outlying GCMs are excluded from the analysis. No clear justification is given for doing this but, if what's written is correct, it is very worrying because it suggests that the analysis uses only those models that agree with each other. This is very poor scientific practice, and will also lead to underestimation of uncertainty in the final results.

We simply noted that this model appears almost like an outlier with respect to the multimodel synthesis, which is why it is not widely included in the synthesis (note, however, that the intersection is not empty, just improbable). We also specified which model exhibits this behavior when rewriting the text.

**Line 207** 'Recall that we have...' when I read the paper, I didn't recognise that this information had been provided previously. All of the relevant information needs to be provided, clearly and unambiguously, at the outset.

We recall the notations of the observations, described in Sect. 2, where the data are grouped together to make reading easier.

**Line 208** "Our goal is to estimate the distribution..." really? I was unaware of this on reading the paper. And there has still been no clear statement of what *kappa* actually is, or why we might want to learn about it.

The section has been rewritten.

**Line 210** the 'double conditioning' notation in this equation does not exist in conventional practice, as far as I'm aware. I think the denominator is incorrect, as well: what's being done here is Bayes' theorem conditional on  $X_t^{*,o}$  which should lead to

$$\mathbb{P}(\kappa|T_t^o,X_t^{*,o}) = \frac{\mathbb{P}(T_t^o|\kappa,X_t^{*,o})\mathbb{P}(\kappa|X_t^{*,o})}{\mathbb{P}(T_t^o|X_t^{*,o})}$$

I suspect there is also an unstated assumption that  $X_t^{*,o}$  is irrelevant for  $T_t^o$  once  $\kappa$  is known, so that  $\mathbb{P}(T_t^o|\kappa, X_t^{*,o}) = \mathbb{P}(T_t^o|\kappa)$ . Apart from the denominator, this then gives equation (5) in the paper.

By being more rigorous about the notation, your equation appears to be:

$$\mathbb{P}(\kappa | T_t^o, X_t^{*,o}) = \mathbb{P}[(\kappa | \{T_t = T_t^o\}) | \{X_t = X_t^{*,o}\}]$$

Whereas ours is:

$$\mathbb{P}[\kappa | (\{T_t = T_t^o\} \cap \{X_t = X_t^{*,o}\})]$$

This explains the differences between your calculation and ours.

**Lines 211–212** "In other words..." this makes very heavy weather of quite basic material, while at the same time failing to provide enough information as to what's being done and why.

This is the translation of Eq. (5). It may be clearer with the correct equation.

Line 212 "are constrained" by what? Why?

See comment below.

**Line 216** why do you want "the posterior of global and regional temperatures"?

We think that at this stage, important information has been missed. The section has been rewritten and we think that it should now contain all relevant information.

**Line 218** in this equation, the left-hand side depends on *t* but the right-hand side doesn't. I have no idea what is intended. What are the dimensions of the subsequent matrices? What are you doing?

The exact form of this matrices is now given in the Supplementaty material, Sect. S.1.3.1. The time axis was in matrix A, which contains the spline basis.

Line 224 again, what are the dimensions of these quantities?

See answer above.

**Line 227**  $N^{SSP}$  isn't defined.

We apologize for this error, the notations have been revised.

Line 228 what do you mean by 'null values'? If you mean zeroes, say so.

It was just zero, so we put zeros.

**Lines 234–235** what do you mean by "scenarios" and "observations" here? Are the "scenario" data from the climate models? If so, what's the basis for this assumption? After all, the available scenarios (assuming you mean the SSPs that are included in the analysis) are arbitrary: the observations can't be the average over all collections of scenarios.

If the scenario data here are not from the climate models, where are they from?

Scenarios here come from the multi-model synthesis (the prior), conditioned by a SSP. The observational constraints using the Gaussian conditioning theorem consist of writing the observations as a transformation of the prior by a projection matrix, plus an observed error term. Here, as we are dealing with several scenarios simultaneously, we have a particular configuration: we may have several scenarios over the observed period. We simply propose to take the average of the scenarios over this period, which we justify by the fact that the scenarios are almost the same over the period 2014-2025. We have rewritten the text to clarify our approach.

**Line 237** how is it 'more general' to take the average of the scenarios? All you're doing is to making one choice instead of another.

We now clearly present the two approaches with their advantages and disadvantages, and explain why we chose one over the other. In a nutshell:

- One approach where the average of the scenarios can be associated with the observations. This is justified by the fact that the climate scenarios are sufficiently similar over the period 2015–2024.
- One approach where we choose a scenario, which is therefore slightly less general because it assumes expertise about the future that we are borrowing.

Line 240 what does 'this constraint' refer to?

The constraint of  $\kappa$  by  $X_t^{*,o}$ .

**Line 241** 'the covariate Europe'?! At this point, I'm starting to wonder whether all of the authors checked the manuscript before it was submitted.

It is certainly a French idiom that apparently does not translate into English. The section has been rewritten.

**Line 263** use of the NUTS is sensible, but there's no need to provide details of, e.g. the generation of proposal distributions or anything else in the paragraph. The explanation provided isn't enough for readers who are not unfamiliar with the algorithm, and they don't need to understand how NUTS works in any case. All a reader needs to know is that this is a a more modern MCMC method that can often overcome the difficulties described in the previous paragraph.

The section has been rewritten, and this information has been moved to App. B.3.2.

**Line 280** what are "the laws"?

Laws, distributions, probability distributions. It is true that we have used these words as synonyms. We have rephrased.

**Line 281** again, 'recall' implies that the reader has been told already.

The equations describing GEV have been moved to Appendix B.1, which we can refer to for the readers information.

**Line 284** what's an 'intensity'? Don't you just have a value of *T* here?

Yes, that is the language used in attribution. We have rephrased it.

**Line 291** I don't think 'deduce' is the right word here. Rather, you can construct measures of the change.

Modified.

**Line 295** Figure S4 is another that belongs in the main paper.

The section has been rewritten.

**Lines 297** 'we calculate...' why? \*NB\* the answer to this may be obvious to someone who understands what the authors are trying to do, but I lost track of that long ago.

The section has been rewritten.

**Lines 301–303** why are you calculating the Wasserstein distance? Few readers will have a good intuition for what a given Wasserstein distance looks like, in terms of the distributions being compared. And most will be interested primarily in whether the distributions differ to an extent that would change any substantive conclusions of interest.

The section has been rewritten.

**Line 312** 'slightly ahead' in what sense?

The section has been rewritten.

**Line 320** is the 'simultaneous analysis of several thousand grid points' often required? Given that you're just doing one grid point at a time (I think), attribution studies would usually focus on specific events that affect a subset of the grid points. More importantly though: in these kinds of applications, the focus is usually on weather that is simultaneously extreme over a large area. As far as I can see, the present paper doesn't address that problem: some discussion of this is needed, as part of the context for the work.

A discussion has been added.

**Line 326** I wonder whether this kind of "package manual" material is appropriate for inclusion in a journal article? This query is perhaps for the journal editor.

This article focuses as much on methodology as it does on tools, which is why we chose GMD. We therefore believe that it has its place here.

Line 377 what does 'since 1940' mean in 'the maximum observed in 2024 since 1940'?

This is the maximal temperature observed between 1940 and 2024. We have reformulated.

Line 389 typo 'conter-factual'.

Thanks, corrected.

**Line 451** what would be the challenges in extending the methodology to other variables such as wind and precipitation? It seems to me that it would be basically the same: it's all just based on the GEV distribution for annual maxima.

To our knowledge, no one has yet performed a multi-model synthesis followed by constraint with observations on variables other than temperature. In our previous paper (Robin and Ribes, 2020), the entire Sect. 3.3 sought to verify the best compromise (in terms of statistical model choice) in GCMs, and to verify the quality of the final fit to observations. This procedure would obviously need to be repeated when dealing with new variables. For information, the statistical model of Eq. 1, often used for precipitation, assumes a strong relationship between  $\mu$  and  $\sigma$  (if one increases, so does the other), but we have been able to verify that this assumption is generally false in GCMs. Extending it to other variables is therefore a slightly more complex task than your proposal.

**Caption to Table 2** what are these values averaged over? Grid cells, I assume — it would be helpful to clarify this. See also the earlier suggestion for moving this table (and Table 3) to the supplement.

Yes, the values are the average for the area, and the tables are in the supplement. It is now clarified.

**Algorithm 1** this is a completely standard algorithm, and there is no need to include it (but if you do include it, the cross-references at the end are broken — similarly for Algorithm 2 on the next page).

It is certainly standard, but not necessarily known by the climate community. After rewriting the equations, we no longer needed it and removed it. The cross-references were indeed broken, which was corrected by updating the Copernicus template.

**Appendix A** I don't find Appendix A very helpful. The material on GAMs in Appendix A1 is mostly fairly standard, but seems more complicated than necessary; while I don't understand Appendix A2 because I still don't know what the purpose is.

All of this has been completely reworded in sections S.1.1. and S.1.3.1.

Finally: out of interest, before submitting my report I looked at the comments from the other reviewers. There is quite a lot of overlap with my comments above, particularly from Anonymous Reviewer 1 who also found the paper hard to follow. To be completely clear therefore: in this report, everything except this paragraph was written independently and without looking at the other review reports.

Thank you for your valuable comments and your honesty. As repeated along our responses, our article has been deeply modified (especially the method section) to clarify our approach and goals.

**Bibliography**

- Geoffroy, O., D. Saint-Martin, G. Bellon, A. Voldoire, D. J. L. Olivié, and S. Tytéca (Mar. 2013). "Transient Climate Response in a Two-Layer Energy-Balance Model. Part II: Representation of the Efficacy of Deep-Ocean Heat Uptake and Validation for CMIP5 AOGCMs". In: J. Clim. 26.6, pp. 1859–1876. ISSN: 0894-8755, 1520-0442. DOI: 10.1175/JCLI-D-12-00196.1.
- Held, I. M., M. Winton, K. Takahashi, T. Delworth, F. Zeng, and G. K. Vallis (May 2010). "Probing the Fast and Slow Components of Global Warming by Returning Abruptly to Preindustrial Forcing". In: J. Clim. 23.9, pp. 2418–2427. ISSN: 0894-8755, 1520-0442. DOI: 10. 1175 / 2009JCLI3466.1.
- Leach, N. J., S. Jenkins, Z. Nicholls, C. J. Smith, J. Lynch, M. Cain, T. Walsh, B. Wu, J. Tsutsui, and M. R. Allen (May 2021). "FaIRv2.0.0: A Generalized Impulse Response Model for Climate Uncertainty and Future Scenario Exploration". In: *Geosci. Model Dev.* 14.5, pp. 3007–3036. ISSN: 1991-959X. DOI: 10.5194/gmd-14-3007-2021.
- Ribes, A., F. W. Zwiers, J.-M. Azaïs, and P. Naveau (2017). "A New Statistical Approach to Climate Change Detection and Attribution". In: *Clim Dyn* 48.1, pp. 367–386. ISSN: 1432-0894. DOI: 10. 1007/s00382-016-3079-6.
- Robin, Y. and A. Ribes (2020). "Nonstationary Extreme Value Analysis for Event Attribution Combining Climate Models and Observations". In: *Adv. Stat. Clim. Meteorol. Oceanogr.* 6.2, pp. 205–221. ISSN: 2364-3579. DOI: 10.5194/ascmo-6-205-2020.
- Rougier, J., M. Goldstein, and L. House (Sept. 2013). "Second-Order Exchangeability Analysis for Multimodel Ensembles". In: *J Am Stat Assoc* 108.503, pp. 852–863. ISSN: 0162-1459. DOI: 10.1080/01621459.2013.802963.
- Smith, C. (Aug. 2020). Effective Radiative Forcing Time Series from the Shared Socioeconomic Pathways. Zenodo. DOI: 10.5281/zenodo. 3973015.
- Smith, C., D. P. Cummins, H.-B. Fredriksen, Z. Nicholls, M. Meinshausen, M. Allen, S. Jenkins,

- N. Leach, C. Mathison, and A.-I. Partanen (Dec. 2024). "Fair-Calibrate v1.4.1: Calibration, Constraining, and Validation of the FaIR Simple Climate Model for Reliable Future Climate Projections". In: *Geosci. Model Dev.* 17.23, pp. 8569–8592. ISSN: 1991-959X. DOI: 10.5194/gmd-17-8569-2024.
- Tradowsky, J. S., S. Y. Philip, F. Kreienkamp, S. F. Kew, P. Lorenz, J. Arrighi, T. Bettmann, S. Caluwaerts, S. C. Chan, L. De Cruz, H. de Vries, N. Demuth, A. Ferrone, E. M. Fischer, H. J. Fowler, K. Goergen, D. Heinrich, Y. Henrichs, F. Kaspar, G. Lenderink, E. Nilson, F. E. L. Otto, F. Ragone, S. I. Seneviratne, R. K. Singh, A. Skålevåg, P. Termonia, L. Thalheimer, M. van Aalst, J. Van den Bergh, H. Van de Vyver, S. Vannitsem, G. J. van Oldenborgh, B. Van Schaeybroeck, R. Vautard, D. Vonk, and N. Wanders (June 2023). "Attribution of the Heavy Rainfall Events Leading to Severe Flooding in Western Europe during July 2021". In: Clim Change 176.7, p. 90. ISSN: 1573-1480. DOI: 10.1007/ s10584-023-03502-7.
- Uhe, P., S. Philip, S. Kew, K. Shah, J. Kimutai, E. Mwangi, G. J. van Oldenborgh, R. Singh, J. Arrighi, E. Jjemba, H. Cullen, and F. Otto (2018). "Attributing Drivers of the 2016 Kenyan Drought". In: *Int. J. Climatol.* 38.S1, e554–e568. ISSN: 1097-0088. DOI: 10.1002/joc.5389.
- van der Wiel, K., S. B. Kapnick, G. J. van Oldenborgh, K. Whan, S. Philip, G. A. Vecchi, R. K. Singh, J. Arrighi, and H. Cullen (Feb. 2017). "Rapid Attribution of the August 2016 Flood-Inducing Extreme Precipitation in South Louisiana to Climate Change". In: *Hydrol. Earth Syst. Sci.* 21.2, pp. 897–921. ISSN: 1027-5606. DOI: 10.5194/hess-21-897-2017.
- van Oldenborgh, G. J., K. van der Wiel, A. Sebastian, R. Singh, J. Arrighi, F. Otto, K. Haustein, S. Li, G. Vecchi, and H. Cullen (Dec. 2017). "Attribution of Extreme Rainfall from Hurricane Harvey, August 2017". In: *Environ. Res. Lett.* 12.12, p. 124009. ISSN: 1748-9326. DOI: 10 . 1088 / 1748-9326/aa9ef2.

---

## Author Comment (AC4)

**Response to Reviewer 4 comments about the article "A Bayesian Statistical Method to Estimate the Climatology of Extreme Temperature under Multiple Scenarios: the ANKIALE Package"**

ROBIN, Y., VRAC, M., RIBES, A., BARBAUX, O. and NAVEAU, P.

October 23, 2025

**Note** In this document, the text in regular format corresponds to the reviewers questions. The answers from authors are given in the grey blocks.

**1 Reviewer 4 (Anonymous)**

**1.1 Intro**

This paper extends previously published methods to estimate the changing likelihood from the past to the future of a specified extremes metric due to historical forcings and future emission scenarios. In addition the likelihoods from a counterfactual world, one without anthropogenic emissions is estimated. The new extension to the methodology is enabling a single counterfactual world to be estimated from multiple different scenario runs. The authors also revise the Bayesian approach to reflect advances in this field. For the analysis the authors have developed a software package that is publicly available to facilitate other researchers wishing to perform similar tasks.

We would like to thank Reviewer 4 for this summary and for recognizing our efforts to make a Bayesian-based statistical analysis method for extreme events accessible to as many people as possible.

My request for major revision is not due to significant concerns of the analysis they have performed but because of the lack of placing their work in the context of the field more generally and the belief that in its present form the text is only accessible to a very small audience. I found what the authors have exactly done and why quite difficult to ascertain and feel that I have only got the general gist rather than a good understanding.

We are sorry that you have not gained a good understanding of the method. We have deeply reworked on the text (particularly Section 3) with this in mind, and including the suggested references in particular.

**1.2 Context of other work**

There is now quite a body of work addressing the production of posteriors of climate variables from the combination of observations with climate models more generally and extreme

specific. It would be helpful if they placed their work in the context of these other pieces of work.

A discussion has been added to provide much more context to the present work.

**1.2.1 General Bayesian climate projection**

- A. Ribes et al. (Jan. 2021). "Making Climate Projections Conditional on Historical Observations". In: *Sci. Adv.* 7.4, eabc0671. DOI: 10.1126/sciadv.abc0671)
- C. J. Smith et al. (2021). "Energy Budget Constraints on the Time History of Aerosol Forcing and Climate Sensitivity". In: *J. Geophys. Res. Atmos.* 126.13, e2020JD033622. ISSN: 2169-8996. DOI: 10.1029/2020JD033622
- G. C. Hegerl et al. (June 2021). "Toward Consistent Observational Constraints in Climate Predictions and Projections". In: *Front. Clim.* 3. ISSN: 2624-9553. DOI: 10.3389/fclim. 2021.678109
- L. Brunner et al. (Oct. 2020). "Comparing Methods to Constrain Future European Climate Projections Using a Consistent Framework". In: *J. Clim.* 33.20, pp. 8671–8692. ISSN: 0894-8755, 1520-0442. DOI: 10.1175/JCLI-D-19-0953.1
- L. Brunner et al. (Nov. 2019). "Quantifying Uncertainty in European Climate Projections Using Combined Performance-Independence Weighting". In: *Environ. Res. Lett.* 14.12, p. 124010. ISSN: 1748-9326. DOI: 10.1088/1748-9326/ab492f
- J. M. Murphy et al. (2018). *UKCP18 Land Projections: Science Report*. Tech. rep. Met Office Hadley Centre
- R. Knutti et al. (2017). "A Climate Model Projection Weighting Scheme Accounting for Performance and Interdependence". In: *Geophys. Res. Lett.* 44.4, pp. 1909–1918. ISSN: 1944-8007. DOI: 10.1002/2016GL072012
- B. M. Sanderson et al. (July 2015). "A Representative Democracy to Reduce Interdependency in a Multimodel Ensemble". In: J. Clim. 28.13, pp. 5171–5194. ISSN: 0894-8755, 1520-0442. DOI: 10.1175/JCLI-D-14-00362.1
- G. R. Harris et al. (June 2013). "Probabilistic Projections of Transient Climate Change". In: *Clim Dyn* 40.11, pp. 2937–2972. ISSN: 1432-0894. DOI: 10.1007/s00382-012-1647-y

**1.2.2 Extreme specific probabilistic projection**

- J. M. Murphy et al. (2020). *UKCP Additional Land Products: Probabilistic Projections of Climate Extremes*. Tech. rep. Met Office Hadley Centre
- S. J. Brown et al. (Nov. 2014). "Climate Projections of Future Extreme Events Accounting for Modelling Uncertainties and Historical Simulation Biases". In: *Clim Dyn* 43.9, pp. 2681–2705. ISSN: 1432-0894. DOI: 10.1007/s00382-014-2080-1

**1.3 Accessibility**

I do not underestimate how difficult this challenge is and it is difficult to know how to advise on this. Apart from the hurdle of the Bayesian terminology I think the reader will struggle to put all the pieces together. This could be alleviated with some text at the beginning of the Methods section giving an outline of the whole procedure, how the different parts fit together, the assumptions of the approach, how climate model deficiencies are accounted for and how biases with observations are dealt with.

For those readers who will probably never be that comfortable with the more statistical aspects of the paper I think they will be greatly helped if there was more emphasis and care in the physical interpretation of the method and discussion of the example analysis. Also, a comparison with other posterior work who's emphasis is more towards the physical modelling uncertainty of future climate For example such a discussion would include the impact of this being an ensemble of opportunity, how carbon cycle uncertainty and aerosol modelling uncertainty is samples and their consequences for the results presented. Also a discussion on the relative importance of  $X_R$  and  $X_G$ , what different do they bring to the analysis

Thanks for your interesting suggestions. The entire Section 3 has been entirely rewritten in this regard in order to make it much more accessible.

**1.4 Other general comments**

1. I know section 5 is a really important part of the paper but it really does interrupt the flow. Might the authors consider moving it to an appendix?

Part of the message of this paper is precisely to promote this tool (and that is one of the reasons why we chose GMD), so we do not want to move it to the appendix. However, we have reworked it to make it easier to read.

2. Are return periods and their changes particularly helpful due to their extremely nonlinear behaviour? The issue with looking at return periods in the present day is that the return period will be dominated by uncertainty in the shape. The actual increase in extreme temperatures, however, will be dominated by the change in location. Present day factual and counterfactual comparisons if looking at return periods will be dominated by shape uncertainty whereas present to future comparisons of the 100y return level will be dominated by location change and its uncertainty.

These crucial questions go beyond the scope of this article, which focuses on presenting a tool for estimating return periods based on the GEV model commonly used in this context. They have been be mentioned in the new conclusion, and it would be particularly interesting to compare this with statistical models that strongly constrain the shape through the upper bound (Noyelle et al., 2025).

3. The adaptation sphere is very focussed on high resolution modelling of the future climate (perhaps too much). As this paper is also concerned with determining the likelihoods of future extremes it would be useful if the authors could comment on how their method might accommodate regional climate modelling.

This has been discussed in the new conclusion, with the idea that regional models can be integrated in the same way as GCMs.

4.  $X^R$  and  $X^G$  - these will be highly correlated. Can you demonstrate that both are required? If you only had one will not the parameters not just get adjusted to compensate? Or put it another way, does the small bit of extra info when using both lead to a significantly better outcome?

Indeed, especially for the attribution of current events, using  $X^G$  or  $X^R$  as a covariate will not change the result much (which is what the WWA typically does). Now, looking ahead to 2100, even though both series start at  $\sim 0$  in 1850 (in terms of anomaly), the date on which the trend emerges is not the same (due to aerosols) and the end point is not the same (it is warmer in Europe than globally). We believe that these differences would be significant for local temperatures, which depend more on regional than global temperatures. The idea here of keeping both covariates is to be able to address the dependence between global, regional and local warming. This makes it possible, for example, to examine regional and local warming if global warming is fixed at a certain value.

**1.5 Minor comments**

**1.5.1 Abstract and Intro**

**Abstract general**

- could be improved to better describe the solution it is providing in more general terms (currently only 81 words)
- does not mention the improved Bayesian sampling mentioned in the introduction. This seems a key point of the paper.
- Line 3 I think the abstract needs to mention that your "observations" are ERA5
- Line 5 "Counterfactual world" this is jumping in very deep very quickly into DA jargon

For the last three comments, we have reformulated the abstract.

**Line 29 complining**

Done.

**Lines 24-28** I would be helpful to have some non DA focussed literature on extremes in climate

**Lines 29-34** very limited review of other literature attempting Bayesian approaches to present/future climate, see above

For the last two comments, we have added references.

- **Lines 34-35** Flow of text is a bit confusing. It is not clear the next paragraph is addressing these two issues because of the way it starts. The "Here," suggest the paragraph is setting off on a new topic.
- **Lines 46-47** Could you please compare like with like. Currently it is CPU time vs wall time. Surely wall time is primarily dependent on how much compute resource you have at your disposal.
- Line 53 "observed" not true observations should be something like "as represented in ERA5"
- **Lines 59,60** "classical attribution... specific definition" Not sure this will make sense to the reader

For the last four comments, the text has been reworded.

**1.5.2 Data**

Line 64: refer to refers

Done.

**Line 69** being pedantic 0h to 23h misses out one hour but I know what you mean, perhaps 0:00 to 23:59?

We have provided the times at which the data is produced by ERA5 and the model, i.e., 00:00 and 23:00. It is corrected in the new text.

**Line 74** Presumably three is some urban warming in the Paris observations. Please could you comment on how this affects the results somewhere, perhaps in section 6?

Indeed, three days correspond to the duration of the 2019 heatwave in France. In general, mortality increases sharply with the duration of heatwaves (D'Ippoliti et al., 2010), and a duration of three days allows us to capture this effect. Increasing the duration from a statistical point of view mainly amounts to reducing the sample size (the longer the heatwaves, the rarer they are), which limits inference. When choosing the variable, we added an explanation to this effect.

**1.5.3** Method**

**Line 106** I am confused by the phrase "We add to the model". Are you suggesting  $(X^R + X^G)$ ? If so, put it in the definition of eq 1. But if not perhaps "In addition, we can replace  $X^R$  with  $X^G$ .." or something similar would be clearer.

We meant that the model contains an additional element, which is the covariate  $X^{\varepsilon}G\varepsilon$ . We agree that this formulation might have been confusing. The entire Section 3 has been rewritten

**Line 111** English has gone a bit wrong here

The entire Section 3 has been rewritten.

**Line 123** The notation for the scenarios is grim. Could we not have  $X^{R,A,S_i}$  and define  $S_i$  elsewhere?

We have changed the notations to make them easier to read.

**Line 123** It would be good to acknowledge the unavoidable assumption that the climate system responds linearly to forcing. e.g  $\mu_t$  and  $\sigma_t$  are constant wrt different forcing scenarios.

You probably meant that  $\mu_0$ ,  $\mu_1$ ,  $\sigma_0$ ,  $\sigma_1$  and  $\xi_0$  are constants and do not depend on the scenario. This has been clarified in the entirely rewritten Section 3.

**Lines 124-130** This assumes all GCMs are equal. It would be good to acknowledge this and that other approaches have seen it necessary not to make this assumption (references below)

The entire Section 3 has been rewritten.

Lines 131-132 this has been said earlier.

Deleted.

**Lines 143-144** it would be good to say that the energy balance model is forced with natural forcings only and the radiative forcings are natural only.

It is now added in the revised version.

**Line 151**  $\theta^R$  and  $\theta^R$  - second should be  $\theta^G$ ?

Of course, corrected.

**Line 171** Fig S3 seems to indicate  $\sigma_t$  is very very small. Worth commenting on the physical significance I think.

You probably meant  $\sigma_1$ . This term is generally quite weak, although it can be stronger depending on the model. We discussed this in the new version, in Sect. 3.3.3.

**Lines 171,200** Fig S3 and Fig 2c-f and their discussion. I think it would help the reader if you reminded them that these plots are site specific using your Paris observations

It has been better highlighted.

**Line 177** I think you can only say impossible if the probability of the shape being  $\geq 0$  is zero. I don't think you have yet shown this.

Thanks you for this remark. The sentence has been removed.

**Line 189** *m* is used as a superscript and a subscript in this line. Is this what you mean?

Exactly, it is corrected.

Line 189 This is the first mention of bias between the climate models and truth which is very late in the paper. Too late. There should at least be a pointer earlier in the text that climate model bias is addressed later in the method description. Also I don't think you can deal with the issue of climate model bias in just 12 words! What sources of bias is it accounting for? In The GEV parameters? In the covariates? A description of how and how well is surely needed.

Indeed. A discussion on model biases has been added when climate models are introduced.

**Line 196** "grid point containing" earlier you were fitting to station obs for the plots (l137) - has it changed?

Thank you for pointing this out. This was an error on our part, it was indeed the observation station. However, this has been changed, as we now only keep ERA5 in the revised version, due to some inhomogenity in the Meteo-France time series.

Line 196 Fig2b: 1940-1960 All GCMs are cool wrt Obs. Please comment.

This could possibly indicate that the number of degrees of freedom of the splines is a little too low to capture this part of the signal. A comment to this effect has been added.

**Line 201 black ellipses missing**

Sorry, this has been corrected.

**Line 241 covariate FOR Europe**

Thanks, corrected.

**1.5.4 Comparison**

**Section 4** I'd suggest renaming to "Comparison with the independent scenarios method".

Thanks, but we have moved the section into the methodology section.

**Line 275** What are the consequences of  $\mu_1$  being so different for obs vs gcm? Some would argue that if the climate model is so biased can we trust its physical representation of real world extremes.

In fact,  $\mu_1$  is wide in the observations because it is very poorly estimated: it depends solely on the trend, which has been significant over the last 30 to 40 years. This is why it is much better constrained in the models, since the scenarios allow 80 years of data to be added. A comment to this effect has been added.

**Line 280** "Based on the estimates of the laws" - not sure this will mean very much to most people.

We have rephrased that to make it clearer.

**Line 302** "average energy" - I wonder if this is the right term in a geophysical journal? Joules?

We have removed Wasserstein distances to clarify the presentation.

**Line 304** I do wonder how much the average reader will get out of Fig 3 and I note that there is not that much discussion in the text for it. Perhaps just have the last column in a  $2 \times 2$  format?

This section has been completely redesigned.

**Line 314** "does a good job" - perhaps a bit too colloquial?

Yes, removed.

• Is there a low bias in ERA5 for some regions? eg UK had 40C in 2022 although this was only a single day. Kay et al. (2025) has much lower return periods for single day events and one would have thought that to first order changes in return period will be somewhat similar for different metrics of extreme hot temperatures.

This is entirely plausible, as in ERA5 the surface (i.e. the temperature at 2 m) is not explicitly resolved, but is an interpolation of the atmospheric reanalysis. Significant biases are observed, for example, in relation to E-OBS (see Fig. 1 at the end of the document).

• I think maps of GEV parameters would be very interesting to most readers, say for 2000, 2024, 2080? In the supplementary info if needs be.

The GEV parameter maps for the years 1850, 2000, 2024, and 2080 are now plotted for the four scenarios in figures 2-5 of the present answer document. We can see that the parameter  $\mu_t$  evolves over time and is most sensitive to climate change. The parameter  $\sigma_t$  is almost constant (since  $sigma_1$  is almost zero), while  $\xi_t$  is indeed constant over time. We do not believe that these figures add much value compared to the attribution in Section 6, which is why we did not include them in the revised version. Instead, we added a map of  $\mu_0$ ,  $\mu_1$ ,  $\sigma_0$ ,  $\sigma_1$ ,  $\xi_0$  and the 1961/1990 anomaly of TX3x in the new Fig. 4 of the article, as well as a text commenting on them.

**1.5.5 Example**

Fig 4 is an odd beast. It seems like it is implicitly assuming that the climate has been stationary between 1940 and 2024. For example two points nearish to each other might have seen the same max temperature at very different times, say 1940 and 2024 for arguments sake. The probability of those two events are very different as the 2024 climate is much hotter. Yet the calculation of 4c assumes they have the same probability of occurring. At least could we have a complimentary plot of the 100 year return level (or whatever) with and without human influence and the difference please? Also for the caption I found "mean of all scenarios" a bit confusing as during the observed period the forcings are the same? I think it would be clearer to say with and without human influence.

This figure does not assume climate stationarity between 1940 and 2024. Three elements allows the construction of these figures:

- inference of the GEV distribution of maxima (over 3 days) at each grid point, which allows for climate non-stationarity;
- the definition of an event class as the maximum observed throughout the series;
- and finally, the calculation of the probability of this event class, which depends on the year (given that we are in a non-stationary context).

In this figure in particular, we calculate the return periods and intensity changes for the year 2024 specifically, but the inference of the law was indeed non-stationary.

It is true that the expression 'mean of all scenarios' can cause some confusion, so we have rephrased it, and we have added the details in the text.

**Line 373** given the lack of spatial dependence perhaps a warning that these numbers cannot be used to calculate the likelihood of a hot event occurring in a given region or country without a correction to account for extremal dependence.

This warning has been added.

**Lines 383-387** I find this spatial variation in return periods across quite small distances alarming. Some of this will be due to the issue in my point above but it does not seem physical.

You could check how the observed exceedance rates compare between different locations. e.g.

- (a) Those areas where 4b shows very high return periods. What are the empirically observed exceedance rates? Are hot events occurring more frequently than predicted here?
- (b) Also for the regions in 4b with very frequent return rates, N Africa, E Turky are we seeing their frequent occurrence in the observations?
- (c) And eastern France and western Germany (approx. Nancy & Stuttgart) to see if the different return expectations from the plots in these two places are supported by the data.
- (d) Kay et al. (2025) found the 2022 UK record event of exceeding 40C (admittedly a single day maximum, but one would expect different averaging periods to be somewhat in step) to be 1 in 24 years which seems rather at odds with your plot (4b) if >500y for the region where these temperatures occurred. Comment?

It is true that these spatial variations appear to be very significant. In order to examine the situation in detail, we have plotted three grid points on Fig. 7 at the end of this document. One in Paris, the second in North Africa with a return period > 1000 years in 2024, and another nearby point with a return period

**Figure 1:** Difference between E-OBS (Cornes et al., 2018) and ERA5 (Hersbach et al., 2020) for the TX3x variable over Europe. **a)** Average difference over the period 1940–2024. **b)** Maximum difference over the period 1940–2024. **c)** TX3x series from E-OBS (blue) and ERA5 (red) in Paris. **d)** Difference between E-OBS and ERA5 in Paris between 1950 and 2024.

**Figure 2:** Map of the different parameters  $\mu_t$ ,  $\sigma_t$  and  $\xi_t$  in Factual and Counter-factual world of the GEV model after observational constraints, for the years  $t \in \{1850, 2000, 2024, 2080\}$ .

**Figure 3:** Map of the different parameters  $\mu_t$ ,  $\sigma_t$  and  $\xi_t$  in Factual and Counter-factual world of the GEV model after observational constraints, for the years  $t \in \{1850, 2000, 2024, 2080\}$ .

**Figure 4:** Map of the different parameters  $\mu_t$ ,  $\sigma_t$  and  $\xi_t$  in Factual and Counter-factual world of the GEV model after observational constraints, for the years  $t \in \{1850, 2000, 2024, 2080\}$ .

**Figure 5:** Map of the different parameters  $\mu_t$ ,  $\sigma_t$  and  $\xi_t$  in Factual and Counter-factual world of the GEV model after observational constraints, for the years  $t \in \{1850, 2000, 2024, 2080\}$ .

**Figure 6:** Map of the different parameters of the GEV model after observational constraints. **a)** Constant of the location parameter  $\mu_0$ . **b)** Trend of the location parameter  $\mu_1$ . **c)** Constant of the scale parameter  $\exp(\sigma_0)$ . **d)** Trend of the scale parameter  $\sigma_1$ . **e)** Constant of the shape parameter  $\xi_0$ . **e)** Bias of TX3x from ERA5 (mean over 1961 / 1990).

**Figure 7:** Comparison between observations and the inferred GEV distribution for three grid points (one per column). The position of the grid point is shown on the map (last row). The grid points are chosen, in order, in Paris, at a point where the maximum has a return period > 1000 years in 2024, at a point where the maximum has a return period < 10 years in 2024. The first 4 lines (representing, in order, the 4 scenarios SSP1-2.6 to SSP5-8.5) show ERA5 (black dots), the maximum value of ERA5 (black dotted line), as well as the following return levels: 2, 5, 10, 30, 50, 100, and 1000 years. Note that the scale is chosen to be comparable between the three columns (spread of  $20^{\circ}$ C). The fifth line shows the histogram of the p-values of the KS-test of 1000 samples compared to ERA5. The probability indicates the number of tests where the p-value is greater than 5% (threshold where we do not reject that the observations follow the inferred GEV law).

---

## Author Comment (AC5)

**Response to Reviewer 1 comments about the article "A Bayesian Statistical Method to Estimate the Climatology of Extreme Temperature under Multiple Scenarios: the ANKIALE Package"**

ROBIN, Y., VRAC, M., RIBES, A., BARBAUX, O. and NAVEAU, P.

October 23, 2025

**Note** In this document, the text in regular format corresponds to the reviewers questions. The answers from authors are given in the grey blocks.

**1 Reviewer 1 (Anonymous)**

**1.1 Global comments**

I think this is an important piece of work – it extends the use of observational constraints to the estimation of the characteristics of temperature extremes for unobserved past and future periods under both "factual" and "counterfactual" conditions, using a rigorous Bayesian framework in which extreme temperature distribution is described with a suitable extreme value distribution. The ANKIALE package that implements the method should help to make this sophisticated methodology relatively accessible to a broad range of users.

We would like to thank Reviewer 1 for this summary and for recognizing our efforts to make a Bayesian-based statistical analysis method for extreme events accessible to as many people as possible.

Unfortunately, however, the paper would be VERY challenging for the target audience to understand. For this work to be impactful, I think it will be necessary for the authors to think much more carefully about presentation issues, providing more complete and more accessible explanations for the choices that they have made in implementing the package. I think they also need to provide substantially more insight into choices users will have to make when applying the package, with an emphasis on physical considerations as well as statistical and pragmatic considerations. Also, to make the paper accessible to users will require a careful redesign of the notation that is used in the paper, which is hopelessly complex.

We thank Reviewer 1 for these relevant comments, which are in line with those made by the other reviewers. We apologize that the presentation was not as accessible as we would have liked. The paper has been reworked in this regard. We rewrote the paper in order to:

- Provide the minimum theoretical tools, referring to the appendix for technical details when possible,
- Better highlight the highly extensible nature of ANKIALE: ANKIALE currently only handles a GEV model, but other statistical models (suitable for other variables) can easily be added.

**1.2 Specific comments**

**Line 37** It is unclear why using different scenarios would result in different estimates of the counter-factual world.

It would help if a bit more were said here. Reading ahead, it turns out that this is discussed more beginning at line 113 and I can see from that discussion how this could arise. It is not made clear, however, whether the selection of a particular scenario would strongly affect inferences about observed events based on the posterior that results from using that scenario versus inferences that would be made with another scenario. Sensitivity to scenario choice, particularly if large (and especially under counterfactual conditions) would be of concern, but making that go away artificially by using information from all available scenarios doesn't really solve the problem in a satisfying way. It would remain a concern that information from models about the future can somehow affect our understanding of the past unless there is a convincing physical argument about why that makes sense. On the other hand, if the sensitivity is small then there wouldn't be a very compelling reason to bother with the added complexity of the prior and its dependence on the particular experimental design that was adopted in CMIP6. In summary, I think this is crucial point that needs clarification (and in each application, physical justification).

Indeed, our explanations here were a little brief. The estimation of the counterfactual always depends on the scenario. For example, a scenario such as SSP1-1.9 (not used here), which is very 'flat', makes the estimation of the parameters  $\mu_1$  and  $\sigma_1$  that drive the trend (and which are not supposed to depend on external forcings) much more uncertain than for an SSP5-8.5 scenario, where the trend is very strong. Theoretically, since  $\mu_1$  and  $\sigma_1$  do not depend on the scenario, they should be able to be estimated solely with the counterfactual covariate, restricted to the period 1850–1950 (where the anthropogenic term is very weak). However, in practice, this estimation is impossible: the design matrix during regression is ill-conditioned (no maximum rank). Our approach therefore has two advantages: it ensures a scenario-independent counterfactual, as well as estimation for scenarios where inference is difficult because the signal is weak.

Furthermore, knowledge brought by the simulations for the future period does indeed affect parameter estimation. Although this may seem strange from a physical point of view (breaking of causality), from a statistical point of view, the parameters estimation needs to rely on a continuous time series, including historical and future simulations. Therefore, the final estimation, even over the historical period, will depend on the future scenarios.

We added a sentence in the introduction stating that this applies in particular to confidence intervals, and this has been detailed in the methodology.

**Line 70** What is the point of the right-hand panel in Figure 1? It adds a bit of confusion by hinting that you will make inferences about the record event during 1940 to 2024 at every land point in the domain, irrespective of when that event happened or the spatial extent of

the event that produced the record.

The application example in Sect. 6 concerns the assignment of the maximum observed at each grid point, which is therefore given in Fig. 1b. We clarified the reference to the figure by indicating that Fig. 1b is used in Sect. 6.

**Line 75** Comparison with a single long record that is almost surely inhomogeneous (e.g., due to instrument changes, observing procedure changes, development of the urban environment around the station, etc) is not going to do a great deal to increase confidence.

After verification with Météo-France, it appears that this time series has indeed not been homogenised. Then, we redid the example with ERA5 in order to have a single data set.

**Line** 77 Usually, "external" forcing would mean external to the climate system (solar, volcanic, ghg's, aerosols ...) rather than external for France

That is right, we have rephrased it.

**Lines 77-79** If read literally, the sentence could be interpreted as saying that you extract European mean temperatures from HadCRUT5 and global mean temperatures (more correctly, temperature anomalies) from GISTEMP. Why do you use these two datasets rather than just using one?

In addition, Fig. S1 notes the use of the BEST dataset (Berkeley Earth) – why use yet another global dataset when consistent use of one, well regarded dataset, would probably suffice?

When we began this work, we used BEST to estimate the GMST. As BEST is known to have a warm bias, we then switched to GISTEMP. Finally, for calculations on a sub-region such as Europe, we used HadCRUT, which is the data set historically used for Europe in RR20. This explains the presence of GISTEMP and HadCRUT, and the fact that BEST still appears was an error. GISTEMP was retained (i) to provide a test example where the observation series do not cover the same period during the calculation, which made it possible to verify the algorithms, and (ii) because HadCRUT does not cover all the planet in the distant past.

We fixed the error for BEST, but kept two different datasets. A sentence has been added to Sect. 2.1 to explain why we have two data sets.

**Lines 85-86** My understanding is that the historical forcing prescription used in CMIP6 is NOT part of the SSPs, which only cover the period from 2015 onwards.

That is right, the SSPs only cover the period after 2015, with the period 1850–2014 being given by the so-called historical scenarios. It is corrected.

**Lines 90-91** I'm not aware that the IPCC assessed, in its synthesis report, that the current emissions trajectory is leading us towards SSP2-4.5. This is discussed by others, however, so I think you should provide a more suitable reference or delete this statement. Indeed, if you have high confidence in this statement, then it would seem that there would be no need to use the other scenarios.

Sentence deleted.

Line 101 The certainty expressed here that the variable of interest will be GEV distributed

seems a bit of an overstatement. The GEV distribution is a limiting distribution for block maxima that is (sometimes) achieved as the block length grows without bound. Convergence to the limiting distribution (if it happens at all) can only be demonstrated theoretically under very idealized conditions. Nature, and climate models, do not comply with those conditions (we have awkward things like an annual cycle and the presence of multiple extremes processes that complicate life considerably, with the result that the upper tail does not always behave like that expected under idealized mathematical conditions. While we can't really look into the deep upper tail with observations, and can do so, albeit with some difficulty, with climate models (Alaya et al., 2020). Experience shows that the GEV is nevertheless often useful for approximating the distribution of block maxima for blocks of even modest size (e.g., a year, which effectively only samples part of the year due to the annual cycle). The authors know all of this, and it would be good if some of this could be reflected in the paper, particularly as it is intended to introduce the methods and the ANKIALE package to a wide audience who are not as knowledgeable about the application of the GEV distribution and its limitations.

Indeed, this statement was exaggerated. The methodology section has been completely rewritten, and we are adding a discussion about this choice in the appendix.

**Line 103** What is the time range considered? Also, I find the notation here somewhat confusing. Readers in a hurry will confound the index F (for factual) with "future", and might confound the index "0" (zero) with "O" (for "observed"). A further question is whether readers should think of the three components of *X* as being random or fixed.

The time axis varies here from 1850 to 2100 for climate models and from 1950 to today for observations. The notations has been revised at the same time as the theoretical section.

**Line 109** The use of the \* to indicate the reader should make a substitution for R or G is awkward and mostly just makes comprehension a bit more difficult for the reader.

The notations has been revised at the same time as the theoretical section.

**Lines 114-115** Replace "supposed" with "assumed". Also, this assumption merits some discussion.

A discussion has been added.

**Lines 119-120** See the comment concerning line 37 above. This needs discussion – particularly why including different futures would affect our understanding of the past.

See our response to the comment concerning line 37 above.

**Lines 126-127** It might just be a French/English problem, but what this first step in the procedure entails could be better explained. This would include saying what the assumptions are that are implicit in calculating the uncertainty covariance matrix. It is not clear from the notation if there is one such covariance matrix that describes the spread amongst the different  $\theta_m$  (which I am guessing is the case) or whether each  $\theta_m$  has its own uncertainty matrix.

We have completely rewritten the entire methodology section.

**Lines 131-132** It would be much better if this paper could be self-contained rather than sending readers off to another reference for the parts of the methodology that have not changed.

The paper has now been rearranged to contain all the necessary information.

**Lines 137, 139** While the paper is generally readable, there are many minor grammatical errors. Two examples are mentioned here. This is less excusable these days given the wide availability of tools for polishing text (assuming that GMD authors are permitted to use them).

- At line 137, replace "3-days moving average" with "3-day moving averages".
- At line 139, where the sentence seems unclear. In that sentence, rather than "are", do you mean "are estimated with"?

We apologise for these errors and have corrected the text.

**Line 147** The white noise assumption needs some justification. This might be roughly suitable for European regional mean annual surface air temperature anomalies, but the while noise assumption seems a bit less obvious for annual global mean surface temperature anomalies.

This issue was addressed by Ribes et al. (2021) and Qasmi and Ribes (2022). Their conclusion is to use white noise in climate models, but to use a mixture of two AR(1) processes (a slow one for processes of the order of decades, a fast one for inter-annual processes) for the internal variability of observations during constraint. We have modified the text accordingly.

Lines 148-154 I find myself struggling to understand what is really done here, both because of the notation, which is increasingly complex, and because it is not obvious what model output is being used. If a model has 50 ensemble members, do you use all 50? And if so, do you treat that model differently from a model with only 1 ensemble member? What period is considered, how was the choice to use a smoothing spline with only 6 degrees of freedom made, how are the knots placed, do you worry about the fact that the knot placement is arbitrary and that this imposes wave-like fluctuations that are probably not part of the forcing response?

We have rewritten the whole Section 3. In a nutshell:

- All members of a model are used, and therefore models with one member are treated in the same way as those with 50 members. However, the latter have a lower degree of uncertainty, which affects the construction of the prior.
- The use of 6 degrees of freedom for splines was established in the original article by Ribes et al. (2020). It is true that this estimate was valid for the RCP 8.5 (CMIP5) scenario, with natural splines; whereas we use *B*-splines with different CMIP6 scenarios. ANKIALE has now been modified to be able to play on all these parameters simultaneously: base size and degree of freedom can be imposed for each scenario/covariate.

This information has been added to the article.

**Line 156** Figure S2 is referenced well before Fig S1...

It is now corrected.

Line 177 This statement is made with a lot of certainty and conviction, but whether an event would be judged to be impossible, even under anthropogenic forcing, is highly uncertain. It seems clear from Fig. S3 that the value of the shape parameter is driven by the extreme temperature that is farthest from the location parameter and hence must be very uncertain. This relation between the shape parameter and the most extreme temperature presumably occurs because the parameter estimation process enforces the feasibility of the fitted GEV distribution to a variable, temperature, that tends to have light-tailed extreme value distributions.

Indeed, it is well known that the GEV law tends to underestimate the true upper bound of a variable (if it exists). The sentence has been removed.

Lines 184-185 Why is this assumption needed to construct the prior? It's a prior distribution (i.e., a proposal) that will be updated using the observations when the posterior is derived. It seems to me that this assumption is not needed to construct the prior. Given the way the prior is constructed, it would certainly be helpful if we can regard the models as being indistinguishable from each other (i.e., something hopefully like a simple random sample from model space), but even without that, couldn't we construct a prior from the models, understanding that it may not do a good job of representing model uncertainty? The updating does require us to have a joint distribution for model simulated and observed quantities, and developing that joint model may require some additional assumptions – perhaps that's where the "indistinguishable from the truth" assumption comes into play?

Indeed, this hypothesis is not used in itself for the construction of the prior; it serves to justify that this multi-model synthesis method proposed by Ribes et al. (2017) constructs a prior in which the observations are hidden, and therefore that it is reasonable to take it as the prior of reality. The section has been rewritten.

**Lines 189-190** I think this needs discussion – in particular, why internal variability plays a role at all (haven't you filtered it out with the splines?) and what is being partitioned into two components. What is being referred to when you talk about the "common part" of internal variability and each model's additional internal variability??

It is assumed that the covariance matrix  $\Sigma_{\theta_m}$  of the random variable  $\theta^m$  of model m can be decomposed into the sum  $\Sigma_{\theta_m} = \Sigma_u + \Sigma_m$  of two terms of internal variability; This internal variability is not that of global or European average temperatures, but that of the  $\theta$  parameter vector. These two terms are:

- $\Sigma_u$ , which does not depend on the climate model;
- $\Sigma^m$ , which is a term specific to model m.

All of this has been rewritten and clarified in the supplementary material.

**Line 200** I have no idea what is being referred to here (95% of the covariance matrix)...

Apologies for the expression, we are of course referring to the ellipses defined by the covariance matrices, which correspond to the 95% quantile levels of the multivariate Gaussian distribution. We have edited the article for clarity.

**Line 202** Which model is excluded? Note that the UK models maybe be problematic due to a known problem in the coupling between the land surface and atmosphere that leads to

extreme high localized daily maximum temperatures. The problem is documented here1. Note that the Australian ACCESS models, which also use a version of the UK MetOffice atmospheric model, are not affected but the Korean KACE model, which uses both the atmospheric model and the Jules land surface model, is affected (an erratum has not been published for the KACE model).

Thanks for your relevant informations. In our case, this is the Norwegian Earth System Model NCC / NorESM2-LM. We have added this information to the article. (Seland et al., 2020).

**Lines 238-239** Why not use internal variability estimated from climate models rather than relying on the one, very limited realization we have been able to observe?

The Gaussian conditioning theorem requires the knowledge of an estimate for the internal variability of the observations used for the constraint. The internal variability of the models is not necessarily that of the observations.

¹https://errata.ipsl.fr/static/view.html?uid=76b3f818-d65f-c76b-bfd8-cae5bc27825c

**Bibliography**

- Alaya, M. A. B., F. Zwiers, and X. Zhang (Aug. 2020). "An Evaluation of Block-Maximum-Based Estimation of Very Long Return Period Precipitation Extremes with a Large Ensemble Climate Simulation". In: J. Clim. 33.16, pp. 6957–6970. ISSN: 0894-8755, 1520-0442. DOI: 10. 1175/JCLI-D-19-0011.1.
- Qasmi, S. and A. Ribes (Oct. 2022). "Reducing Uncertainty in Local Temperature Projections". In: *Sci. Adv.* 8.41, eabo6872. DOI: 10 . 1126 / sciadv.abo6872.
- Ribes, A., S. Qasmi, and N. P. Gillett (Jan. 2021). "Making Climate Projections Conditional on Historical Observations". In: *Sci. Adv.* 7.4, eabc0671. DOI: 10.1126/sciadv.abc0671.
- Ribes, A., S. Thao, and J. Cattiaux (2020). "Describing the Relationship between a Weather Event and Climate Change: A New Statistical Approach". In: J. Clim. 33.15, pp. 6297–6314. ISSN: 0894-8755, 1520-0442. DOI: 10.1175/JCLI-D-19-0217.1.

- Ribes, A., F. W. Zwiers, J.-M. Azaïs, and P. Naveau (2017). "A New Statistical Approach to Climate Change Detection and Attribution". In: *Clim Dyn* 48.1, pp. 367–386. ISSN: 1432-0894. DOI: 10. 1007/s00382-016-3079-6.
- Seland, Ø., M. Bentsen, D. Olivié, T. Toniazzo, A. Gjermundsen, L. S. Graff, J. B. Debernard, A. K. Gupta, Y.-C. He, A. Kirkevåg, J. Schwinger, J. Tjiputra, K. S. Aas, I. Bethke, Y. Fan, J. Griesfeller, A. Grini, C. Guo, M. Ilicak, I. H. H. Karset, O. Landgren, J. Liakka, K. O. Moseid, A. Nummelin, C. Spensberger, H. Tang, Z. Zhang, C. Heinze, T. Iversen, and M. Schulz (Dec. 2020). "Overview of the Norwegian Earth System Model (NorESM2) and Key Climate Response of CMIP6 DECK, Historical, and Scenario Simulations". In: *Geosci. Model Dev.* 13.12, pp. 6165–6200. ISSN: 1991-959X. DOI: 10.5194/gmd-13-6165-2020.